# Towards Understanding Why FixMatch Generalizes Better Than Supervised Learning

**Jingyang Li**[1]    **Jiachun Pan**[1]    **Vincent Y. F. Tan**[1]    **Kim-Chuan Toh**[1]    **Pan Zhou**[2]*

[1]National University of Singapore          [2] Singapore Management University

li_jingyang@u.nus.edu          pan.jiachun@outlook.com
{vtan,mattohkc}@nus.edu.sg          panzhou@smu.edu.sg

## Abstract

Semi-supervised learning (SSL), exemplified by FixMatch (Sohn et al., 2020), has shown significant generalization advantages over supervised learning (SL), particularly in the context of deep neural networks (DNNs). However, it is still unclear, from a theoretical standpoint, why FixMatch-like SSL algorithms generalize better than SL on DNNs. In this work, we present the first theoretical justification for the enhanced test accuracy observed in FixMatch-like SSL applied to DNNs by taking convolutional neural networks (CNNs) on classification tasks as an example. Our theoretical analysis reveals that the semantic feature learning processes in FixMatch and SL are rather different. In particular, FixMatch learns all the discriminative features of each semantic class, while SL only randomly captures a subset of features due to the well-known lottery ticket hypothesis. Furthermore, we show that our analysis framework can be applied to other FixMatch-like SSL methods, e.g., FlexMatch, FreeMatch, Dash, and SoftMatch. Inspired by our theoretical analysis, we develop an improved variant of FixMatch, termed Semantic-Aware FixMatch (SA-FixMatch). Experimental results corroborate our theoretical findings and the enhanced generalization capability of SA-FixMatch.

## 1 Introduction

Deep learning has made significant strides in various domains, including computer vision and natural language modeling (He et al., 2016; Vaswani et al., 2017; Radford et al., 2018; Dosovitskiy et al., 2020; Ho et al., 2020; Mildenhall et al., 2021; Ouyang et al., 2022; Schick et al., 2023). These advancements largely stem from scalable supervised learning, where increasing both network size and labeled dataset size typically enhances performance. However, in real-world scenarios, labeled data are often scarce. The benefits of larger datasets come at a high cost, as labeling requires human effort and can be prohibitively expensive, particularly in domains that rely on expert annotation (Sohn et al., 2020; Ouali et al., 2020; Zhou et al., 2020; Zhang et al., 2021a; Pan et al., 2022).

To address this challenge, semi-supervised learning (SSL) (Berthelot et al., 2019b; Sohn et al., 2020; Zhang et al., 2021a) has emerged as a promising solution, demonstrating effectiveness across various tasks. The methodology of SSL involves training a network on both labeled and unlabeled data, where pseudo-labels for the unlabeled data are generated during training. As a leading SSL approach, FixMatch (Sohn et al., 2020) first generates a pseudo-label using the current model's prediction on a weakly augmented unlabeled image. It then selects the highly-confident pseudo-label as the training label of the strongly-augmented version of the same image, and trains the model together with the vanilla labeled data. By accessing large amount of cheap unlabeled data with minimal human effort, FixMatch has effortlessly and greatly improved supervised learning. Moreover, thanks to its effectiveness and simplicity, FixMatch has inspired many SoTA FixMatch-like SSL works, e.g., FlexMatch (Zhang et al., 2021a), FreeMatch (Wang et al., 2022b), Dash (Xu et al., 2021), and SoftMatch (Chen et al., 2023), and is seeing increasing applications across many deep learning tasks (Xie et al., 2020; Xu et al., 2021; Schmutz et al., 2022; Wang et al., 2022b; Chen et al., 2023).

Despite FixMatch's practical success, its theoretical foundations lag behind its applications. Specifically, it remains unclear how FixMatch and its SL counterpart perform on deep neural networks,

---

*Corresponding author.

despite strong interest. Moreover, few theoretical studies investigate why SSL outperforms SL in test performance on networks, let alone FixMatch. Most existing works (He et al., 2022; Ţifrea et al., 2023) analyze oversimplified models, such as linear learners, which differ significantly from the highly nonlinear and non-convex networks used in real-world SSL. Consequently, these studies fail to capture FixMatch's learning mechanism. Other works (Rigollet, 2007; Van Engelen & Hoos, 2020; Guo et al., 2020) treat models as black-box functions under restrictive conditions, offering insights that do not account for the CNN dependencies crucial to FixMatch's superiority.

**Contributions.** To address these issues, we theoretically justify the superior test performance of FixMatch-like SSL over SL in classification tasks, using FixMatch as a case study. We analyze the semantic feature learning processes in FixMatch and SL, explaining their test performance differences and inspiring an improved FixMatch variant. Our key contributions are summarized as follows:

Firstly, we prove that FixMatch achieves superior test accuracy over SL on a three-layer CNN. Specifically, under the widely acknowledged multi-view data assumption (Allen-Zhu & Li, 2023), where multiple/single semantic features exist in multi/single-view data, FixMatch consistently achieves zero training and test classification errors on both multi-view and single-view data. In contrast, while SL achieves zero test classification error on multi-view data, it suffers up to $50\%$ test error on single-view data, showcasing FixMatch's superior generalization capacity compared to SL.

Secondly, our analysis highlights distinct feature learning processes between FixMatch and SL, directly affecting their test performance. We show that FixMatch comprehensively captures all semantic features within each class, virtually eliminating test classification errors. But SL learns only a partial set of these semantic features, and often fails on single-view samples due to the unlearned features, explaining its poor test classification accuracy on single-view data.

Finally, inspired by these insights, we introduce an improved version of FixMatch termed Semantic-Aware FixMatch (SA-FixMatch). This variant enhances FixMatch by masking learned semantics in unlabeled data, compelling the network to learn the remaining features missed by the current network. Our experimental evaluations confirm that SA-FixMatch achieves better generalization performance than FixMatch across various classification benchmarks.

## 2 RELATED WORKS

**Modern Deep SSL Algorithms.** Pseudo-labeling (Scudder, 1965; McLachlan, 1975) and consistency regularization (Bachman et al., 2014; Sajjadi et al., 2016; Laine & Aila, 2016) are the two important principles responsible for the success of modern deep SSL algorithms (Berthelot et al., 2019b;a; Xie et al., 2020; Zhang et al., 2021a; Xu et al., 2021; Wang et al., 2022b; Chen et al., 2023). FixMatch (Sohn et al., 2020), as a remarkable deep SSL algorithm, combines these principles with weak and strong data augmentations, achieving competitive results especially when labeled data is limited. Following FixMatch, several works, e.g., FlexMatch (Zhang et al., 2021a), Dash (Xu et al., 2021), FreeMatch (Wang et al., 2022b), and SoftMatch (Chen et al., 2023), try to improve FixMatch by adopting a flexible confidence threshold rather than the hard and fixed threshold adopted by FixMatch. These modern deep SSL algorithms can achieve remarkable test accuracy even trained with one labeled sample per semantic class (Sohn et al., 2020; Zhang et al., 2021a; Chen et al., 2023).

**SSL Generalization Error.** Previous works on generalization capacity of SSL focus on a general machine learning setting (Rigollet, 2007; Singh et al., 2008; Van Engelen & Hoos, 2020; Wei et al., 2020; Guo et al., 2020; Mey & Loog, 2022). In particular, the authors here view the model as a black-box function under certain assumptions which does not reveal the dependence on model design. Some recent works (He et al., 2022; Ţifrea et al., 2023) analyze the generalization performance of SSL under the binary Gaussian mixture data distribution for linear learning models. The over-simplified model is significantly different from the highly nonlinear and non-convex neural networks.

**Feature Learning.** Prior works on feature learning has shed light on how neural networks learn and represent data (Wen & Li, 2021; 2022; Allen-Zhu & Li, 2022; 2023). For instance, Allen-Zhu & Li (2023) explored how ensemble methods and knowledge distillation enhance generalization, while Wen & Li (2021) explained how contrastive learning captures sparse features and avoids spurious dense ones. Empirically, Park et al. (2018) proposed adversarial dropout to adapt neural networks based on their learning status, improving test performance. Despite these advances, to the best of our knowledge, this work is the first to analyze feature learning of neural networks in semi-supervised settings, with our theory-inspired SA-FixMatch achieving superior generalization performance.

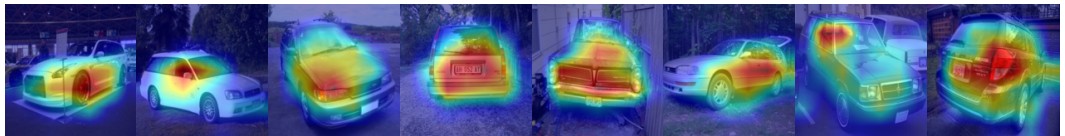

Figure 1: Visualization of pretrained ResNet-50 (He et al., 2016) using Grad-CAM. ResNet-50 locates different regions for different car images, e.g., wheel, rearview mirror, front light, and door.

## 3  PROBLEM SETUP

Here we first introduce the necessary multi-view data assumption used in this work, and then present FixMatch, a popular and classic SSL approach, to train a three-layered CNN on a $k$-class classification problem. For brevity, we use $O(\cdot), \Omega(\cdot), \Theta(\cdot)$ to hide constants w.r.t. $k$ and use $\tilde{O}(\cdot), \tilde{\Omega}(\cdot), \tilde{\Theta}(\cdot)$ to hide polylogarithmic factors. Let $\text{poly}(k)$ and $\text{polylog}(k)$ respectively denote $\Theta(k^C)$ and $\Theta(\log^C k)$ with a constant $C > 0$. We use $[n]$ to denote the set of $\{1, 2, \ldots, n\}$.

### 3.1  MULTI-VIEW DATA DISTRIBUTION

Following Allen-Zhu & Li (2023), we adopt the multi-view data assumption, which posits that each semantic class comprises multiple distinct semantic features—such as car lights and wheels—that can independently facilitate correct classification. To empirically validate this, we use Grad-CAM (Selvaraju et al., 2017) to identify class-specific regions in images. As shown in Figure 1, Grad-CAM highlights distinct, non-overlapping regions, such as different parts of a car, that contribute to recognition. These results support the findings of Allen-Zhu & Li (2023), confirming the existence of multiple independent semantic features within each class.

Now we introduce the multi-view data assumption in Allen-Zhu & Li (2023), which considers a dataset with $k$ semantic classes. Let each sample pair $(X, y)$ consists of the sample $X$, which is comprised of a set of $P$ patches $\{x_p \in \mathbb{R}^d\}_{p=1}^P$, and $y \in [k]$ as the class label. We assume each class $i$ has two semantic features, $v_{i,1}$ and $v_{i,2}$ in $\mathbb{R}^d$, capable of independently ensuring correct classification. While this analysis focuses on two features per class, the methodology extends to multiple features. Below we define $\mathcal{V}$ as the set of all semantic features across the $k$ classes:

$$\mathcal{V} = \{v_{i,1}, v_{i,2} \mid \|v_{i,1}\|_2 = \|v_{i,2}\|_2 = 1, v_{i,l} \perp v_{i',l'} \text{ if } (i,l) \neq (i',l')\}_{i=1}^k. \quad (1)$$

The conditions ensure the distinction of each class and its semantic features. Accordingly, we define the multi-view and singe-view distributions $\mathcal{D}_m$ and $\mathcal{D}_s$, where data from $\mathcal{D}_m$ has two semantic features, and those from $\mathcal{D}_s$ have only one. Set sparsity parameter $s = \text{polylog}(k)$ and constant $C_p$.

**Definition 1** (Informal, Data distribution (Allen-Zhu & Li, 2023)). *The data distribution $\mathcal{D}$ contains data from the multi-view data distribution $\mathcal{D}_m$ with probability $1 - \mu$, and data from the single-view data distribution $\mathcal{D}_s$ with probability $\mu = \frac{1}{\text{poly}(k)}$. We define $(X, y) \sim \mathcal{D}$ by randomly uniformly selecting a label $y \in [k]$ and generate data $X$ accordingly as follows.*

*(a) Sample a set of noisy features $\mathcal{V}'$ uniformly at random from $\{v_{i,1}, v_{i,2}\}_{i \neq y}$, each with probability $s/k$. Then the whole feature set of $X$ is $\mathcal{V}(X) = \mathcal{V}' \cup \{v_{y,1}, v_{y,2}\}$, i.e., the noisy feature set $\mathcal{V}'$ plus the main features $\{v_{y,1}, v_{y,2}\}$.*

*(b) For each $v \in \mathcal{V}(X)$, pick $C_p$ disjoint patches in $[P]$ and denote them as $\mathcal{P}_v(X)$. For a patch $p \in \mathcal{P}_v(X)$, we set $x_p = z_p v + \text{"noises"} \in \mathbb{R}^d$, where the coefficients $z_p \geq 0$ satisfy:*
*(b1) For "multi-view" data $(X, y) \in \mathcal{D}_m$, $\sum_{p \in \mathcal{P}_v(X)} z_p \in [1, O(1)]$ when $v \in \{v_{y,1}, v_{y,2}\}$ and $\sum_{p \in \mathcal{P}_v(X)} z_p \in [\Omega(1), 0.4]$ when $v \in \mathcal{V}(X) \setminus \{v_{y,1}, v_{y,2}\}$.*
*(b2) For "single-view" data $(X, y) \in \mathcal{D}_s$, pick a value $\hat{l} \in [2]$ randomly uniformly as the index of the main feature. Then $\sum_{p \in \mathcal{P}_v(X)} z_p \in [1, O(1)]$ when $v = v_{y,\hat{l}}$, $\sum_{p \in \mathcal{P}_v(X)} z_p \in [\rho, O(\rho)]$ ($\rho = k^{-0.01}$) when $v = v_{y,3-\hat{l}}$, and $\sum_{p \in \mathcal{P}_v(X)} z_p = \frac{1}{\text{polylog}(k)}$ when $v \in \mathcal{V}(X) \setminus \{v_{y,1}, v_{y,2}\}$.*
*(c) For each purely noisy patch $p \in [P] \setminus \cup_{v \in \mathcal{V}} \mathcal{P}_v(X)$, we set $x_p = \text{"noises"}$.*

For the details of Def. 1, please see Def. 7 in Appendix A, and also see more explanations in Appendix L. According to the definition, a multi-view sample $(X, y) \in \mathcal{D}_m$ has patches with two semantic features $v_{y,1}$ and $v_{y,2}$ plus some noises, while a single-view sample $(X, y) \in \mathcal{D}_s$ has patches with only one semantic feature $v_{y,1}$ or $v_{y,2}$ plus noises.

## 3.2 FixMatch for Training Neural networks

Here we introduce the representative SSL approach, FixMatch and its variants, on a $k$-class classification problem, the most popular task in SSL.

**Neural Network** For the network, we assume it as a three-layer CNN which has $mk$ convolutional kernels $\{w_{i,r}\}_{i\in[k],r\in[m]}$. Its classification probability $\mathbf{logit}_i(F, X)$ on class $i \in [k]$ is defined as

$$\mathbf{logit}_i(F, X) = \exp(F_i(X)) / \sum_{j\in[k]} \exp(F_j(X)), \tag{2}$$

where $F(X) = (F_1(X), \cdots, F_k(X)) \in \mathbb{R}^k$ is defined as

$$F_i(X) = \sum_{r\in[m]} \sum_{p\in[P]} \overline{\mathrm{ReLU}}(\langle w_{i,r}, x_p \rangle), \quad \forall i \in [k]. \tag{3}$$

Here $\overline{\mathrm{ReLU}}$ (Allen-Zhu & Li, 2023) is a smoothed ReLU that outputs zero for negative values, reduces small positive values to diminish noises, and maintains a linear relationship for larger inputs. This ensures $\overline{\mathrm{ReLU}}$ to focus on important features while filtering out noises. See details in Appendix A.

This three-layer network, comprising linear mapping, activation, and a softmax layer, captures essential neural network components, offering valuable insights into SSL training. Notably, many theoretical studies also use shallow networks (e.g., two-layer models) to gain insights into deep networks (Li & Yuan, 2017; Arora et al., 2019; Zhang et al., 2021b). Moreover, this setup matches the architecture in Allen-Zhu & Li (2023), enabling direct comparisons between our FixMatch results and their SL findings in Sec. 4.2.

**SSL Training** For FixMatch-like SSLs, at $t$-th iteration, it has two types of losses: 1) a supervised one $L_s^{(t)}$ on labeled data, and 2) an unsupervised one $L_u^{(t)}$ on unlabeled data. For $L_s^{(t)}$, it is cross-entropy loss on labeled dataset $\mathcal{Z}_l$:

$$L_s^{(t)} = \mathbb{E}_{(X_l,y)\sim\mathcal{Z}_l} L_s^{(t)}(X_l, y) = \mathbb{E}_{(X_l,y)\sim\mathcal{Z}_l}[-\log \mathbf{logit}_y(F^{(t)}, \alpha(X_l))], \tag{4}$$

where $\alpha(X)$ is a weak augmentation applied to input $X$. In practice, $\alpha(X)$ typically consists of a random horizontal flip and a random crop that retains most region of the image (Sohn et al., 2020; Zhang et al., 2021a), which often do not alter the semantic features. Hence, we treat the weak augmentation as an identity map to simplify our analysis. Additionally, experiments in Appendix K.2 confirm that weak augmentation has minimal impact on SSL training.

For the unsupervised loss $L_u^{(t)}(X_u)$, a weakly-augmented unlabeled sample $\alpha(X_u)$ is fed into the network to obtain classification probabilities $\mathbf{logit}_i(F^{(t)}, \alpha(X_u))$ for $i \in [k]$. If the maximal probability exceeds a confidence threshold $\mathcal{T}_t \in (0,1]$, i.e., $\max_i \mathbf{logit}_i(F^{(t)}, \alpha(X_u)) \geq \mathcal{T}_t$, FixMatch-like SSLs use it as the pseudo-label to supervise the corresponding strongly augmented sample $\mathcal{A}(X_u)$:

$$L_u^{(t)} = \mathbb{E}_{X_u\sim\mathcal{Z}_u} L_u^{(t)}(X_u) = \mathbb{E}_{X_u\sim\mathcal{Z}_u}[-\mathbb{I}_{\{\mathbf{logit}_b(F^{(t)},\alpha(X_u))\geq\mathcal{T}_t\}} \log \mathbf{logit}_b(F^{(t)}, \mathcal{A}(X_u))], \tag{5}$$

where $b$ is the pseudo-label $b = \arg\max_{i\in[k]}\{\mathbf{logit}_i(F^{(t)}, \alpha(X_u))\}$. Here, FixMatch-like SSLs use pseudo-labels generated from weakly-augmented samples to supervise the corresponding strongly-augmented ones, enforcing consistency regularization on model predictions, which we will show in Sec. 4.2 is crucial for the superior generalization performance of SSL compared to SL. For threshold $\mathcal{T}_t$, FixMatch (Sohn et al., 2020) sets it as a constant threshold $\mathcal{T}_t = \tau$ (e.g. 0.95) for high pseudo-label quality. Current SoTA SSLs, e.g., FlexMatch (Zhang et al., 2021a), FreeMatch (Wang et al., 2022b), Dash (Xu et al., 2021), and SoftMatch (Schick et al., 2023), follow FixMatch framework, and often design their own confidence threshold $\mathcal{T}_t$ in Eq. (5). This decides the applicability of our theoretical results on FixMatch in Sec. 4 to these FixMatch-like SSLs. See details in Appendix G.

For strong augmentation $\mathcal{A}(\cdot)$, it often uses CutOut (DeVries & Taylor, 2017) and RandAugment (Cubuk et al., 2020). CutOut randomly masks a large square region of the input image, potentially removing partial semantic features. Experimental results in Appendix K.1 confirm the significant impact of CutOut on image semantics and model performance. RandAugment includes various transformations, e.g., rotation, translation, solarization. Appendix K.1 reveals that those augmentations that may remove data semantics also have a large impact on model performance. Based on these findings, we model the probabilistic feature removal effect of $\mathcal{A}(\cdot)$ for our analysis in Sec. 4.1.

Now given the training loss $L^{(t)} = L_s^{(t)} + \lambda L_u^{(t)}$ at the $t$-th iteration, we adopt the widely used gradient descent (GD) to update the model parameters $\{w_{i,r}\}_{i\in[k],r\in[m]}$ in the network:

$$w_{i,r}^{(t+1)} = w_{i,r}^{(t)} - \eta\nabla_{w_{i,r}}L_s^{(t)} - \lambda\eta\nabla_{w_{i,r}}L_u^{(t)}, \tag{6}$$

where $\eta \geq 0$ is a learning rate, and $\lambda > 0$ is the weight to balance the two losses. According to the common practice (Sohn et al., 2020; Zhang et al., 2021a; Xu et al., 2021; Wang et al., 2022b; Chen et al., 2023), we set $\lambda = 1$ in both our theoretical analysis and experiments.

## 4 MAIN RESULTS

In this section, we first prove the superior generalization performance of FixMatch compared with SL. Next we analyze the intrinsic reasons for its superiority over SL via revealing and comparing the semantic feature learning process. Finally, inspired by our theoretical insights, we propose a Semantic-Aware FixMatch (SA-FixMatch) to better learn the semantic features.

### 4.1 RESULTS ON TEST PERFORMANCE

Here we analyze the performance of FixMatch, and compare it with its SL counterpart, whose implementation is simply setting the weight $\lambda = 0$ for the unsupervised loss in Eq. (6).

As discussed in Sec. 3.1, we assume the training dataset $\mathcal{Z}$ follows the multi-view distribution $\mathcal{D}$ (Def. 1), with multi-view and single-view sample ratios of $1 - \mu$ and $\mu$, respectively. Each class $i \in [k]$ has two i.i.d. semantic features $v_{i,1}$ and $v_{i,2}$, both of which are capable of predicting label $i$. Multi-view samples contain both features, while single-view samples have only one. For clarity, we denote the labeled multi-view and single-view subsets as $\mathcal{Z}_{l,m}$ and $\mathcal{Z}_{l,s}$, and the unlabeled subsets as $\mathcal{Z}_{u,m}$ and $\mathcal{Z}_{u,s}$. Below, we outline the necessary assumptions on the dataset and model initialization.

**Assumption 2.** *(a) The training dataset $\mathcal{Z}$ follows the distribution $\mathcal{D}$, and the size of the unlabeled data satisfies $N_u = |\mathcal{Z}_{u,m} \cup \mathcal{Z}_{u,s}| = |\mathcal{Z}_{l,m} \cup \mathcal{Z}_{l,s}| \cdot \mathrm{poly}(k)$.*

*(b) Each convolution kernel $w_{i,r}^{(0)}$ ($i \in [k]$, $r \in [m]$) is initialized by a Gaussian distribution $\mathcal{N}(0, \sigma_0^2 \mathbf{I})$, where $\sigma_0^{q-2} = 1/k$ and $q \geq 3$ is given in the definition of $\overline{\mathrm{ReLU}}$.*

Assumption 2(a) indicates that number of unlabeled data significantly exceeds that of the labeled data, a common scenario given the lower cost of acquiring unlabeled versus labeled data. The Gaussian initialization in Assumption 2(b) accords with the standard initialization in practice, and is mild. Moreover, we also need assumptions on the strong augmentation $\mathcal{A}(\cdot)$ to formulate the effect of consistency regularization in unsupervised loss Eq. (5).

**Assumption 3.** *Suppose for a given image, strong augmentation $\mathcal{A}(\cdot)$ randomly removes its semantic patches and noisy patches with probabilities $\pi_2$ and $1 - \pi_2$, respectively.*
*1) For a single-view image, the sole semantic feature is removed with probability $\pi_2$.*
*2) For a multi-view image, either of the two features, $v_{i,1}$ or $v_{i,2}$, is removed with probabilities $\pi_1\pi_2$ and $(1 - \pi_1)\pi_2$, respectively. We define strong augmentation $\mathcal{A}(\cdot)$ for multi-view data: for $p \in [P]$,*

$$\mathcal{A}(x_p) = \begin{cases} \max(\epsilon_1, \epsilon_2)x_p, & \text{if } v_{y,1} \text{ is in the patch } x_p, \\ \max(1 - \epsilon_1, \epsilon_2)x_p, & \text{if } v_{y,2} \text{ is in the patch } x_p, \\ (1 - \epsilon_2)x_p, & \text{otherwise (noisy patch)}, \end{cases} \tag{7}$$

*where $\epsilon_1$ and $\epsilon_2$ are i.i.d. Bernoulli variables, respectively equaling to 0 with probabilities $\pi_1$ and $\pi_2$.*

As discussed in Sec. 3, for strong augmentation $\mathcal{A}(\cdot)$, we focus on its probabilistic feature removal effect on the input image, caused by techniques like CutOut and certain operations in RandAugment, such as solarization. The use of the max function ensures that $\epsilon_1$ is active when $\epsilon_2 = 0$, indicating that $\mathcal{A}(\cdot)$ removes one feature at a time. Further details are provided in Appendix A.

Based on the above assumptions, we analyze the training and test performance of FixMatch, and summarize our main results in Theorem 4 with its proof in Appendix F.

**Theorem 4.** *Suppose Assumptions 2, 3 hold. For sufficiently large $k$ and $m = \mathrm{polylog}(k)$, setting $\eta \leq 1/\mathrm{poly}(k)$ and running FixMatch for $T = \mathrm{poly}(k)/\eta$ iterations ensures:*

*(a) Training performance is good. For all training samples $(X, y) \in \mathcal{Z}$, with probability at least $1 - e^{-\Omega(\log^2 k)}$, we have*

$$F_y^{(T)}(X) \geq \max_{j \neq y} F_j^{(T)}(X) + \Omega(\log k).$$

*(b) Test performance is good. With probability at least $1 - e^{-\Omega(\log^2 k)}$ over the selection of any multi-view test sample $(X, y) \sim \mathcal{D}_m$ and single-view test sample $(X, y) \sim \mathcal{D}_s$, we have*

$$F_y^{(T)}(X) \geq \max_{j \neq y} F_j^{(T)}(X) + \Omega(\log k).$$

Theorem 4(a) shows that after $T = \text{poly}(k)/\eta$ training iterations, the network $F^{(T)}$ trained by FixMatch can well fit the training dataset $\mathcal{Z}$, achieving zero classification error. Specifically, for any training sample $(X, y) \in \mathcal{Z}$, the predicted value $F_y(X)$ for the true label $y$ consistently exceeds the predictions $F_j(X)$ for all $j \neq y$, ensuring correct classification. More importantly, Theorem 4(b) establishes that the trained network $F^{(T)}$ can also accurately classify test samples $(X, y) \sim \mathcal{D}_m \cup \mathcal{D}_s$, validating FixMatch's strong generalization performance.

Now we compare FixMatch with SL (i.e. $\lambda = 0$ in Eq. (6)) under the same data distribution and the same network. According to Allen-Zhu & Li (2023), under the same assumption of Theorem 4, after running standard SL for $T = \text{poly}(k)/\eta$ iterations, SL can achieve good training performance as in Theorem 4(a). However, SL exhibits inferior test performance compared to FixMatch. Specifically, both methods achieve zero classification error on multi-view samples $(X, y) \sim \mathcal{D}_m$, while on single-view data $(X, y) \sim \mathcal{D}_s$, SL achieves only about 50% classification accuracy, significantly lower than FixMatch's nearly 100% accuracy. See Appendix B for more details on SL.

For other FixMatch-like SSLs such as FlexMatch (Zhang et al., 2021a), FreeMatch (Wang et al., 2022b), Dash (Xu et al., 2021), and SoftMatch (Chen et al., 2023), our theoretical results in Theorem 4 and the comparison with SL are also broadly applicable. Due to space limitations, we defer the discussions to Appendix G. These theoretical results justify the superiority of FixMatch-like SSLs over SL, aligning with empirical evidence from several studies (Sohn et al., 2020; Zhang et al., 2021a; Wang et al., 2022b; Xu et al., 2021; Chen et al., 2023).

## 4.2 RESULTS ON FEATURE LEARNING PROCESS

Here we analyze the feature learning process in FixMatch and SL, and explain their rather different test performance as shown in Sec. 4.1. To monitor the feature learning process, we define

$$\Phi_{i,l}^{(t)} := \sum\nolimits_{r \in [m]} [\langle w_{i,r}^{(t)}, v_{i,l} \rangle]^+, \quad i \in [k], \quad l \in [2]$$

as an indicator of feature learning for $v_{i,l}$ in class $i$. It represents the total positive correlation between feature $v_{i,l}$ and all $m$ convolution kernels $w_{i,r}$ ($r \in [m]$) at iteration $t$. A larger $\Phi_{i,l}^{(t)}$ indicates better capture and utilization of $v_{i,l}$ for classification. See Appendix D for further discussion.

Next, FixMatch applies a confidence threshold ($\tau$) to regulate the unsupervised loss in Eq. (5), dividing its feature learning process into Phase I and Phase II. In Phase I, the network relies primarily on supervised loss, as it cannot yet generate confident pseudo-labels. As training progresses, the network learns partial features, improving its ability to predict confident pseudo-labels for unlabeled data. This transition marks the start of Phase II, where the unsupervised loss plays a larger role, driven by consistency regularization between weakly and strongly augmented samples.

Now we are ready to present the feature learning process of FixMatch and SL in Theorem 5.

**Theorem 5.** *Suppose Assumptions 2, 3 hold. For sufficiently large $k$ and $m = \text{polylog}(k)$, setting $\eta \leq 1/\text{poly}(k)$ and $\tau = 1 - \tilde{O}(1/s^2)$ ensures that, with probability at least $1 - e^{-\Omega(\log^2 k)}$:*

*(a) FixMatch. At the end of Phase I, which runs for $T_1 = \text{poly}(k)/\eta$ iterations,*

$$\Phi_{i,l}^{(T_1)} \geq \Omega(\log k), \quad \Phi_{i,3-l}^{(T_1)} \leq 1/\text{polylog}(k), \quad \forall i \in [k], \exists l \in [2]. \tag{8}$$

*After Phase II, which runs for another $T_2 = \text{poly}(k)/\eta$ iterations,*

$$\Phi_{i,l}^{(T_1+T_2)} \geq \Omega(\log k), \quad \forall i \in [k], \forall l \in [2]. \tag{9}$$

*(b) Supervised Learning. After $T \geq \text{poly}(k)/\eta$ iterations, Eq. (8) always holds.*

See its proof in Appendix F. Theorem 5(a) indicates that Phase I in FixMatch continues for $T_1 = \text{poly}(k)/\eta$ iterations. During this phase, the network learns only one of the two semantic features per class. Specifically, in Eq. (8), for any class $i \in [k]$, there exists an index $l \in [2]$ so that the correlation score $\Phi_{i,l}^{(T_1)}$ exceeds $\Omega(\log k)$, showing feature $v_{i,l}$ is captured; and the score $\Phi_{i,3-l}^{(T_1)}$ remains low, indicating failure of learning $v_{i,3-l}$. Then we analyze classification performance when Eq. (8) holds.

**Corollary 6.** *Under the same conditions as Theorem 5. Assume Eq. (8) holds for the trained network $F^{(T)}$. For any sample $X$ from class $i$ containing the feature $v_{i,l}$, the network $F^{(T)}$ can correctly predict label $i$. Conversely, if $X$ contains only the feature $v_{i,3-l}$, $F^{(T)}$ would misclassify $X$.*

See its proof in Appendix D. According to Corollary 6, after Phase I, the network can correctly classify multi-view samples, as each contains two semantic features and the network learns at least one of them. However, for single-view samples, which contain only one semantic feature, the classification accuracy is around 50% since the network may not learn the specific feature present. Then by running another $T_2$ iterations in Phase II, Theorem 5(a) shows that FixMatch enables the network to capture both semantic features $v_{i,1}$ and $v_{i,2}$ for each class $i \in [k]$. As indicated by Eq. (9), all features achieve large correlation scores $\Phi_{i,l}^{(T_1+T_2)}$ for all $i \in [k], l \in [2]$. Therefore, by Corollary 6, the network trained by FixMatch can correctly classify all training and test samples with high probability, explaining the strong generalization performance observed in Theorem 4.

For Phase II of FixMatch, the reason for it to learn the semantic features missed in Phase I is as follows. Having learned one semantic feature per class in Phase I, the network is capable of generating highly confident pseudo-labels for weakly-augmented multi-view samples. As the confidence threshold $\tau = 1 - \tilde{O}(1/s^2)$ is close to 1 (e.g., $\tau = 0.95$), it ensures the correctness of these pseudo-labels. Then, FixMatch uses these correct pseudo-labels to supervise the corresponding strongly-augmented samples via consistency regularization. As shown in Eq. (7), strong augmentation $\mathcal{A}(\cdot)$ randomly removes the learned features in unlabeled multi-view samples with probabilities $\pi_1\pi_2$ or $(1-\pi_1)\pi_2$, effectively converting these samples into single-view data containing the unlearned feature. Given the large volume of unlabeled data as specified in Assumption 2, these transformed single-view samples are significant in their size. Accordingly, they dominate the unsupervised loss, since the rest samples containing the learned feature are already correctly classified by the network after Phase I and contribute minimally to the training loss. Consequently, the unsupervised loss enforces the network to learn the unlearned feature in Phase II.

For SL, Theorem 5(b) shows that with high probability, SL learns only one of the two features for each class, consistent with Phase I in FixMatch. By Corollary 6, SL can correctly classify multi-view data using the single learned feature but achieves only about 50% test accuracy on single-view data due to the unlearned feature, aligning with Sec. 4.1. In contrast, FixMatch achieves nearly 100% test accuracy on both multi-view and single-view data, as it learns both semantic features for each class.

The key difference between FixMatch and SL lies in the additional unsupervised loss, which essentially serves as consistency regularization. Beyond SSL, consistency regularization also plays a crucial role in other learning paradigms. For example, in self-supervised learning, it promotes the acquisition of richer semantic features during pretraining. Our theoretical analysis offers valuable insights into these broader settings, and we leave the exploration as future work.

**Comparison to Other SSL Analysis.** This work differs from previous works from two key aspects. (a) Our work provides the first analysis for FixMatch-like SSLs on CNNs. In contrast, many other works (He et al., 2022; Ţifrea et al., 2023) analyze over-simplified models, e.g., linear learning models, that differs substantially from the highly nonlinear and non-convex networks used in SSL. Some other works (Rigollet, 2007; Singh et al., 2008; Van Engelen & Hoos, 2020) view the model as a black-box function and do not reveal insights to model design. (b) This work is also the first one to reveal the feature learning process of SSL, deepening the understanding to SSL and unveiling the intrinsic reasons of the superiority of SSL over its SL counterpart.

## 4.3    Semantic-Aware FixMatch

The analysis of feature learning Phase II in Sec. 4.2 shows the crucial role of strong augmentation $\mathcal{A}(\cdot)$ via consistency regularization in Eq. (5) to learn the features missed in Phase I. However, according to Eq. (7), $\mathcal{A}(\cdot)$ only removes the learned feature with probabilities $\pi_1\pi_2$ or $(1-\pi_1)\pi_2$. This means given $N_{u,m}$ unlabeled multi-view samples, $\mathcal{A}(\cdot)$ can generate at most $N_{\mathcal{A}} = \max(\pi_1\pi_2, (1-\pi_1)\pi_2)N_{u,m}$ samples containing only the missed features to enforce the network to learn them in Phase II. So FixMatch does not fully utilize unlabeled data in Phase II to learn comprehensive features, especially when $\pi_2$ is small, which usually happens when semantics only occupy a small portion of the image so that strong augmentation $\mathcal{A}(\cdot)$ like CutOut (DeVries & Taylor, 2017) and RandAugment has small probability to remove semantics (e.g., in ImageNet, see Appendix K.6).

Motivated by this finding, we propose Semantic-Aware FixMatch (SA-FixMatch) to improve the probability of removing learned features by replacing random CutOut in FixMatch's strong augmentation $\mathcal{A}(\cdot)$ with Semantic-Aware CutOut (SA-CutOut). Specifically, if the unlabeled sample $X$ has highly confident pseudo-label, SA-CutOut first performs Grad-CAM (Selvaraju et al., 2017) on the

network $F$ to localize the learned semantic regions which contribute to the network's class prediction and can be regarded as features. Then for each semantic region, SA-CutOut finds its region center, i.e., the point with highest attention score in the region, and then averages attention score within a $q \times q$ bounding box centered at this point (e.g., $q = 16$). Finally, SA-CutOut selects one semantic region with the highest average score for masking. Here masking semantic region with the highest score can enforce the network to learn the remaining features that are not well learned or missed in Phase I, as they will not be detected by Grad-CAM or detected with relatively low attention scores. In this way, SA-FixMatch can enhance vanilla FixMatch to better use unlabeled data to learn comprehensive semantic features. For analysis, see our formulation of $\mathcal{A}(\cdot)$ with SA-CutOut in Appendix E.

In Theorem 13 of Appendix A, we prove that SA-FixMatch enjoys the same good training and test accuracy in Theorem 4, but reduces the required number of unlabeled data samples $N_u$ in vanilla FixMatch to $N_c = \max\{\pi_1 \pi_2, (1 - \pi_1)\pi_2\}N_u$, where $N_u$ is given in Assumption 2. This data efficiency stems from SA-FixMatch's use of SA-CutOut, which selectively removes the well-learned features, thereby compelling the network to focus on learning previously missed or unlearned features. Detailed theoretical discussions and proofs are presented in Appendix E, illustrating how SA-FixMatch outperforms vanilla FixMatch in terms of data efficiency and better test performance.

Moreover, our analysis of SA-FixMatch remains valid even when the labeled and unlabeled datasets contain the same images, as SA-CutOut deterministically removes the learned features. This ensures the network continues to learn new semantic features in Phase II even with the same images as unlabeled data. Experiments in Sec. 5.4 further confirm that SA-FixMatch outperforms FixMatch and SL in this setting, validating the effectiveness of SA-CutOut in enhancing feature learning.

As discussed in Sec. 3.2, SoTA deep SSLs, including FlexMatch (Zhang et al., 2021a), FreeMatch (Wang et al., 2022b), Dash (Xu et al., 2021), and SoftMatch (Chen et al., 2023), often build upon FixMatch, and only modify the confidence threshold $\mathcal{T}_t$ in Eq. (5). Hence, SA-CutOut is also applicable to these FixMatch-like SSLs to enhance performance. Experimental results in Sec. 5.3 validates the effectiveness and compatibility of SA-CutOut.

## 5 EXPERIMENTS

To corroborate our theoretical results, we evaluate SL, FixMatch, and SA-FixMatch on CIFAR-100 (Krizhevsky et al., 2009), STL-10 (Coates et al., 2011), Imagewoof (Howard & Gugger, 2020), and ImageNet (Deng et al., 2009). Following standard SSL evaluation protocols (Sohn et al., 2020; Zhang et al., 2021a; Wang et al., 2022a), we use WRN-28-8 (Zagoruyko & Komodakis, 2016) for CIFAR-100, WRN-37-2 (Zhou et al., 2020) for STL-10 and Imagewoof, and ResNet-50 (He et al., 2016) for ImageNet. We also apply SA-CutOut to other FixMatch-like SSL methods and compare their performance against the originals. All experiments are repeated three times, reporting the mean and standard deviation. Further experimental details are provided in Appendix K.4 and K.5.

### 5.1 CLASSIFICATION RESULTS

Here we evaluate the generalization performance of SL, FixMatch, and SA-FixMatch under varying amounts of labeled data. Following standard SSL benchmarks (Sohn et al., 2020; Zhang et al., 2021a; Chen et al., 2023), we use the entire training dataset as unlabeled dataset.

Table 1 shows that on STL-10, FixMatch and SA-FixMatch outperform SL by over 28% in test accuracy across all settings. Similar substantial improvements are observed on other datasets, such as 13%+ on CIFAR-100 and Imagewoof, and 6%+ on ImageNet. These results highlight the superiority of SSL methods over conventional SL and align with our theoretical findings in Sec. 4.1.

Meanwhile, Table 1 shows that SA-FixMatch outperforms vanilla FixMatch across all datasets. On Imagewoof, it improves average test accuracy by over 1.5%, while on ImageNet, it achieves a 1.38% gain. On CIFAR-100 and STL-10, SA-FixMatch also consistently surpasses FixMatch, though with a smaller margin. This variation arises because, in CIFAR-100 and STL-10, the semantic subject occupies most of the image (see Appendix K.6), allowing a random square mask in CutOut to effectively remove partial semantic features with high probability, producing similar masking effects as SA-CutOut. However, reducing the CutOut mask size lowers the likelihood of masking semantic features, leading to performance degradation (Table 4). In contrast, on Imagewoof and ImageNet, where the semantic subject occupies less than a quarter of the image (see Appendix K.6), a random square mask in CutOut is less likely to remove semantic features, making SA-CutOut significantly more effective and resulting in SA-FixMatch achieving much better test performance than FixMatch.

Table 1: Comparison of Test Accuracy (%) using the entire training dataset as unlabeled data.

| Dataset | CIFAR-100 | | | STL-10 | | | Imagewoof | | | ImageNet |
|---|---|---|---|---|---|---|---|---|---|---|
| Label Amount | 400 | 2500 | 10000 | 40 | 250 | 1000 | 250 | 1000 | 2000 | 100K |
| SL | $11.45 \pm 0.12$ | $40.45 \pm 0.50$ | $63.77 \pm 0.29$ | $23.61 \pm 1.62$ | $38.83 \pm 1.12$ | $64.08 \pm 0.47$ | $25.94 \pm 1.54$ | $42.04 \pm 0.90$ | $60.77 \pm 1.04$ | $44.62 \pm 1.16$ |
| FixMatch | $55.16 \pm 0.63$ | $71.36 \pm 0.44$ | $77.25 \pm 0.22$ | $70.00 \pm 4.02$ | $88.73 \pm 0.92$ | $93.45 \pm 0.19$ | $43.00 \pm 1.46$ | $64.91 \pm 1.18$ | $74.05 \pm 0.15$ | $50.80 \pm 0.73$ |
| SA-FixMatch | $55.57 \pm 0.43$ | $72.12 \pm 0.20$ | $77.46 \pm 0.16$ | $71.81 \pm 4.23$ | $89.45 \pm 1.19$ | $94.04 \pm 0.19$ | $46.73 \pm 1.36$ | $67.76 \pm 1.29$ | $75.62 \pm 0.13$ | $52.18 \pm 0.32$ |

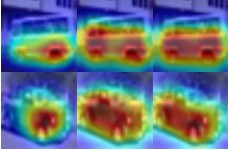 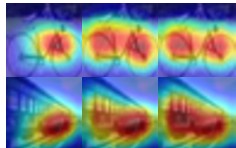 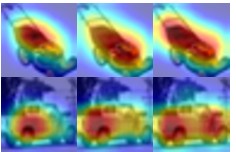 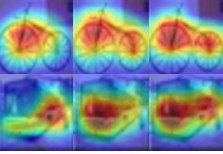

Figure 2: Visualization of WRN-28-8 via Grad-CAM on CIFAR-100. Each group of three images corresponds to models trained with SL (left), FixMatch (middle), and SA-FixMatch (right).

The superior test accuracy of SA-FixMatch over FixMatch aligns with our theoretical analysis in Sec. 4.3. To achieve strong test performance in Theorem 4, Phase II of SSL must effectively remove well-learned features to enforce learning of missed semantic features from Phase I. While FixMatch relies on CutOut to randomly mask learned features, SA-FixMatch consistently masks them using SA-CutOut (Sec. 4.3). As a result, with a fixed unlabeled dataset size, SA-FixMatch utilizes unlabeled data more effectively for feature learning, leading to better test performance.

## 5.2 SEMANTIC FEATURE LEARNING

To visualize the semantic features learned by networks trained by SL, FixMatch, and SA-FixMatch, we use Grad-CAM (Selvaraju et al., 2017) to highlight regions of input images that contribute to the model's class-specific predictions. For SL, FixMatch, and SA-FixMatch, we follow the default setting of Grad-CAM, and apply it to the last convolutional layer of the WRN-28-8 network on CIFAR-100.

Figure 2 shows that the network trained by SL often captures a single semantic feature since Grad-CAM only localizes one small image region, e.g., bicycle front wheel. Differently, networks trained by FixMatch can often grab multiple features for some classes, e.g., bicycle front and back wheels, but still misses some features for certain classes, e.g., bus compartment. By comparison, networks trained by SA-FixMatch reveals better semantic feature learning performance, since it often captures multiple semantic features, e.g., bicycle front and back wheels, bus front and compartment. The reason behind these phenomena is that as theoretically analyzed in Sec. 4.2, for classes which have multiple semantic features, SL can only learn a single semantic feature, while FixMatch and SA-FixMatch are capable of learning all the semantic features via the two-phase (supervised and unsupervised) learning process. Moreover, as shown in Sec. 4.3, compared with FixMatch, SA-FixMatch can more effectively use unlabeled data as it better removes well-learned features for enforcing network to learn missed features in data. Thus, SA-FixMatch is more likely to capture all semantic features of the data in practice with a fix number of unlabeled training data as observed in Figure 2.

## 5.3 SA-CUTOUT ON FIXMATCH VARIANTS

SA-CutOut is compatible with other deep SSL methods, such as FlexMatch (Zhang et al., 2021a), FreeMatch (Wang et al., 2022b), Dash (Xu et al., 2021), and SoftMatch (Chen et al., 2023), since as discussed in Sec. 3.2, the main difference between these deep SSL methods and FixMatch is their choice of confidence threshold $\mathcal{T}_t$. Here we apply SA-CutOut to these algorithms and compare their test accuracies with the original methods on STL-10 and CIFAR-100 dataset. From Table 2, one can observe that on STL-10, application of SA-CutOut increases the test accuracies of FlexMatch and FreeMatch by 2.6%+, and the test accuracies of Dash and SoftMatch by 5.4%+. On CIFAR-100, SA-CutOut increases the test accuracies of FreeMatch and Dash by 0.65%+, SoftMatch and FlexMatch by 0.5%+. This validates our analysis in Sec. 4.3 that SA-CutOut can more effectively use unlabeled data to learn comprehensive semantic features and thereby achieve higher test accuracy.

## 5.4 ABLATION STUDY

**Same Training Dataset** We evaluate (SA-)FixMatch and SL under the same training dataset setting, as described in Sec. 4.3. As shown in Table 3, SA-FixMatch significantly outperforms SL, reaffirming

Table 2: Comparison of Test accuracy (%) of SSL algorithms with CutOut and SA-CutOut on STL-10 with 40 labeled data and CIFAR-100 with 400 labeled data.

| Dataset | STL-10 | | | | CIFAR-100 | | | |
|---|---|---|---|---|---|---|---|---|
| Algorithm | FlexMatch | FreeMatch | Dash | SoftMatch | FlexMatch | FreeMatch | Dash | SoftMatch |
| CutOut | $72.13 \pm 5.66$ | $75.29 \pm 1.29$ | $67.51 \pm 1.47$ | $78.55 \pm 2.90$ | $59.65 \pm 1.14$ | $58.44 \pm 1.92$ | $48.56 \pm 2.16$ | $60.16 \pm 2.22$ |
| SA-CutOut | $75.91 \pm 5.59$ | $77.91 \pm 2.01$ | $78.41 \pm 1.91$ | $84.04 \pm 4.67$ | $60.16 \pm 1.06$ | $59.12 \pm 1.69$ | $50.24 \pm 1.82$ | $60.69 \pm 1.95$ |

Table 3: Comparison of Test Accuracy (%) using the same training dataset for (SA-)FixMatch and SL.

| Dataset | STL-10 | | | CIFAR-100 | | | ImageNet |
|---|---|---|---|---|---|---|---|
| Data Amount | 40 | 250 | 1000 | 400 | 2500 | 10000 | 100K |
| SL | 19.93 | 44.06 | 67.29 | 9.87 | 40.98 | 63.48 | 41.82 |
| FixMatch | 38.88 | 64.70 | 79.15 | 18.58 | 47.20 | 67.94 | 43.34 |
| SA-FixMatch | 40.25 | 65.85 | 79.74 | 19.72 | 47.71 | 68.30 | 44.88 |

Table 4: Effect of (SA-)CutOut mask size on test accuracy (%) on CIFAR-100 with 400 labeled data.

| Mask Size | 4 | 8 | 12 | 16 |
|---|---|---|---|---|
| FixMatch | 48.65 | 50.11 | 53.48 | 55.23 |
| SA-FixMatch | 52.71 | 52.95 | 55.37 | 55.78 |

the superiority of SSL over SL and further validating our theoretical insights in Sec. 4.3. Moreover, SA-FixMatch surpasses FixMatch, demonstrating the effectiveness of our proposed method.

**Maske Size**    Here we investigate the effect of the mask size in (SA-)CutOut on the performance of (SA-)FixMatch. For CIFAR-100 whose image size is $32 \times 32$, we set the mask size in (SA-)CutOut as 4, 8, 12, and 16 to train the WRN-28-8 network. Table 4 shows that 1) as mask size grows, both the test accuracy of FixMatch and SA-FixMatch improves; 2) when mask size is small, SA-FixMatch makes significant improvement over FixMatch, e.g., 4%+ when using a mask size of 4; 3) as mask size grows, the improvement of SA-FixMatch over FixMatch becomes reduced, e.g., 0.55% when using a mask size of 16. For 1), as mask size in (SA-)CutOut increases, the learned features in the image are more likely to be removed, which is the key for (SA-)FixMatch to learn comprehensive semantics in Phase II as analyzed in Sec. 4.2. This explains the better performance of FixMatch and SA-FixMatch when their mask sizes increase. For 2), when using small masks, a random mask in CutOut has much lower probability to remove learned features compared with SA-CutOut. Thus, SA-FixMatch has much better performance than FixMatch. For 3), as mask size grows, a random mask in CutOut also has large probability to mask learned features in the image. This explains the reduced gap between SA-FixMatch and FixMatch.

## 6    CONCLUSION

By examining the classical FixMatch, we first provide theoretical justifications for the superior test performance of SSL over SL on neural networks. Then we uncover the differences in the feature learning processes between FixMatch and SL, explaining their distinct test performances. Inspired by theoretical insights, a practical enhancement called SA-FixMatch is proposed and validated through experiments, showcasing the potential for our newly developed theoretical understanding to inform improved SSL methodologies. Apart from FixMatch-like SSL, there are also other effective SSL frameworks whose analyses and comparisons are left as our future work.

**Limitations.** (a) Apart from FixMatch-like SSLs, we did not analyze other SSL frameworks, like MeanTeacher (Tarvainen & Valpola, 2017) and MixMatch (Berthelot et al., 2019b). However, current SoTA deep SSLs like FlexMatch, FreeMatch, Dash, and SoftMatch all follow the FixMatch framework, indicating the generalizability of our theoretical analysis on them. See details in Sec. 3.2 and Appendix G. (b) Due to limited GPU resources, we use small datasets, e.g. STL-10 and CIFAR-100, instead of large datasets like ImageNet to test SA-CutOut on other SoTA SSLs. Future work involves testing SA-CutOut on other SSLs methods (other than FixMatch) and on larger datasets.

## ACKNOWLEDGMENTS AND AUTHOR CONTRIBUTIONS

J. Pan and V. Y. F. Tan are supported by a Singapore Ministry of Education (MOE) AcRF Tier 2 grant (A-8000423-00-00). J. Li led the development of the theoretical framework, provided the rigorous proofs, and also led the experimental design and implementation. J. Pan contributed the initial sketch of the main proof. All authors discussed the results and contributed to the final manuscript.

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

## A    THEOREM STATEMENT

In this section, we formally state the relevant data assumptions and theorems. Building on the proof framework of Allen-Zhu & Li (2023), our results extend their findings from supervised learning (SL) to semi-supervised learning (SSL). To maintain consistency, we adopt their notation throughout our proof. Specifically, we follow their data distribution assumptions and extend their analysis from SL to SSL through a two-phase learning process.

To formally define the data distribution, we set global constant $C_p$, sparsity parameter $s = \text{polylog}(k)$, feature noise parameter $\gamma = \frac{1}{\text{poly}(k)}$, and random noise parameter $\sigma_p = \frac{1}{\sqrt{d}\,\text{polylog}(k)}$ to control noises in data. Here, feature noise implies that a sample from class $i$ primarily exhibits feature $v_{i,l}$ (with $l \in [2]$), but also includes minor scaled features $v_{j,l}$ (with $j \neq i$) from other classes. Each sample pair $(X, y)$ consists of the sample $X$, which is comprised of a set of $P = k^2$ patches $\{x_i \in \mathbb{R}^d\}_{i=1}^{P}$, and $y \in [k]$ as the class label. The following describes the data generation process.

**Definition 7** (data distributions for single-view $\mathcal{D}_s$ and multi-view data $\mathcal{D}_m$ (Allen-Zhu & Li, 2023)). *Data distribution $\mathcal{D}$ consists of data from multi-view data $\mathcal{D}_m$ with probability $1 - \mu$ and from single-view data $\mathcal{D}_s$ with probability $\mu = 1/\text{poly}(k)$. We define $(X, y) \sim \mathcal{D}$ by randomly uniformly selecting a label $y \in [k]$ and generating data $X$ as follows.*

1) *Sample a set of noisy features $\mathcal{V}'$ uniformly at random from $\{v_{i,1}, v_{i,2}\}_{i \neq y}$ each with probability $s/k$.*

2) *Denote $\mathcal{V}(X) = \mathcal{V}' \cup \{v_{y,1}, v_{y,2}\}$ as the set of feature vectors used in data $X$.*

3) *For each $v \in \mathcal{V}(X)$, pick $C_p$ disjoint patches in $[P]$ and denote it as $\mathcal{P}_v(X)$ (the distribution of these patches can be arbitrary). We denote $\mathcal{P}(X) = \cup_{v \in \mathcal{V}(X)} \mathcal{P}_v(X)$.*

4) *If $\mathcal{D} = \mathcal{D}_s$ is the single-view distribution, pick a value $\hat{l} = \hat{l}(X) \in [2]$ uniformly at random.*

5) *For each $p \in \mathcal{P}_v(X)$ for some $v \in \mathcal{V}(X)$, given feature noise $\alpha_{p,v'} \in [0, \gamma]$, we set*
$$x_p = z_p v + \sum\nolimits_{v' \in \mathcal{V}} \alpha_{p,v'} v' + \xi_p,$$
*where $\xi_p \in \mathcal{N}(0, \sigma_p^2 \mathbf{I})$ is an independent random Gaussian noise. The coefficients $z_p \geq 0$ satisfy*

   - *For "multi-view" data $(X, y) \in \mathcal{D}_m$, when $v \in \{v_{y,1}, v_{y,2}\}$, $\sum_{p \in \mathcal{P}_v(X)} z_p \in [1, O(1)]$ and $\sum_{p \in \mathcal{P}_v(X)} z_p^q \in [1, O(1)]$ for an integer $q \geq 3$, and the marginal distribution of $\sum_{p \in \mathcal{P}_v(X)} z_p$ is left-close. When $v \in \mathcal{V}(X) \setminus \{v_{y,1}, v_{y,2}\}$, $\sum_{p \in \mathcal{P}_v(X)} z_p \in [\Omega(1), 0.4]$, and the marginal distribution of $\sum_{p \in \mathcal{P}_v(X)} z_p$ is right-close.*

   - *For "single-view" data $(X, y) \in \mathcal{D}_s$, when $v = v_{y,\hat{l}}$, $\sum_{p \in \mathcal{P}_v(X)} z_p \in [1, O(1)]$ for the integer $q \geq 3$. When $v = v_{y,3-\hat{l}}$, $\sum_{p \in \mathcal{P}_v(X)} z_p \in [\rho, O(\rho)]$ (we set $\rho = k^{-0.01}$ for simplicity). When $v \in \mathcal{V}(X) \setminus \{v_{y,1}, v_{y,2}\}$, $\sum_{p \in \mathcal{P}_v(X)} z_p \in [\Omega(\Gamma), \Gamma]$, where $\Gamma = 1/\text{polylog}(k)$, and the marginal distribution of $\sum_{p \in \mathcal{P}_v(X)} z_p$ is right-close.*

6) *For each $p \in [P] \setminus \mathcal{P}(X)$, with an independent random Gaussian noise $\xi_p \sim \mathcal{N}(0, \frac{\gamma^2 k^2}{d} \mathbf{I})$,*
$$x_p = \sum\nolimits_{v' \in \mathcal{V}} \alpha_{p,v'} v' + \xi_p,$$
*where each $\alpha_{p,v'} \in [0, \gamma]$ is the feature noise.*

Based on the definition of data distribution $\mathcal{D}$, we define the training dataset $\mathcal{Z}$ as follows.

**Definition 8.** *Assume the distribution $\mathcal{D}$ consists of samples from $\mathcal{D}_m$ w.p. $1 - \mu$ and from $\mathcal{D}_s$ w.p. $\mu$. We are given $N_l$ labeled training samples and $N_u$ unlabeled training samples from $\mathcal{D}$, where typically $N_u \gg N_l$. The training dataset is denoted as $\mathcal{Z} = \mathcal{Z}_{l,m} \cup \mathcal{Z}_{l,s} \cup \mathcal{Z}_{u,m} \cup \mathcal{Z}_{u,s}$, where $\mathcal{Z}_{l,m}$ and $\mathcal{Z}_{l,s}$ represent the multi-view and single-view labeled data, respectively, and $\mathcal{Z}_{u,m}$ and $\mathcal{Z}_{u,s}$ represent the multi-view and single-view unlabeled data, respectively. We denote $(X, y) \sim \mathcal{Z}$ as a pair $(X, y)$ sampled uniformly at random from the empirical training dataset $\mathcal{Z}$.*

Then, we introduce the smoothed ReLU function $\overline{\text{ReLU}}$ (Allen-Zhu & Li, 2023) in detail: for an integer $q \geq 3$ and a threshold $\varrho = \frac{1}{\text{polylog}(k)}, \overline{\text{ReLU}}(z) = 0$ if $z \leq 0, \overline{\text{ReLU}}(z) = \frac{z^q}{(q\varrho^{q-1})}$ if $z \in [0, \varrho]$ and $\overline{\text{ReLU}}(z) = z - (1 - \frac{1}{q})\varrho$ if $z \geq \varrho$. This configuration ensures a linear relationship for large $z$ values while significantly reducing the impact of low-magnitude noises for small $z$ values, thereby enhancing the separation of true features from noises.

We also introduce our assumption on FixMatch's strong augmentation $\mathcal{A}(\cdot)$, which is composed by CutOut (DeVries & Taylor, 2017) and RandAugment (Cubuk et al., 2020). As discussed in Sec. 3.2 and Appendix K.1, we focus on its probabilistic feature removal effect.

**Assumption 9.** *Suppose that for a given image, strong augmentation $\mathcal{A}(\cdot)$ randomly removes its semantic patches and noisy patches with probabilities $\pi_2$ and $1 - \pi_2$, respectively. For a single-view image, the sole semantic feature is removed with probability $\pi_2$. For a multi-view image, either of the two features, $v_{i,1}$ or $v_{i,2}$, is removed with probabilities $\pi_1\pi_2$ and $(1 - \pi_1)\pi_2$, respectively. We define the strong augmentation $\mathcal{A}(\cdot)$ for multi-view data as follows: for $p \in [P]$,*

$$\mathcal{A}(x_p) = \begin{cases} \max(\epsilon_1, \epsilon_2)x_p, & \text{if } v_{y,1} \text{ is in the patch } x_p, \\ \max(1 - \epsilon_1, \epsilon_2)x_p, & \text{if } v_{y,2} \text{ is in the patch } x_p, \\ (1 - \epsilon_2)x_p, & \text{otherwise (noisy patch)}, \end{cases} \quad (10)$$

*where $\epsilon_1$ and $\epsilon_2$ are independent Bernoulli random variables, each equal to 0 with probabilities $\pi_1$ and $\pi_2$, respectively.*

Here we use the "max" function to ensure $\epsilon_1$ is active when $\epsilon_2 = 0$, which implies that $\mathcal{A}(\cdot)$ selects one feature to remove at a time. The reason behind this assumption is that as we can observe from Figure 1 and 2, different semantic features in a multi-view image are spatially distinct. Consequently, the likelihood of a square patch from random CutOut and transformations from RandAugment to remove both features is substantially lower than removing just one. To simplify our theoretical analysis, we therefore assume that $\mathcal{A}(\cdot)$ targets a single feature for removal in each instance.

Then we introduce the parameter assumption necessary to the proof. As we follow the proof framework of Allen-Zhu & Li (2023), the assumptions on most of the parameters are similar.

**Parameter Assumption 10.** *We assume that*

- *$q \geq 3$ and $\sigma_0^{q-2} = 1/k$, where $\sigma_0$ gives the initialization magnitude.*

- *$\gamma \leq \tilde{O}(\frac{\sigma_0}{k})$ and $\gamma^q \leq \tilde{\Theta}(\frac{1}{k^{q-1}mP})$, where $\gamma$ controls the feature noise.*

- *The size of single-view labeled training data $N_{l,s} = \tilde{o}(k/\rho)$ and $N_{l,s} \leq \frac{k^2}{s}\rho^{q-1}$.*

- *$N_l \geq N_{l,s} \cdot \text{poly}(k), \eta T_1 \geq N_l \cdot \text{poly}(k)$, and $\sqrt{d} \geq \eta T_1 \cdot \text{poly}(k)$.*

- *The weight for unsupervised loss $\lambda = 1$ and the confidence threshold $\tau = 1 - \tilde{O}(\frac{1}{s^2})$.*

- *The number of unlabeled data for FixMatch $N_u \geq \eta T_2 \cdot \text{poly}(k)$ with $\eta T_2 \geq \text{poly}(k)$, and the ratio of single-view unlabeled data $\frac{N_{u,s}}{N_u} \leq \frac{k^2}{\eta s T_2}$.*

Here the first four parameter assumptions are followed from Allen-Zhu & Li (2023) for supervised learning Phase I, and the last two parameter assumptions are specific to the unsupervised loss Eq. (5) in learning Phase II. Define $\Phi_{i,l}^{(t)} := \sum_{r \in [m]} [\langle w_{i,r}^{(t)}, v_{i,l} \rangle]^+$ and $\Phi_i^{(t)} := \sum_{l \in [2]} \Phi_{i,l}^{(t)}$. We have the following theorem for vanilla FixMatch under CutOut:

**Theorem 11** (Peformance on FixMatch). *For sufficiently large $k > 0$, for every $m = \text{polylog}(k), \eta \leq \frac{1}{\text{poly}(k)}$, setting $T = T_1 + T_2$ with $T_1 = \frac{\text{poly}(k)}{\eta}$ and $T_2 = \frac{\text{poly}(k)}{\eta}$, when Parameter Assumption 10 is satisfied, with probability at least $1 - e^{-\Omega(\log^2 k)}$,*

- *(training accuracy is perfect) for every $(X, y) \in \mathcal{Z}$:*

$$\forall i \neq y : F_y^{(T)}(X) \geq F_i^{(T)}(X) + \Omega(\log k).$$

- *(multi-view testing is good) for every $i, j \in [k]$, we have $\tilde{O}(1) \geq \Phi_i^{(T)} \geq 0.4\Phi_j^{(T)} + \Omega(\log k)$, and thus*

$$\Pr_{(X,y) \in \mathcal{D}_m} \left[ F_y^{(T)}(X) \geq \max_{j \neq y} F_j^{(T)}(X) + \Omega(\log k) \right] \geq 1 - e^{-\Omega(\log^2 k)}.$$

- *(single-view testing is good) for every $i \in [k]$ and $l \in [2]$, we have $\Phi_{i,l}^{(T)} \geq \Omega(\log k)$, and thus*

$$\Pr_{(X,y) \in \mathcal{D}_s} \left[ F_y^{(T)}(X) \geq \max_{j \neq y} F_j^{(T)}(X) + \Omega(\log k) \right] \geq 1 - e^{-\Omega(\log^2 k)}.$$

For Semantic-Aware FixMatch (SA-FixMatch), we denote the number of unlabeled data in this case as $N_c$. Then we have the following assumption on $N_c$.

**Parameter Assumption 12.** $N_c = \max\{\pi_1\pi_2, (1 - \pi_1)\pi_2\}N_u.$

Here $\pi_1 \in (0, 1)$ and $\pi_2 \in (0, 1)$ are the probabilities defined in Assumption 9, where $\pi_2$ is typically small $(1/\text{poly}(k))$, as explained in Appendix H. From Parameter Assumption 12, we observe that the requirement for the number of unlabeled samples in SA-FixMatch is significantly smaller compared to that in FixMatch.

Under Parameter Assumptions 10 and 12, SA-FixMatch achieves the same performance results as Theorem 11, but with a reduced requirement for the number of unlabeled data, decreasing from $N_u$ to $N_c$. Thus, we state the following theorem regarding the performance of SA-FixMatch.

**Theorem 13** (Performance on SA-FixMatch). *Under Parameter Assumption 10 and 12, SA-FixMatch can achieve the same training and test performance as FixMatch in Theorem 11.*

Our main proof of Theorem 11 and Theorem 13 for FixMatch and SA-FixMatch includes analyses on a two-phase learning process. In Phase I, the network relies primarily on the supervised loss due to its inability to generate highly confident pseudo-labels and the large confidence threshold $\tau$ in Eq. (5). According to the results in Allen-Zhu & Li (2023), partial features are learned during the supervised learning Phase I. We review the results on supervised training in Appendix B.

Then in Phase II, the network predicts highly confident pseudo-labels for weakly-augmented samples and uses these correct pseudo-labels to supervise the corresponding strongly-augmented samples via consistency regularization. To theoretically analyze the learning process in Phase II, we build on the proof framework of Allen-Zhu & Li (2023) and demonstrate how the network learns the unlearned features while preserving the learned features during Phase II. Specifically, we present the induction hypothesis for Phase II in Appendix C, along with gradient calculations and function approximations for the unsupervised loss Eq. (5) in Appendix D. We then provide a detailed proof of SA-FixMatch in Appendix E and extend the results to FixMatch in Appendix F. Finally, we generalize our proof to other FixMatch-like SSL methods in Appendix G.

## B  RESULTS ON SUPERVISED LEARNING

In this section, we first recall the results in SL that were derived in Allen-Zhu & Li (2023). Before showing their main results, we first introduce some necessary notations. For every $i \in [k]$, define $\Phi_{i,l}^{(t)} := \sum_{r \in [m]} [\langle w_{i,r}^{(t)}, v_{i,l} \rangle]^+$ and $\Phi_i^{(t)} := \sum_{l \in [2]} \Phi_{i,l}^{(t)}$. Define

$$\Lambda_i^{(t)} := \max_{r \in [m], l \in [2]} [\langle w_{i,r}^{(t)}, v_{i,l} \rangle]^+ \quad \text{and} \quad \Lambda_{i,l}^{(t)} := \max_{r \in [m]} [\langle w_{i,r}^{(t)}, v_{i,l} \rangle]^+,$$

where $\Lambda_{i,l}$ indicates the largest correlation between the feature vector $v_{i,l}$ and all neurons $w_{i,r}$ ($r \in [m]$) from class $i$. Then we define the "view lottery winning" set:

$$\mathcal{M} := \left\{ (i, l^*) \in [k] \times [2] \,\middle|\, \Lambda_{i,l^*}^{(0)} \geq \Lambda_{i,3-l^*}^{(0)} \left( 1 + \frac{2}{\log^2 m} \right) \right\}.$$

The intuition behind $\mathcal{M}$ is that, subject to model initialization, if $(i, l) \in \mathcal{M}$, then the feature $v_{i,l}$ will be learned by the model during supervised learning process and the feature $v_{i,3-l}$ will be missed. The set $\mathcal{M}$ satisfies the following property (refer to the Proposition C.2. of Allen-Zhu & Li (2023)):

**Proposition 14.** *Suppose $m \leq \text{poly}(k)$. For every $i \in [k]$, $\Pr[(i,1) \in \mathcal{M} \text{ or } (i,2) \in \mathcal{M}] \geq 1 - o(1)$.*

Based on Theorem 1 of Allen-Zhu & Li (2023), after training for $T$ iterations with the supervised training loss $L_s^{(t)} = \mathbb{E}_{(X,y) \sim \mathcal{Z}_l} \left[ -\log \mathbf{logit}_y(F^{(t)}, X) \right]$, the training accuracy on labeled samples is perfect and $L_s^{(T)}$ approaches zero, i.e., for every $(X, y) \in \mathcal{Z}_l$,

$$\forall i \neq y : F_y^{(T)}(X) \geq F_i^{(T)}(X) + \Omega(\log k),$$

and we have $L_s^{(T)} \leq \frac{1}{\text{poly}(k)}$. Besides, it satisfies that $0.4\Phi_i^{(T)} - \Phi_j^{(T)} \leq -\Omega(\log k)$ for every pair $i, j \in [k]$. This means that at least one of $\Phi_{i,1}^{(T)}$ or $\Phi_{i,2}^{(T)}$ for all $i \in [k]$ increase to a large scale of $\Theta(\log(k))$, which means at least one of $v_{i,1}$ and $v_{i,2}$ for all $i \in [k]$ is learned after supervised training for $T$ iterations. Thus, all multi-view training data are classified correctly. For single-view training data without the learned features, they are classified correctly by memorizing the noises in the data during the supervised training process. Then for the test accuracy, for the multi-view data point $(X, y) \sim \mathcal{D}_m$, with the probability at least $1 - e^{-\Omega(\log^2 k)}$, it has

$$\mathbf{logit}_y(F^{(T)}, X) \geq 1 - \tilde{O}\left(\frac{1}{s^2}\right),$$

and

$$\Pr_{(X,y) \sim \mathcal{D}_m} \left[ F_y^{(T)}(X) \geq \max_{j \neq y} F_j^{(T)}(X) + \Omega(\log k) \right] \geq 1 - e^{-\Omega(\log^2 k)}.$$

This means that the test accuracy of multi-view data is good. However, for the single-view data $(X, y) \sim \mathcal{D}_s$, whenever $(i, l^*) \in \mathcal{M}$, we have $\Phi_{i,3-l^*}^{(T)} \ll \frac{1}{\text{polylog}(k)}$ and

$$\Pr_{(X,y) \sim \mathcal{D}_s} \left[ F_y^{(T)}(X) \geq \max_{j \neq y} F_j^{(T)}(X) - \frac{1}{\text{polylog}(k)} \right] \leq \frac{1}{2}(1 + o(1)),$$

which means that the test accuracy on single-view data is nearly 50%.

The results in Allen-Zhu & Li (2023) fully indicate the feature learning process of SL. The main reason for the imperfect performance of SL is that, due to "lottery winning", it only captures one of the two semantic features for each class during the supervised training process. Therefore, for single-view data without this feature, it has low test accuracy.

In the following, we will consider the effect of loss $L_u^{(t)}$ on unlabeled data for training:

$$L_u^{(t)} = \mathbb{E}_{(X,y) \sim \mathcal{Z}_u} \left[ \mathbb{I}_{\{\max_i \mathbf{logit}_i(F^{(t)}, \alpha(X)) \geq \tau\}} \cdot -\log \mathbf{logit}_b(F^{(t)}, \mathcal{A}(X)) \right].$$

where $b = \arg\max_{i \in [k]} \mathbf{logit}_i(F^{(t)}, \alpha(X))$, $\tau$ is the confidence threshold and $\alpha, \mathcal{A}$ are the weak and strong augmentations, respectively. For the simplicity of proof, we set $\alpha$ to be identity mapping. In the following, we will prove Theorem 11. By setting $\tau = 1 - \tilde{O}(1/s^2)$, we will show that after training the supervised network $F^{(T_1)}$ with the unsupervised loss $L_u^{(t)}$ for an additional $T_2 = \frac{\text{poly}(k)}{\eta}$ epochs, the FixMatch-trained network $F^{(T)}$ learns complete semantic features for all classes, achieving perfect test performance on both multi-view and single-view data.

## C INDUCTION HYPOTHESIS

In this section, to prove our theorem, similar to Allen-Zhu & Li (2023), we present an induction hypothesis for every training iteration $t$ in Phase II. We first show the loss function in Phase II.

**Loss Function.** Recall $\mathbf{logit}_i(F, X) := \frac{e^{F_i(X)}}{\sum_{j \in [k]} e^{F_j(X)}}$. In learning Phase I, before the network learned partial features to make confident prediction, only the supervised loss $L_s^{(t)}$ takes effect

$$L_s^{(t)} = \mathbb{E}_{(X,y) \sim \mathcal{Z}_l} \left[ -\log \mathbf{logit}_y(F^{(t)}, X) \right].$$

In Phase II, after we train the network $F$ for $T_1 = \frac{\text{poly}(k)}{\eta}$ epochs using $L_s^{(t)}$ in the Phase I, according to the results in Appendix B, one of the features in each class is captured. Then we consider to optimize the network $F^{(T_1)}$ using the following combination of losses:

$$L^{(t)} = \mathbb{E}_{(X,y)\sim\mathcal{Z}_l}\left[-\log\mathbf{logit}_y(F^{(t)}, X)\right]$$
$$+ \lambda\mathbb{E}_{X\sim\mathcal{Z}_u}\left[\mathbb{I}_{\{\max_i\mathbf{logit}_i(F^{(t)},\alpha(X))\geq\tau\}}\cdot-\log\mathbf{logit}_b(F^{(t)}, \mathcal{A}(X))\right],$$

where $b = \arg\max_{i\in[k]}\mathbf{logit}_i(F^{(t)}, \alpha(X))$. Recall $\tau = 1-\tilde{O}(1/s^2)$, when $t \geq T_1$ and we use $F^{(t)}$ to classify the unlabeled data $X \sim \mathcal{Z}_u$, we will get a correct pseudo-label with high probability, i.e., $b = y$, where $y$ denotes the ground truth label of $X$. This means that for $X \sim \mathcal{Z}_{u,m}$, with probability at least $1 - e^{-\Omega(\log^2 k)}$, $\mathbf{logit}_y(F^{(t)}, X) \geq \tau$ and for $X \sim \mathcal{Z}_{u,s}$, when $(y,l^*) \in \mathcal{M}$ and $\hat{l}(X) = l^*$, with the probability at least $1 - e^{-\Omega(\log^2 k)}$, $\mathbf{logit}_y(F^{(t)}, X) \geq \tau$. We denote the samples in $\mathcal{Z}_u$ that satisfy $\mathbf{logit}_y(F^{(t)}, X) \geq \tau$ as $\tilde{\mathcal{Z}}_u$ and let $\tilde{N}_u = |\tilde{\mathcal{Z}}_u|$. In this way, we can further simplify the loss as

$$L^{(t)} = L_s^{(t)} + \lambda L_u^{(t)}$$
$$= \mathbb{E}_{(X,y)\sim\mathcal{Z}_l}\left[-\log\mathbf{logit}_y(F^{(t)}, X)\right] + \lambda\mathbb{E}_{X\sim\tilde{\mathcal{Z}}_u}\left[-\log\mathbf{logit}_b(F^{(t)}, \mathcal{A}(X))\right]. \tag{11}$$

We introduce the following induction hypothesis:

**Induction Hypothesis 15.** *During Phase II ($t \geq T_1$), for every $l \in [2]$, for every $r \in [m]$, for every $X \in \tilde{\mathcal{Z}}_u$ and $i \in [k]$,*

    *(a) For every $p \in \mathcal{P}_{v_{i,l}}(X)$, we have: $\langle w_{i,r}^{(t)}, x_p\rangle = \langle w_{i,r}^{(t)}, v_{i,l}\rangle z_p \pm \tilde{o}(\sigma_0)$.*

    *(b) For every $p \in \mathcal{P}(X)\setminus(\mathcal{P}_{v_{i,1}}(X)\cup\mathcal{P}_{v_{i,2}}(X))$, we have: $|\langle w_{i,r}^{(t)}, x_p\rangle| \leq \tilde{O}(\sigma_0)$.*

    *(c) For every $p \in [P]\setminus\mathcal{P}(X)$, we have $|\langle w_{i,r}^{(t)}, x_p\rangle| \leq \tilde{O}(\sigma_0\gamma k)$.*

*Moreover, we have for every $i \in [k]$, every $l \in [2]$,*

    *(d) $\Phi_{i,l}^{(t)} \geq \tilde{\Omega}(\sigma_0)$ and $\Phi_{i,l}^{(t)} \leq \tilde{O}(1)$.*

    *(e) for every $r \in [m]$, it holds that $\langle w_{i,r}^{(t)}, v_{i,l}\rangle \geq -\tilde{O}(\sigma_0)$.*

*Recall that $\Phi_{i,l}^{(t)} := \sum_{r\in[m]}[\langle w_{i,r}^{(t)}, v_{i,l}\rangle]^+$ and $\Phi_i^{(t)} := \sum_{l\in[2]}\Phi_{i,l}^{(t)}$.*

The intuition behind Induction Hypothesis 15 is that training with semi-supervised loss Eq. (11) filters out feature noises and background noises for both multi-view data and single-view data. This can be seen in comparison with Induction Hypothesis C.3 of Allen-Zhu & Li (2023). With the help of Induction Hypothesis 15, we can prove that the correlations between $w_{i,r}$ and learned features in Phase I are retained in Phase II, and the correlations between $w_{i,r}$ and unlearned features will increase to a large scale ($\log(k)$) in the end of learning Phase II.

## D    GRADIENT CALCULATIONS AND FUNCTION APPROXIMATION

**Gradient Calculation.** We present the gradient calculations for the cross-entropy loss $L_u(F; X, y) = -\log\mathbf{logit}_y(F, \mathcal{A}(X))$ on unlabeled data $X$ with correctly predicted pseudo-label $y$. With a slight abuse of notation, we use $(X, y) \sim \tilde{\mathcal{Z}}_u$ to denote unlabeled data $X \sim \tilde{\mathcal{Z}}_u$ along with its corresponding ground truth label $y$.

**Fact 16.** *Given data point $(X, y) \sim \tilde{\mathcal{Z}}_u$, for every $i \in [k], r \in [m]$,*

$$-\nabla_{w_{i,r}} L_u(F; X, y) = (1 - \mathbf{logit}_i(F, \mathcal{A}(X))) \sum_{p \in [P]} \overline{\mathrm{ReLU}}'(\langle w_{i,r}, \mathcal{A}(x_p)\rangle)\mathcal{A}(x_p), \quad when\ i = y,$$

(12)

$$-\nabla_{w_{i,r}} L_u(F; X, y) = -\mathbf{logit}_i(F, \mathcal{A}(X)) \sum_{p \in [P]} \overline{\mathrm{ReLU}}'(\langle w_{i,r}, \mathcal{A}(x_p)\rangle)\mathcal{A}(x_p), \quad when\ i \neq y.$$

(13)

**Definition 17.** *For each data point $X$, we define a value $V_{i,r,l}(X)$ as*

$$V_{i,r,l}(X) := \mathbb{I}_{v_{i,l} \in \mathcal{V}(X)} \sum_{p \in \mathcal{P}_{v_{i,l}}(X)} \overline{\mathrm{ReLU}}'(\langle w_{i,r}, \mathcal{A}(x_p)\rangle)\mathcal{A}(z_p).$$

**Definition 18.** *We also define small error terms which will be frequently used:*

$$\mathcal{E}_1 := \tilde{O}(\sigma_0^{q-1})\gamma s \qquad \mathcal{E}_{2,i,r}(X) := O(\gamma(V_{i,r,1}(X) + V_{i,r,2}(X)))$$
$$\mathcal{E}_3 := \tilde{O}(\sigma_0 \gamma k)^{q-1} \gamma P \qquad \mathcal{E}_{4,j,l}(X) := \tilde{O}(\sigma_0^{q-1})\mathbb{I}_{v_{j,l} \in \mathcal{V}(X)}.$$

Then we have the following bounds for positive gradients, i.e., when $i = y$:

**Claim 19** (positive gradients). *Suppose Induction Hypothesis 15 holds at iteration $t$. For every $(X, y) \in \tilde{\mathcal{Z}}_u$, every $r \in [m]$, every $l \in [2]$, and $i = y$, we have*

*(a) $\langle -\nabla_{w_{i,r}} L_u(F^{(t)}; X, y), v_{i,l} \rangle \geq \left( V_{i,r,l}(X) - \tilde{O}(\sigma_p P) \right) \left( 1 - \mathbf{logit}_i(F^{(t)}, \mathcal{A}(X)) \right).$*

*(b) $\langle -\nabla_{w_{i,r}} L_u(F^{(t)}; X, y), v_{i,l} \rangle \leq (V_{i,r,l}(X) + \mathcal{E}_1 + \mathcal{E}_3) \left( 1 - \mathbf{logit}_i(F^{(t)}, \mathcal{A}(X)) \right).$*

*(c) For every $j \in [k] \setminus \{i\}$,*
$$|\langle -\nabla_{w_{i,r}} L_u(F^{(t)}; X, y), v_{j,l} \rangle| \leq (\mathcal{E}_1 + \mathcal{E}_{2,i,r}(X) + \mathcal{E}_3 + \mathcal{E}_{4,j,l}(X)) \left( 1 - \mathbf{logit}_i(F^{(t)}, \mathcal{A}(X)) \right).$$

We also have the following claim about the negative gradients (i.e., $i \neq y$). The proof of positive and negative gradients is identical to the proof in Allen-Zhu & Li (2023), except that in our case, we have the augmentation operations on the data patches.

**Claim 20** (negative gradients). *Suppose Induction Hypothesis 15 holds at iteration $t$. For every $(X, y) \sim \tilde{\mathcal{Z}}_u$, every $r \in [m]$, every $l \in [2]$, and $i \in [k] \setminus \{y\}$, we have*

*(a) $\langle -\nabla_{w_{i,r}} L_u(F^{(t)}; X, y), v_{i,l} \rangle \geq -\mathbf{logit}_i(F^{(t)}, \mathcal{A}(X)) (\mathcal{E}_1 + \mathcal{E}_3 + V_{i,r,l}(X)).$*

*(b) For every $j \in [k]$: $\langle -\nabla_{w_{i,r}} L_u(F^{(t)}; X, y), v_{j,l} \rangle \leq \mathbf{logit}_i(F^{(t)}, \mathcal{A}(X))\tilde{O}(\sigma_p P).$*

*(c) For every $j \in [k] \setminus \{i\}$: $\langle -\nabla_{w_{i,r}} L_u(F^{(t)}; X, y), v_{j,l} \rangle \geq -\mathbf{logit}_i(F^{(t)}, \mathcal{A}(X)) (\mathcal{E}_1 + \mathcal{E}_3 + \mathcal{E}_{4,j,l}(X)).$*

**Function Approximation.** Let us denote $Z_{i,l}^{(t)}(X) := \mathbb{I}_{v_{i,l} \in \mathcal{V}(X)} \left( \sum_{p \in \mathcal{P}_{v_{i,l}}(X)} \mathcal{A}(z_p) \right)$, we can easily derive the following result on function approximation.

**Claim 21** (function approximation). *Suppose Induction Hypothesis 15 holds at iteration $t$ and suppose $s \leq \tilde{O}(\frac{1}{\sigma_0^q m})$ and $\gamma \leq \tilde{O}(\frac{1}{\sigma_0 k(mP)^{1/q}})$, we have:*

- *for every $t$, every $(X, y) \in \tilde{\mathcal{Z}}_u$ and $i \in [k]$, we have*

$$F_i^{(t)}(X) = \sum_{l \in [2]} \left( \Phi_{i,l}^{(t)} \times Z_{i,l}^{(t)}(X) \right) \pm O(\frac{1}{\mathrm{polylog}(k)}).$$

- *for every $(X, y) \sim \mathcal{D}$, with probability at least $1 - e^{-\Omega(\log^2 k)}$, it satisfies for every $i \in [k]$,*

$$F_i^{(t)}(X) = \sum_{l \in [2]} \left( \Phi_{i,l}^{(t)} \times Z_{i,l}^{(t)}(X) \right) \pm O(\frac{1}{\mathrm{polylog}(k)}).$$

**Claim 22** (classification test performance). *Suppose Parameter Assumption 10 holds. Assume for* $\forall i \in [k], \exists l \in [2]$ *such that* $\Phi_{i,l} \geq \Omega(\log k)$ *and* $\Phi_{i,3-l} \leq \frac{1}{\text{polylog}(k)}$ *in the trained network $F$. Then the following statements hold with probability at least $1 - e^{-\Omega(\log^2 k)}$:*

- *For any $(X, y) \sim \mathcal{D}$ which contains $v_{y,l}$ as the main semantic feature, network $F$ can correctly predict the label $y$ of $X$.*

- *For any $(X, y) \sim \mathcal{D}$ only with $v_{y,3-l}$ as the main semantic feature, the network $F$ would mistakenly predict the label of $X$.*

*Proof.* Based on our definition of multi-view and single-view data, for any multi-view data $(X, y) \sim \mathcal{D}_m$, when we have $\Phi_i \geq \Omega(\log k)$ ($\forall i \in [k]$), according to Claim 21 and Claim D.16 in Allen-Zhu & Li (2023), we have $0.4\Phi_i - \Phi_j \leq -\Omega(\log k)$, which means $F_y(X) \geq \max_{j \neq y} F_j(X) + \Omega(\log k)$. For any single-view data $(X, y) \sim \mathcal{D}_s$ with $v_{y,l}$ as the main semantic feature, according to Claim 21, $F_y(X) \geq \Omega(\log k)$ and for $i \neq y$, $F_i(X) \leq O(\Gamma)$. Thus, we have $F_y(X) \geq \max_{j \neq y} F_j(X) + \Omega(\log k)$. In the above two cases, the network $F$ can correctly predict the label $y$ of $X$.

For any single-view data $(X, y) \sim \mathcal{D}_s$ with $v_{y,3-l}$ as the main semantic feature, according to Claim 21, $F_y(X) \leq \tilde{O}(\rho) + \frac{1}{\text{polylog}(k)}$ and with probability at least $1 - e^{-\Omega(\log^2 k)}$ there exists $i \in [k]$ and $i \neq y$ such that $F_i(X) \geq \tilde{\Omega}(\Gamma)$. This means that $F_y(X) \leq \max_{i \neq y} F_i(X) - \frac{1}{\text{polylog}(k)}$. In this case, the network $F$ will mistakenly predict the label of $X$. $\square$

# E   PROOF FOR SEMANTIC-AWARE FIXMATCH

Here we consider to prove the SA-FixMatch case first. SA-FixMatch replaces CutOut operation in strong augmentation of FixMatch with SA-CutOut, which deterministically removes the learned features in Phase I. This helps to reduce the number of unlabeled samples needed during the Phase II training as shown in Assumption 12. Since the learned features of Phase I are deterministically removed in SA-FixMatch, for the simplicity of theoretical analysis, we first prove the results under SA-FixMatch and then we can easily generalize the results to FixMatch.

For theoretical proof, we assume that Grad-CAM in SA-CutOut can correctly identify the learned feature after the first stage. In this case, the formulation of strong augmentation $\mathcal{A}(\cdot)$ with SA-CutOut for $X \sim \tilde{\mathcal{Z}}_u$ and $(i, l^*) \in \mathcal{M} \cap \mathcal{V}(X)$ ($l^*$ varies depending on $i$) is

$$\mathcal{A}(x_p) = \begin{cases} 0, & \text{if } p \in \mathcal{P}_{v_{i,l^*}}(X), \\ x_p, & \text{otherwise.} \end{cases} \tag{14}$$

In the following, we will begin to prove Theorem 13. We first introduce some useful claims as consequences of the Induction Hypothesis 15.

## E.1   USEFUL CLAIMS

Based on the results from Allen-Zhu & Li (2023), at the end of learning Phase I, for $(i, l^*) \in \mathcal{M}$, we have $\Phi_{i,l^*}^{(T_1)} \geq \Omega(\log k)$ while $\Phi_{i,3-l^*}^{(T_1)} \leq 1/\text{polylog}(k)$. Below the first claim addresses the initial growth of the correlations between $w_{i,r}$ and the unlearned feature ($\Phi_{i,3-l^*}^{(t)}$) in learning Phase II, and the second claim asserts that the correlations between $w_{i,r}$ and the learned feature ($\Phi_{i,l^*}^{(t)}$) are preserved during learning Phase II. Here we first give a naive bound on the **logit** function based on function approximation result Claim 21.

**Claim 23** (approximation of logits). *Suppose Induction Hypothesis 15 holds at iteration $t$, and suppose $s \leq \tilde{O}(\frac{1}{\sigma_0^q m})$ and $\gamma \leq \tilde{O}(\frac{1}{\sigma_0 k(mP)^{1/q}})$, then*

- *for every $(X, y) \sim \tilde{\mathcal{Z}}_{u,m}$ and $(i, l^*) \in \mathcal{M}$: $\mathbf{logit}_i(F^{(t)}, \mathcal{A}(X)) = O\left(\frac{e^{O(\Phi_{i,3-l^*})}}{e^{O(\Phi_{i,3-l^*})} + k}\right)$.*

- *for every $(X, y) \sim \tilde{\mathcal{Z}}_{u,s}$ and every $i \in [k]$: $\mathbf{logit}_i(F^{(t)}, \mathcal{A}(X)) = O\left(\frac{1}{k}\right)$.*

*Proof.* Recall $F_i^{(t)}(\mathcal{A}(X)) = \sum_{r\in[m]}\sum_{p\in[P]} \overline{\text{ReLU}}(\langle w_{i,r}, \mathcal{A}(x_p)\rangle)$. According to Claim 21, data distribution Def. 7 and data augmentation defined in Eq. (14), we have that for $(X,y) \sim \tilde{\mathcal{Z}}_{u,m}$ and $(i, l^*) \in \mathcal{M}$ ($l^*$ varies depending on $i$),

$$0 \leq F_y^{(t)}(\mathcal{A}(X)) \leq \Phi_{y,3-l^*}^{(t)} \cdot O(1) + O(\frac{1}{\text{polylog}(k)}),$$

and for $i \in [k] \setminus \{y\}$,

$$0 \leq F_i^{(t)}(\mathcal{A}(X)) \leq \Phi_{i,3-l^*}^{(t)} \cdot 0.4 + O(\frac{1}{\text{polylog}(k)}).$$

Thus, combining the above results, for every $(X,y) \sim \tilde{\mathcal{Z}}_{u,m}$ and $(i, l^*) \in \mathcal{M}$, we have for every $i \in [k]$,

$$\textbf{logit}_i(F^{(t)}, \mathcal{A}(X)) = O\left(\frac{e^{O(\Phi_{i,3-l^*}^{(t)})}}{e^{O(\Phi_{i,3-l^*}^{(t)})} + k}\right).$$

On the other hand, for the single-view data in $\tilde{\mathcal{Z}}_{u,s}$, as the only class-specific semantic feature is removed after we conduct strong augmentation, only noisy unlearned features and background noises remain. Thus, for $(X,y) \sim \tilde{\mathcal{Z}}_{u,s}$ and $(i, l^*) \in \mathcal{M}$, we have for $i \neq y$,

$$0 \leq F_y^{(t)}(\mathcal{A}(X)) \leq O(1) \quad \text{and} \quad 0 \leq F_i^{(t)}(\mathcal{A}(X)) \leq \Phi_{i,3-l^*}^{(t)} \cdot O(\Gamma) + O(\frac{1}{\text{polylog}(k)}) \leq O(1),$$

and thus we have for every $i \in [k]$,

$$\textbf{logit}_i(F^{(t)}, \mathcal{A}(X)) = O\left(\frac{1}{k}\right).$$

$\square$

Now we are ready to prove the following claim on the initial growth of $\Phi_{i,3-l^*}^{(t)}$ with $(i, l^*) \in \mathcal{M}$.

**Claim 24** (initial growth). *Suppose Induction Hypothesis 15 holds at iteration $t$, then for every $i \in [k]$ with $(i, l^*) \in \mathcal{M}$, suppose $\Phi_{i,3-l^*}^{(t)} \leq O(1)$, then it satisfies*

$$\Phi_{i,3-l^*}^{(t+1)} = \Phi_{i,3-l^*}^{(t)} + \tilde{\Theta}\left(\frac{\eta}{k}\right) \overline{\text{ReLU}}'(\Phi_{i,3-l^*}^{(t)}).$$

*Proof.* For any $w_{i,r}$ and $v_{i,3-l^*}$ ($i \in [k], r \in [m]$), we have

$$\langle w_{i,r}^{(t+1)}, v_{i,3-l^*}\rangle = \langle w_{i,r}^{(t)}, v_{i,3-l^*}\rangle - \eta\mathbb{E}_{(X,y)\sim\mathcal{Z}_l}[\langle\nabla_{w_{i,r}}L_s(F^{(t)}; X, y), v_{i,3-l^*}\rangle]$$
$$- \eta\mathbb{E}_{(X,y)\sim\tilde{\mathcal{Z}}_u}[\langle\nabla_{w_{i,r}}L_u(F^{(t)}; X, y), v_{i,3-l^*}\rangle].$$

For the loss term on $\mathcal{Z}_l$, we have

$$- \mathbb{E}_{(X,y)\sim\mathcal{Z}_l}[\langle\nabla_{w_{i,r}}L_s(F^{(t)}; X, y), v_{i,3-l^*}\rangle]$$
$$= \mathbb{E}_{(X,y)\sim\mathcal{Z}_l}\Big[\mathbb{I}_{i=y}(1 - \textbf{logit}_i(F, X))\sum_{p\in[P]}\overline{\text{ReLU}}'(\langle w_{i,r}, x_p\rangle)\langle x_p, v_{i,3-l^*}\rangle -$$
$$\mathbb{I}_{i\neq y}\textbf{logit}_i(F, X)\sum_{p\in[P]}\overline{\text{ReLU}}'(\langle w_{i,r}, x_p\rangle)\langle x_p, v_{i,3-l^*}\rangle\Big].$$

Based on the results from Allen-Zhu & Li (2023), when $t \geq T_1$, we have

$$\mathbb{E}_{(X,y)\sim\mathcal{Z}_l}[(1 - \textbf{logit}_y(F, X))] \leq \frac{1}{\text{poly}(k)} \quad \text{and} \quad \mathbb{E}_{(X,y)\sim\mathcal{Z}_l}[\mathbb{I}_{i\neq y}\textbf{logit}_i(F, X)] \leq \frac{1}{\text{poly}(k)},$$

which means

$$-\mathbb{E}_{(X,y)\sim\mathcal{Z}_l}[\langle\nabla_{w_{i,r}}L_s(F^{(t)}; X, y), v_{i,3-l^*}\rangle] \leq \frac{1}{\text{poly}(k)},$$

i.e., the supervised loss is fully minimized in Phase I and contributes little in Phase II.

Then, from Claim 19 and Claim 20, we know

$$\langle w_{i,r}^{(t+1)}, v_{i,3-l^*} \rangle \geq \langle w_{i,r}^{(t)}, v_{i,3-l^*} \rangle + \eta \mathbb{E}_{(X,y)\sim\tilde{\mathcal{Z}}_u}[\mathbb{I}_{y=i}(V_{i,r,3-l^*}(X) - \tilde{O}(\sigma_p P))(1 - \mathbf{logit}_i(F^{(t)}, \mathcal{A}(X)))$$
$$- \mathbb{I}_{y\neq i}(\mathcal{E}_1 + \mathcal{E}_3 + \mathbb{I}_{v_{i,3-l^*}\in\mathcal{P}(X)} V_{i,r,3-l^*}(X))\mathbf{logit}_i(F^{(t)}, \mathcal{A}(X))].$$

Consider $r = \arg\max_{r\in[m]}\{\langle w_{i,r}^{(t)}, v_{i,3-l^*} \rangle\}$, then as $m = \text{polylog}(k)$, we know $\langle w_{i,r}^{(t)}, v_{i,3-l^*} \rangle \geq \tilde{\Omega}(\Phi_{i,3-l^*}^{(t)})$. Recall $V_{i,r,3-l^*}(X) := \mathbb{I}_{v_{i,3-l^*}\in\mathcal{V}(X)} \sum_{p\in\mathcal{P}_{v_{i,3-l^*}}(X)} \overline{\text{ReLU}}'(\langle w_{i,r}^{(t)}, \mathcal{A}(x_p)\rangle)\mathcal{A}(z_p)$, according to Induction Hypothesis 15(a) and definition in Eq. (14), we have

$$V_{i,r,3-l^*}(X) = \mathbb{I}_{v_{i,3-l^*}\in\mathcal{V}(X)} \sum_{p\in\mathcal{P}_{v_{i,3-l^*}}(X)} \overline{\text{ReLU}}'(\langle w_{i,r}^{(t)}, v_{i,3-l^*} \rangle z_p + \tilde{o}(\sigma_0))z_p.$$

- When $i = y$, at least for $(X, y) \sim \tilde{\mathcal{Z}}_{u,m}$, we have $\sum_{p\in\mathcal{P}_{v_{i,3-l^*}}(X)} z_p \geq 1$, and together with $|\mathcal{P}_{v_{i,3-l^*}}| \leq C_p$, we know $V_{i,r,3-l^*} \geq \Omega(1) \cdot \overline{\text{ReLU}}'(\langle w_{i,r}^{(t)}, v_{i,3-l^*} \rangle)$.

- When $i \neq y$ and when $v_{i,3-l^*} \in \mathcal{P}(X)$, we can use $\sum_{p\in\mathcal{P}_{v_{i,3-l^*}}(X)} z_p \leq 0.4$ to derive that $V_{i,r,3-l^*} \leq 0.4 \cdot \overline{\text{ReLU}}'(\langle w_{i,r}^{(t)}, v_{i,3-l^*} \rangle)$.

Moreover, when $\Phi_{i,3-l^*}^{(t)} \leq O(1)$, by Claim 23, we have $\mathbf{logit}_i(F^{(t)}, \mathcal{A}(X)) \leq O(\frac{1}{k})$. Then we can derive that

$$\langle w_{i,r}^{(t+1)}, v_{i,3-l^*} \rangle \geq \langle w_{i,r}^{(t)}, v_{i,l} \rangle + \eta \mathbb{E}_{(X,y)\sim\tilde{\mathcal{Z}}_u}[\mathbb{I}_{y=i} \cdot \Omega(1) - O(1) \cdot \mathbb{I}_{y\neq i}\mathbb{I}_{v_{i,3-l^*}\in\mathcal{P}(X)} \cdot \frac{1}{k}]$$
$$\cdot \overline{\text{ReLU}}'\langle w_{i,r}^{(t)}, v_{i,3-l^*} \rangle - \eta\tilde{O}(\frac{\sigma_p P + \mathcal{E}_1 + \mathcal{E}_3}{k}).$$

Finally, recall that $\Pr(v_{i,3-l^*} \in \mathcal{P}(X)|i \neq y) = \frac{s}{k} \ll o(1)$, we have that

$$\langle w_{i,r}^{(t+1)}, v_{i,3-l^*} \rangle \geq \langle w_{i,r}^{(t)}, v_{i,3-l^*} \rangle + \tilde{\Omega}\left(\frac{\eta}{k}\right) \overline{\text{ReLU}}'(\langle w_{i,r}^{(t)}, v_{i,3-l^*} \rangle).$$

Similarly, using Claim 19 and 20, we can derive:

$$\langle w_{i,r}^{(t+1)}, v_{i,3-l^*} \rangle \leq \langle w_{i,r}^{(t)}, v_{i,3-l^*} \rangle + \eta \mathbb{E}_{(X,y)\sim\tilde{\mathcal{Z}}_u}[\mathbb{I}_{y=i}(V_{i,r,3-l^*}(X) + \mathcal{E}_1 + \mathcal{E}_3)(1 - \mathbf{logit}_i(F^{(t)}, X))$$
$$- \mathbb{I}_{y\neq i}\tilde{O}(\sigma_p P)\mathbf{logit}_i(F^{(t)}, X)].$$

With similar analyses to the upper bound, we can derive the lower bound

$$\langle w_{i,r}^{(t+1)}, v_{i,3-l^*} \rangle \leq \langle w_{i,r}^{(t)}, v_{i,3-l^*} \rangle + \tilde{O}\left(\frac{\eta}{k}\right) \overline{\text{ReLU}}'(\langle w_{i,r}^{(t)}, v_{i,3-l^*} \rangle).$$

$\square$

With initial growth analysis in Claim 24, similar to Claim D.11 in Allen-Zhu & Li (2023), we can obtain the following result:

**Claim 25.** *Define iteration threshold* $T_0 := \tilde{\Theta}\left(\frac{k}{\eta\sigma_0^{q-2}}\right)$, *then for every* $i \in [k], (i, l^*) \in \mathcal{M}$ *and* $t \geq T_1 + T_0$, *it satisfies that* $\Phi_{i,3-l^*}^{(t)} = \Theta(1)$.

As we stated in Claim 23, model prediction for the augmented single-view data in $\tilde{\mathcal{Z}}_{u,s}$ are kept to the scale of $O\left(\frac{1}{k}\right)$, since after strong augmentation there remains only noises. Now we present the convergence of multi-view data in $\tilde{\mathcal{Z}}_{u,m}$ from $T_1 + T_0$ till the end.

**Claim 26** (multi-view error till the end). *Suppose that the Induction Hypothesis 15 holds for every iteration* $T_1 < t \leq T_1 + T_2$, *and suppose* $\frac{\tilde{N}_{c,s}}{\tilde{N}_c} \leq \frac{k^2}{\eta s T_2}$, *then*

$$\sum_{t=T_1+T_0}^{T_1+T_2} \mathbb{E}_{(X,y)\sim\tilde{\mathcal{Z}}_{u,m}}[1 - \mathbf{logit}_y(F^{(t)}, \mathcal{A}(X))] \leq \tilde{O}\left(\frac{k}{\eta}\right).$$

*Proof.* For any $w_{i,r}$ and $v_{i,3-l^*}$ ($i \in [k], r \in [m]$), we have

$$\langle w_{i,r}^{(t+1)}, v_{i,3-l^*} \rangle = \langle w_{i,r}^{(t)}, v_{i,3-l^*} \rangle - \eta \mathbb{E}_{(X,y)\sim\mathcal{Z}_l}[\langle \nabla_{w_{i,r}} L_s(F^{(t)}; X, y), v_{i,3-l^*} \rangle]$$
$$- \eta \mathbb{E}_{(X,y)\sim\tilde{\mathcal{Z}}_u}[\langle \nabla_{w_{i,r}} L_u(F^{(t)}; X, y), v_{i,3-l^*} \rangle].$$

For the loss term on $\mathcal{Z}_l$, as discussed above, we have

$$-\mathbb{E}_{(X,y)\sim\mathcal{Z}_l}[\langle \nabla_{w_{i,r}} L_s(F^{(t)}; X, y), v_{i,3-l^*} \rangle] \leq \frac{1}{\text{poly}(k)}.$$

Again, by Claim 19 and Claim 20, we know

$$\langle w_{i,r}^{(t+1)}, v_{i,3-l^*} \rangle \geq \langle w_{i,r}^{(t)}, v_{i,3-l^*} \rangle + \eta \mathbb{E}_{(X,y)\sim\tilde{\mathcal{Z}}_u}[\mathbb{I}_{y=i}(V_{i,r,3-l^*}(X) - \tilde{O}(\sigma_p P))(1 - \mathbf{logit}_i(F^{(t)}, \mathcal{A}(X)))$$
$$- \mathbb{I}_{y \neq i}(\mathcal{E}_1 + \mathcal{E}_3 + \mathbb{I}_{v_{i,3-l^*}\in\mathcal{P}(X)} V_{i,r,3-l^*}(X))\mathbf{logit}_i(F^{(t)}, \mathcal{A}(X))].$$

Take $r = \arg\max_{r\in[m]}\{\langle w_{i,r}^{(t)}, v_{i,3-l^*} \rangle\}$, then by $m = \text{polylog}(k)$ we know $\langle w_{i,r}^{(t)}, v_{i,3-l^*} \rangle \geq \tilde{\Omega}(\Phi_{i,3-l^*}^{(t)}) = \tilde{\Omega}(1)$ for $t \geq T_1 + T_0$.

Recall $V_{i,r,3-l^*}(X) := \mathbb{I}_{v_{i,3-l^*}\in\mathcal{V}(X)} \sum_{p\in\mathcal{P}_{v_{i,3-l^*}}(X)} \overline{\text{ReLU}}'(\langle w_{i,r}^{(t)}, \mathcal{A}(x_p) \rangle)\mathcal{A}(z_p)$ and our definition of $\mathcal{A}$, using the Induction Hypothesis 15, we have that for $(X, y) \sim \tilde{\mathcal{Z}}_{u,m}$, it satisfies

$$V_{i,r,3-l^*}(X) = \sum_{p\in\mathcal{P}_{v_{i,3-l^*}}(X)} \overline{\text{ReLU}}'(\langle w_{i,r}^{(t)}, v_{i,3-l^*} \rangle z_p \pm \tilde{o}(\sigma_0))z_p.$$

Since we have $\langle w_{i,r}^{(t)}, v_{i,3-l^*} \rangle \geq \tilde{\Omega}(1) \gg \varrho$ and $|\mathcal{P}_{v_{i,3-l^*}}(X)| \leq O(1)$, for most of $p \in \mathcal{P}_{v_{i,3-l^*}}$, we have already in the linear regime of $\overline{\text{ReLU}}$ so

$$0.9 \sum_{p\in\mathcal{P}_{v_{i,3-l^*}}(X)} z_p \leq V_{i,r,3-l^*}(X) \leq \sum_{p\in\mathcal{P}_{v_{i,3-l^*}}(X)} z_p.$$

Thus, when $(X, y) \sim \tilde{\mathcal{Z}}_{u,m}$ and $y = i$, we have $V_{i,r,3-l^*}(X) \geq 0.9$; when $(X, y) \sim \tilde{\mathcal{Z}}_{u,m}$, $y \neq i$ and $v_{i,3-l^*} \in \mathcal{P}(X)$, we have $V_{i,r,3-l^*}(X) \leq 0.4$. When $(X, y) \sim \tilde{\mathcal{Z}}_{u,s}$ and $y = i$, we have $V_{i,r,3-l^*}(X) \geq 0$; when $(X, y) \sim \tilde{\mathcal{Z}}_{u,s}$, $y \neq i$ and $v_{i,3-l^*} \in \mathcal{P}(X)$, we have $V_{i,r,3-l^*}(X) \leq O(\Gamma) \ll o(1)$. Then we can derive that

$$\langle w_{i,r}^{(t+1)}, v_{i,3-l^*} \rangle$$
$$\geq \langle w_{i,r}^{(t)}, v_{i,3-l^*} \rangle + \eta \mathbb{E}_{(X,y)\sim\tilde{\mathcal{Z}}_{u,m}}[0.89 \cdot \mathbb{I}_{y=i}(1 - \mathbf{logit}_i(F^{(t)}, \mathcal{A}(X)))]$$
$$- \eta \mathbb{E}_{(X,y)\sim\tilde{\mathcal{Z}}_{u,m}}[\mathbb{I}_{y\neq i}(\mathcal{E}_1 + \mathcal{E}_3 + 0.4\mathbb{I}_{v_{i,3-l^*}\in\mathcal{P}(X)})\mathbf{logit}_i(F^{(t)}, \mathcal{A}(X))]$$
$$- O\left(\frac{\eta\tilde{N}_{c,s}}{\tilde{N}_c}\right) \mathbb{E}_{(X,y)\sim\tilde{\mathcal{Z}}_{u,s}}[\tilde{O}(\sigma_p P)\mathbb{I}_{y=i}(1 - \mathbf{logit}_i(F^{(t)}, \mathcal{A}(X)))]$$
$$- O\left(\frac{\eta\tilde{N}_{c,s}}{\tilde{N}_c}\right) \mathbb{E}_{(X,y)\sim\tilde{\mathcal{Z}}_{u,s}}[\mathbb{I}_{y\neq i}(\mathcal{E}_1 + \mathcal{E}_3 + \mathbb{I}_{v_{i,3-l^*}\in\mathcal{P}(X)})\mathbf{logit}_i(F^{(t)}, \mathcal{A}(X))]$$
$$\geq \langle w_{i,r}^{(t)}, v_{i,3-l^*} \rangle + \Omega\left(\frac{\eta}{k}\right) \mathbb{E}_{(X,y)\sim\tilde{\mathcal{Z}}_{u,m}}[(1 - \mathbf{logit}_y(F^{(t)}, \mathcal{A}(X)))] - O\left(\frac{\eta s\tilde{N}_{c,s}}{k^2\tilde{N}_c}\right),$$

where the last step is based on Claim 23 and the fact that $\Pr(v_{i,l} \in \mathcal{P}(X)|i \neq y) = \frac{s}{k} \ll o(1)$. Thus, when summing up all $r \in [m]$, and telescoping from $T_1 + T_0$ to $T_1 + T_2$, we have

$$\Phi_{i,3-l^*}^{(T_1+T_2)} \geq \Phi_{i,3-l^*}^{(T_1+T_0)} + \tilde{\Omega}\left(\frac{\eta}{k}\right) \sum_{t=T_1+T_0}^{T_1+T_2} \mathbb{E}_{(X,y)\sim\tilde{\mathcal{Z}}_{u,m}}[(1 - \mathbf{logit}_y(F^{(t)}, \mathcal{A}(X)))] - \tilde{O}\left(\frac{T_2\eta s\tilde{N}_{c,s}}{k^2\tilde{N}_c}\right).$$

Then combining that $\Phi_{i,3-l^*}^{(t)} \leq \tilde{O}(1)$ from the Induction Hypothesis 15(d), we have:

$$\sum_{t=T_1+T_0}^{T_1+T_2} \mathbb{E}_{(X,y)\sim\tilde{\mathcal{Z}}_{u,m}}[(1 - \mathbf{logit}_i(F^{(t)}, \mathcal{A}(X)))] \leq \tilde{O}\left(\frac{k}{\eta}\right) + \tilde{O}\left(\frac{T_2 s\tilde{N}_{c,s}}{k\tilde{N}_c}\right) \leq \tilde{O}\left(\frac{k}{\eta}\right).$$

$\square$

Now we are ready to prove the following claim on the correlations between model kernels and the learned feature $\Phi_{i,l^*}^{(t)}$ is retained during learning Phase II.

**Claim 27** (learned is retained). *Suppose that the Induction Hypothesis 15 holds for every iteration $T_1 < t \leq T_1 + T_2$, under Parameter Assumption 10 and 12, we have for every iteration $T_1 < t \leq T_1 + T_2$:*

$$\forall (i, l^*) \in \mathcal{M}, \quad \Phi_{i,l^*}^{(t)} \geq \Omega(\log(k)).$$

*Proof.* For any $w_{i,r}$ and $v_{i,l^*}$ ($i \in [k], r \in [m]$), we have

$$\langle w_{i,r}^{(t+1)}, v_{i,l^*} \rangle = \langle w_{i,r}^{(t)}, v_{i,l^*} \rangle - \eta \mathbb{E}_{(X,y) \sim \mathcal{Z}_l}[\langle \nabla_{w_{i,r}} L_s(F^{(t)}; X, y), v_{i,l^*} \rangle]$$
$$- \eta \mathbb{E}_{(X,y) \sim \tilde{\mathcal{Z}}_u}[\langle \nabla_{w_{i,r}} L_u(F^{(t)}; X, y), v_{i,l^*} \rangle].$$

For the loss term on $\mathcal{Z}_l$, as discussed in Claim 24, we have

$$-\mathbb{E}_{(X,y) \sim \mathcal{Z}_l}[\langle \nabla_{w_{i,r}} L_s(F^{(t)}; X, y), v_{i,l^*} \rangle] \leq \frac{1}{\text{poly}(k)}.$$

Again, by Claim 19 and Claim 20, we know

$$\langle w_{i,r}^{(t+1)}, v_{i,l^*} \rangle \geq \langle w_{i,r}^{(t)}, v_{i,l^*} \rangle + \eta \mathbb{E}_{(X,y) \sim \tilde{\mathcal{Z}}_u}[\mathbb{I}_{y=i}(V_{i,r,l^*}(X) - \tilde{O}(\sigma_p P))(1 - \mathbf{logit}_i(F^{(t)}, \mathcal{A}(X)))$$
$$- \mathbb{I}_{y \neq i}(\mathcal{E}_1 + \mathcal{E}_3 + \mathbb{I}_{v_{i,l^*} \in \mathcal{P}(X)} V_{i,r,l^*}(X)) \mathbf{logit}_i(F^{(t)}, \mathcal{A}(X))].$$

Recall $V_{i,r,l^*}(X) := \mathbb{I}_{v_{i,l^*} \in \mathcal{V}(X)} \sum_{p \in \mathcal{P}_{v_{i,l^*}}(X)} \overline{\text{ReLU}}'(\langle w_{i,r}^{(t)}, \mathcal{A}(x_p) \rangle) \mathcal{A}(z_p)$ and our definition of strong augmentation in Eq. (14), we have $V_{i,r,l^*}(X) = 0$, so we have by Claim 23 that

$$\langle w_{i,r}^{(t+1)}, v_{i,l^*} \rangle \geq \langle w_{i,r}^{(t)}, v_{i,l^*} \rangle - \tilde{O}\left(\frac{\eta \sigma_p P}{k}\right) \mathbb{E}_{(X,y) \sim \tilde{\mathcal{Z}}_u}[1 - \mathbf{logit}_i(F^{(t)}, \mathcal{A}(X))] - O\left(\frac{\eta(\mathcal{E}_1 + \mathcal{E}_3)}{k}\right).$$

Summing up all $r \in [m]$ and using $m = \text{polylog}(k)$, we have

$$\Phi_{i,l^*}^{(t+1)} \geq \Phi_{i,l^*}^{(t-1)} - \tilde{O}\left(\frac{\eta \sigma_p P}{k}\right) \mathbb{E}_{(X,y) \sim \tilde{\mathcal{Z}}_u}[1 - \mathbf{logit}_i(F^{(t)}, \mathcal{A}(X))] - \tilde{O}\left(\frac{\eta \gamma(\sigma_0^{q-1} s + (\sigma_0 \gamma k)^{q-1} P)}{k}\right).$$

In the following, we separate the process into $T_1 < t \leq T_1 + T_0$ and $t \geq T_1 + T_0$.

**When $T_1 < t \leq T_1 + T_0$.** Recall at the end of learning Phase I, we have $\Phi_{i,l^*}^{(T_1)} \geq \Omega(\log(k))$. Using $T_0 = \tilde{\Theta}\left(\frac{k}{\eta \sigma_0^{q-2}}\right)$ and our Parameter Assumption 10, we have $\Phi_{i,l^*}^{(t)} \geq \Omega(\log(k))$ for every iteration of $T_1 < t \leq T_1 + T_0$.

**When $T_1 + T_0 < t \leq T_1 + T_2$.** By the upper bound on multi-view error in Claim 26, we know $\Phi_{i,l^*}^{(t)} \geq \Omega(\log(k))$ for every iteration of $T_1 + T_0 < t \leq T_1 + T_2$. □

Next, we present our last claim on individual error similar to Claim D.16 in Allen-Zhu & Li (2023). It states that when training error on $\tilde{\mathcal{Z}}_{u,m}$ is small enough, the model has high probability to correctly classify any individual data.

**Claim 28** (individual error). *When $\mathbb{E}_{(X,y) \sim \tilde{\mathcal{Z}}_{u,m}}\left[1 - \mathbf{logit}_y\left(F^{(t)}, X\right)\right] \leq \frac{1}{k^4}$ is sufficiently small, we have for any $(i, 3-l), (j, 3-l') \notin \mathcal{M}$,*

$$0.4 \Phi_{i,3-l}^{(t)} - \Phi_{j,3-l'}^{(t)} \leq -\Omega(\log(k)), \quad \Phi_{i,3-l}^{(t)}, \Phi_{j,3-l'}^{(t)} \geq \Omega(\log(k)),$$

*and therefore for every $(X, y) \in \mathcal{Z}$ (and every $(X, y) \in \mathcal{D}$ w.p. $1 - e^{-\Omega(\log^2(k))}$),*

$$F_y^{(t)}(X) \geq \max_{j \neq y} F_j^{(t)}(X) + \Omega(\log k).$$

*Proof.* Denote by $\tilde{\mathcal{Z}}^*_{u,m}$ for the sample $(X,y) \in \tilde{\mathcal{Z}}_{u,m}$ such that $\sum_{p \in P_{v_{y,3-l^*}}(X)} z_p \leq 1 + \frac{1}{100 \log(k)}$ where $(y, l^*) \in \mathcal{M}$. For a sample $(X,y) \in \tilde{\mathcal{Z}}^*_{u,m}$, denote by $\mathcal{H}(X)$ as the set of all $i \in [k] \setminus \{y\}$ such that $\sum_{p \in P_{v_{i,3-l}}(X)} z_p \geq 0.4 - \frac{1}{100 \log(k)}$ where $(i, l) \in \mathcal{M}$.

Now, suppose $1 - \mathbf{logit}_y(F^{(t)}, X) = \mathcal{E}(X)$, with $\min(1, \beta) \leq 2(1 - \frac{1}{1+\beta})$, we have

$$\min(1, \sum_{i \in [k] \setminus \{y\}} e^{F_i^{(t)}(X) - F_y^{(t)}(X)}) \leq 2\mathcal{E}(X)$$

By Claim 21 and our definition of $\mathcal{H}(X)$, this implies that

$$\min(1, \sum_{i \in \mathcal{H}(X)} e^{0.4 \Phi_{i,3-l}^{(t)} - \Phi_{y,3-l^*}^{(t)}}) \leq 4\mathcal{E}(X).$$

If we denote by $\psi = \mathbb{E}_{(X,y) \sim \tilde{\mathcal{Z}}_{u,m}} [1 - \mathbf{logit}_y(F^{(t)}, X)]$, then

$$\mathbb{E}_{(X,y) \sim \tilde{\mathcal{Z}}_{u,m}} \left[ \min(1, \sum_{i \in \mathcal{H}(X)} e^{0.4 \Phi_{i,3-l}^{(t)} - \Phi_{y,3-l^*}^{(t)}}) \right] \leq O(\psi),$$

$$\implies \mathbb{E}_{(X,y) \sim \tilde{\mathcal{Z}}_{u,m}} \left[ \sum_{i \in \mathcal{H}(X)} \min(\frac{1}{k}, e^{0.4 \Phi_{i,3-l}^{(t)} - \Phi_{y,3-l^*}^{(t)}}) \right] \leq O(\psi).$$

Notice that we can rewrite the LHS so that

$$\mathbb{E}_{(X,y) \sim \tilde{\mathcal{Z}}_{u,m}} \left[ \sum_{j \in [k]} \mathbb{I}_{j=y} \sum_{i \in [k]} \mathbb{I}_{i \in \mathcal{H}(X)} \min(\frac{1}{k}, e^{0.4 \Phi_{i,3-l}^{(t)} - \Phi_{j,3-l'}^{(t)}}) \right] \leq O(\psi),$$

$$\implies \sum_{j \in [k]} \sum_{i \in [k]} \mathbb{I}_{i \neq y} \mathbb{E}_{(X,y) \sim \tilde{\mathcal{Z}}_{u,m}} \left[ \mathbb{I}_{j=y} \mathbb{I}_{i \in \mathcal{H}(X)} \right] \min(\frac{1}{k}, e^{0.4 \Phi_{i,3-l}^{(t)} - \Phi_{j,3-l'}^{(t)}}) \leq O(\psi),$$

where $(j, l') \in \mathcal{M}$. Note for every the probability for every $i \neq j \in [k]$, the probability of generating a sample $(X,y) \in \tilde{\mathcal{Z}}^*_{u,m}$ with $y = j$ and $i \in \mathcal{H}(X)$ is at least $\tilde{\Omega}(\frac{1}{k} \cdot \frac{s^2}{k^2})$. This implies

$$\sum_{i \in [k] \setminus \{j\}} \min(\frac{1}{k}, e^{0.4 \Phi_{i,3-l}^{(t)} - \Phi_{j,3-l'}^{(t)}}) \leq \tilde{O}\left(\frac{k^3}{s^2} \psi\right).$$

Then, with $1 - \frac{1}{1+\beta} \leq \min(1, \beta)$, we have for every $(X,y) \in \tilde{\mathcal{Z}}_{u,m}$,

$$1 - \mathbf{logit}_y(F^{(t)}, X) \leq \min(1, \sum_{i \in [k] \setminus \{y\}} 2 e^{0.4 \Phi_{i,3-l}^{(t)} - \Phi_{y,3-l^*}^{(t)}})$$

$$\leq k \cdot \sum_{i \in [k] \setminus \{y\}} \min(\frac{1}{k}, e^{0.4 \Phi_{i,3-l}^{(t)} - \Phi_{y,3-l^*}^{(t)}}) \leq \tilde{O}\left(\frac{k^4}{s^2} \psi\right). \tag{15}$$

Thus, we can see that when $\psi \leq \frac{1}{k^4}$ is sufficiently small, we have for any $i \in [k] \setminus \{y\}$

$$e^{0.4 \Phi_{i,3-l}^{(t)} - \Phi_{y,3-l^*}^{(t)}} \leq \frac{1}{k} \implies 0.4 \Phi_{i,3-l}^{(t)} - \Phi_{y,3-l^*}^{(t)} \leq -\Omega(\log(k)).$$

By symmetry and non-negativity of $\Phi_{i,3-l}^{(t)}$, we know for any $(i, 3-l), (j, 3-l') \notin \mathcal{M}$, we have:

$$0.4 \Phi_{i,3-l}^{(t)} - \Phi_{j,3-l'}^{(t)} \leq -\Omega(\log(k)), \quad \Phi_{i,3-l}^{(t)}, \Phi_{j,3-l'}^{(t)} \geq \Omega(\log(k)). \tag{16}$$

Since Eq. (16) holds for any $(i, 3-l), (j, 3-l') \notin \mathcal{M}$ at iteration $t$ such that $\mathbb{E}_{(X,y) \sim \tilde{\mathcal{Z}}_m} [1 - \mathbf{logit}_y(F^{(t)}, X)] \leq \frac{1}{k^4}$, and from Claim D.16 in Allen-Zhu & Li (2023) we know Eq. (16) also holds for $\Phi_{i,l}, \Phi_{j,l'}$ for any $(i, l), (j, l') \in \mathcal{M}$ during learning Phase II, so we have

- for every $(X,y) \sim \mathcal{Z}_m$, by Claim 21 we have

$$F_y^{(t)}(X) \geq 1 \cdot \Phi_y - O(\frac{1}{\text{polylog}(k)}) \geq 0.4 \max_{j \neq y} \Phi_j + \Omega(\log(k)) \geq \max_{j \neq y} F_j^{(t)}(X) + \Omega(\log k).$$

- for every $(X, y) \sim \mathcal{Z}_s$, suppose $v_{y,l}$ is its only semantic feature, by Claim 21 we have

$$F_y^{(t)}(X) \geq 1 \cdot \Phi_{y,l} - O(\frac{1}{\text{polylog}(k)}) \geq \Omega(\log(k)),$$

$$F_j^{(t)}(X) \leq O(\Gamma) \cdot \Phi_{j,l} + O(\frac{1}{\text{polylog}(k)}) \leq O(1) \text{ for } j \neq y.$$

Therefore, we have

$$F_y^{(t)}(X) \geq \max_{j \neq y} F_j^{(t)}(X) + \Omega(\log k).$$

$\square$

Let $T_1 + T_0'$ be the first iteration that $\mathbb{E}_{(X,y)\sim\tilde{\mathcal{Z}}_{u,m}}\left[1 - \textbf{logit}_y\left(F^{(t)}, X\right)\right] \leq \frac{1}{k^4}$, then we know for $t \geq T_1 + T_0'$ Eq. (16) always holds, since the objective $L^{(t)} = L_s^{(t)} + \lambda L_u^{(t)}$ ($\lambda = 1$) is $O(1)$-Lipschitz smooth and we are using full gradient descent, which means the objective value is monotonically non-increasing. Since in Phase II, $L_s^{(t)}$ is kept at a small value, $L_u^{(t)}$ is monotonically non-increasing.

## E.2 MAIN LEMMAS TO PROVE THE INDUCTION HYPOTHESIS 15

In this section, we show lemmas that when combined together, shall prove the Induction Hypothesis 15 holds for every iteration.

### E.2.1 CORRELATION GROWTH

**Lemma 29.** *Suppose Parameter 10 holds and suppose Induction Hypothesis 15 holds for all iteration $< t$ starting from $T_1$. Then, letting $\Phi_{i,l}^{(t)} := \sum_{r \in [m]} [\langle w_{i,r}^{(t)}, v_{i,l} \rangle]^+$, we have for every $i \in [k], l \in [2]$,*

$$\Phi_{i,l}^{(t)} \leq \tilde{O}(1).$$

*Proof.* For every $i \in [k]$ and every $(i, l) \in \mathcal{M}$, after the Phase I, we have $\Phi_{i,l}^{(T_1)} \leq \tilde{O}(1)$. In Phase II, as in SA-CutOut, the learned features $(i, l)$ are removed and so the correlations between gradients and learned features are kept small. This means that $\Phi_{i,l}^{(t)} \leq \tilde{O}(1)$ holds true in learning Phase II for $T_1 < t \leq T_1 + T_2$.

Then for the unlearned feature $(i, 3 - l)$, we suppose $t > T_1 + T_0'$ is some iteration so that $\Phi_{i,3-l}^{(t)} \geq \text{polylog}(k)$. We will prove that if we continue from iteration $t$ for at most $T_2$ iterations, we still have $\Phi_{i,3-l}^{(t)} \leq \tilde{O}(1)$. Based on Claim 19, we have that

$$\langle w_{i,r}^{(t+1)}, v_{i,3-l} \rangle$$
$$\leq \langle w_{i,r}^{(t)}, v_{i,3-l} \rangle + \eta \mathbb{E}_{(X,y)\sim\tilde{\mathcal{Z}}_u}\left[\mathbb{I}_{i=y}(\mathcal{E}_1 + \mathcal{E}_3 + V_{i,r,3-l})(1 - \textbf{logit}_i(F^{(t)}, \mathcal{A}(X)))\right]$$
$$\leq \langle w_{i,r}^{(t)}, v_{i,3-l} \rangle + O(\eta)\mathbb{E}_{(X,y)\sim\tilde{\mathcal{Z}}_{u,m}}\left[\mathbb{I}_{i=y}(1 - \textbf{logit}_i(F^{(t)}, \mathcal{A}(X)))\right] + O\left(\frac{\eta\rho\tilde{N}_{c,s}}{k\tilde{N}_c}\right).$$

This is because that when $(X, y) \sim \tilde{\mathcal{Z}}_{u,m}$, we have $V_{i,r,3-l} \leq O(1)$ and when $(X, y) \sim \tilde{\mathcal{Z}}_{u,s}$, we have $V_{i,r,3-l} \leq O(\rho)$. For every $(X, y) \sim \tilde{\mathcal{Z}}_{u,m}$, when $y = i$, we have

$$F_i^{(t)}(\mathcal{A}(X)) \geq \Phi_{i,3-l}^{(t)} \cdot \sum_{p \in \mathcal{P}_{v_{i,3-l}}} z_p - O(\frac{1}{\text{polylog}(k)}) \geq \Phi_{i,3-l}^{(t)} - O(\frac{1}{\text{polylog}(k)}).$$

Then when $j \neq y$ and $(j, l') \in \mathcal{M}$, we have

$$F_j^{(t)}(\mathcal{A}(X)) \leq \Phi_{j,3-l'}^{(t)} \cdot \sum_{p \in \mathcal{P}_{v_{j,3-l'}}} z_p + O(\frac{1}{\text{polylog}(k)}) \leq 0.4\Phi_{j,3-l'}^{(t)} + O(\frac{1}{\text{polylog}(k)}).$$

So by Eq. (16), we have

$$1 - \mathbf{logit}_y(F; \mathcal{A}(X), y) \leq \frac{1}{k^{\Omega(\log k)}}.$$

Summing up over all $r \in [m]$, we have

$$\Phi_{i,3-l}^{(t+1)} \leq \Phi_{i,3-l}^{(t)} + \frac{\eta m}{k^{\Omega(\log k)}} + \tilde{O}\left(\frac{\eta \rho \tilde{N}_{c,s}}{k \tilde{N}_c}\right).$$

Therefore, if we continue for $T_2$ iterations, we still have $\Phi_{i,3-l^*}^{(T_1+T_2)} \leq \tilde{O}(1)$. $\qquad\square$

### E.2.2 Off-Diagonal Correlations are Small

**Lemma 30.** *Suppose Parameter 10 holds and suppose Induction Hypothesis 15 holds for all iteration $< t$ starting from $T_1$. Then,*

$$\forall i \in [k], \forall r \in [m], \forall j \in [k] \setminus \{i\}, \quad |\langle w_{i,r}^{(t)}, v_{j,l}\rangle| \leq \tilde{O}(\sigma_0).$$

*Proof.* In Phase I when $t \leq T_1$, from Lemma D.22 in Allen-Zhu & Li (2023), we have $|\langle w_{i,r}^{(t)}, v_{j,l}\rangle| \leq \tilde{O}(\sigma_0)$. Now we consider Phase II when $t > T_1$, and denote by $R_i^{(t)} := \max_{r \in [m], j \in [k]\setminus\{i\}} |\langle w_{i,r}^{(t)}, v_{j,l}\rangle|$. According to Claim 19 and Claim 20, we have

$$R_i^{(t+1)} \leq R_i^{(t)} + \eta \mathbb{E}_{(X,y)\sim\tilde{\mathcal{Z}}_u}\left[\mathbb{I}_{y=i}(\mathcal{E}_{2,i,r}(X) + \mathcal{E}_1 + \mathcal{E}_3 + \mathcal{E}_{4,j,l}(X))(1 - \mathbf{logit}_i(F^{(t)}, X))\right]$$
$$+ \eta \mathbb{E}_{(X,y)\sim\tilde{\mathcal{Z}}_u}\left[\mathbb{I}_{y\neq i}(\mathcal{E}_1 + \mathcal{E}_3 + \mathcal{E}_{4,j,l}(X))\mathbf{logit}_i(F^{(t)}, X)\right].$$

For single-view data $(X,y) \sim \tilde{\mathcal{Z}}_{u,s}$, by Claim 23, we have $\mathbf{logit}_i(F^{(t)}, \mathcal{A}(X)) = O\left(\frac{1}{k}\right)$ for every $i \in [k]$. In the following, we separate the process into $T_1 < t \leq T_1 + T_0$ and $T_1 + T_0 < t \leq T_1 + T_2$.

**When $T_1 < t \leq T_1 + T_0$.** During this stage, by Claim 23 we know $\mathbf{logit}_i(F^{(t)}, X) = O(\frac{1}{k})$ ($\forall i \in [k]$) for any $(X,y) \sim \tilde{\mathcal{Z}}_u$. We also have $\mathcal{E}_{2,i,r}(X) \leq \tilde{O}(\gamma(\Phi_{i,3-l^*}^{(t)})^{q-1})$ with $(i, l^*) \in \mathcal{M}$, and have $\mathcal{E}_{4,j,l}(X) \leq \tilde{O}(\sigma_0)^{q-1}\mathbb{I}_{v_{j,l}\in\mathcal{V}(X)}$ by definition. Recall when $T_1 < t \leq T_1 + T_0$, by Claim 24, we have $\Phi_{i,3-l^*}^{(t+1)} = \Phi_{i,3-l^*}^{(t)} + \tilde{\Theta}\left(\frac{\eta}{k}\right)\overline{\mathrm{ReLU}}'(\Phi_{i,3-l^*}^{(t)})$, so $\sum_{T_1<t\leq T_1+T_0}\eta(\Phi_{i,3-l^*}^{(t)})^{q-1} \leq \tilde{O}(k)$. Also, $\Pr(v_{i,3-l^*} \in \mathcal{P}(X)|i \neq y) = \frac{s}{k}$. Therefore, for every $T_1 < t \leq T_1 + T_0$ with $T_0 = \tilde{\Theta}(\frac{k}{\eta\sigma_0^{q-2}})$, we have

$$R_i^{(t)} \leq R_i^{(T_1)} + \tilde{O}(\sigma_0) + \tilde{O}\left(\frac{\eta}{k}T_0\right)\left((\sigma_0^{q-1})\gamma s + (\sigma_0\gamma k)^{q-1}\gamma P + (\sigma_0)^{q-1}\frac{s}{k}\right) \leq \tilde{O}(\sigma_0).$$

**When $T_1 + T_0 < t \leq T_1 + T_2$.** During this stage, we have the naive bound on $\mathcal{E}_{2,i,r}(X) \leq \gamma$, so again by Claim 19 and Claim 20, we have

$$R_i^{(t+1)} \leq R_i^{(t)} + \frac{\eta}{k}\mathbb{E}_{(X,y)\sim\tilde{\mathcal{Z}}_u}[(\gamma + (\sigma_0^{q-1})\gamma s + (\sigma_0\gamma k)^{q-1}\gamma P + (\sigma_0)^{q-1}\frac{s}{k})(1 - \mathbf{logit}_i(F^{(t)}, X))].$$

Therefore, by the upper bound on multi-view error in Claim 26 and $\frac{\tilde{N}_{c,s}}{\tilde{N}_c} \leq \frac{k^2}{\eta s T_2}$, we know $R_i^{(t)} \leq \tilde{O}(\sigma_0)$ for $T_1 + T_0 < t \leq T_1 + T_2$. $\qquad\square$

### E.2.3 Noise Correlation is Small

**Lemma 31.** *Suppose Parameter 10 holds and suppose Induction Hypothesis 15 holds for all iteration $< t$ starting from $T_1$. For every $l \in [2]$, for every $r \in [m]$, for every $(X,y) \in \tilde{\mathcal{Z}}_u$ and $i \in [k]$:*

    *(a) For every $p \in \mathcal{P}_{v_{i,l}}(X)$, we have: $|\langle w_{i,r}^{(t)}, \xi_p\rangle| \leq \tilde{o}(\sigma_0)$.*

    *(b) For every $p \in \mathcal{P}(X) \setminus (\mathcal{P}_{v_{i,1}}(X) \cup \mathcal{P}_{v_{i,2}}(X))$, we have: $|\langle w_{i,r}^{(t)}, \xi_p\rangle| \leq \tilde{O}(\sigma_0)$.*

*(c) For every $p \in [P] \setminus \mathcal{P}(X)$, we have:* $|\langle w_{i,r}^{(t)}, \xi_p \rangle| \leq \tilde{O}(\sigma_0 \gamma k)$.

**Proof.** Based on gradient calculation Fact 16 and $|\langle x'_{p'}, \xi_p \rangle| \leq \tilde{O}(\sigma_p) \leq o(\frac{1}{\sqrt{d}})$ if $X' \neq X$ or $p' \neq p$, we have that for every $(X, y) \sim \tilde{\mathcal{Z}}_u$ and $p \in [P]$, if $i = y$

$$\langle w_{i,r}^{(t+1)}, \xi_p \rangle = \langle w_{i,r}^{(t)}, \xi_p \rangle + \tilde{\Theta}(\frac{\eta}{\tilde{N}_c}) \overline{\text{ReLU}}'(\langle w_{i,r}^{(t)}, x_p \rangle)(1 - \textbf{logit}_i(F^{(t)}, X)) \pm \frac{\eta}{\sqrt{d}}.$$

Else if $i \neq y$,

$$\langle w_{i,r}^{(t+1)}, \xi_p \rangle = \langle w_{i,r}^{(t)}, \xi_p \rangle - \tilde{\Theta}(\frac{\eta}{\tilde{N}_c}) \overline{\text{ReLU}}'(\langle w_{i,r}^{(t)}, x_p \rangle) \textbf{logit}_i(F^{(t)}, X) \pm \frac{\eta}{\sqrt{d}}.$$

Suppose that it satisfies that $|\langle w_{i,r}^{(t)}, x_p \rangle| \leq A$ for every $t < t_0$ where $t_0$ is any iteration $T_1 \leq t_0 \leq T_1 + T_2$. When $T_1 \leq t \leq T_1 + T_0$, we have that

$$\langle w_{i,r}^{(t)}, \xi_p \rangle \leq \langle w_{i,r}^{(T_1)}, \xi_p \rangle + \tilde{O}\left(\frac{T_0 \eta A^{q-1}}{\tilde{N}_c}\right) + \frac{T_0 \eta}{\sqrt{d}}$$

$$\leq \tilde{o}(\sigma_0) + \tilde{O}\left(\frac{k A^{q-1}}{\tilde{N}_c \sigma_0^{q-2}}\right) + \frac{T_0 \eta}{\sqrt{d}},$$

where the last step is because $T_0 = \tilde{\Theta}(\frac{k}{\eta \sigma_0^{q-2}})$. When $T_1 + T_0 \leq t \leq T_1 + T_2$, for multi-view data $(X, y) \sim \tilde{\mathcal{Z}}_{u,m}$, based on (15) in Claim 28, we can obtain that

$$\langle w_{i,r}^{(t)}, \xi_p \rangle \leq \langle w_{i,r}^{(T_1+T_0)}, \xi_p \rangle + \tilde{O}\left(\frac{k^5 A^{q-1}}{s^2 \tilde{N}_c}\right) + \frac{(T_2 - T_0)\eta}{\sqrt{d}}$$

$$\leq \tilde{O}\left(\frac{k A^{q-1}}{\tilde{N}_c \sigma_0^{q-2}} + \frac{k^5 A^{q-1}}{s^2 \tilde{N}_c}\right) + \frac{T_2 \eta}{\sqrt{d}}.$$

For single-view data $(X, y) \sim \tilde{\mathcal{Z}}_{u,s}$, we have that

$$\langle w_{i,r}^{(t)}, \xi_p \rangle \leq \langle w_{i,r}^{(T_1+T_0)}, \xi_p \rangle + \tilde{O}\left(\frac{T_2 \eta A^{q-1}}{\tilde{N}_c}\right) + \frac{(T_2 - T_0)\eta}{\sqrt{d}}$$

$$\leq \tilde{O}\left(\frac{k A^{q-1}}{\tilde{N}_c \sigma_0^{q-2}} + \frac{T_2 \eta A^{q-1}}{\tilde{N}_c}\right) + \frac{T_2 \eta}{\sqrt{d}}.$$

When $p \in \mathcal{P}_{v_{i,l}}(X)$, we have $|\langle w_{i,r}^{(t)}, x_p \rangle| \leq \tilde{O}(1)$ from Induction Hypothesis 15. Then plugging in $A = \tilde{O}(1), \tilde{N}_c \geq \tilde{\Omega}\left(\frac{k}{\sigma_0^{q-1}}\right), \tilde{N}_c \geq \tilde{\Omega}\left(\frac{k^5}{\sigma_0}\right)$ and $\tilde{N}_c \geq \eta T_2 \text{poly}(k)$, we can obtain that $|\langle w_{i,r}^{(t)}, \xi_p \rangle| \leq \tilde{o}(\sigma_0)$.

When $p \in \mathcal{P}(X) \setminus (\mathcal{P}_{v_{i,1}}(X) \cup \mathcal{P}_{v_{i,2}}(X))$, we have $|\langle w_{i,r}^{(t)}, x_p \rangle| \leq \tilde{O}(\sigma_0)$ from the Induction Hypothesis 15. Then plugging in $A = \tilde{O}(\sigma_0), \tilde{N}_c \geq k^5$ and $\tilde{N}_c \geq \eta T_2 \text{poly}(k)$, we can obtain that $|\langle w_{i,r}^{(t)}, \xi_p \rangle| \leq \tilde{O}(\sigma_0)$.

When $p \in [P] \setminus \mathcal{P}(X)$, we have $|\langle w_{i,r}^{(t)}, x_p \rangle| \leq \tilde{O}(\sigma_0 \gamma k)$ from Induction Hypothesis 15. Then plugging in $A = \tilde{O}(\sigma_0 \gamma k), \tilde{N}_c \geq k^5$ and $\tilde{N}_c \geq \eta T_2 \text{poly}(k)$, we can obtain that $|\langle w_{i,r}^{(t)}, \xi_p \rangle| \leq \tilde{O}(\sigma_0 \gamma k)$.

□

### E.2.4 DIAGONAL CORRELATIONS ARE NEARLY NON-NEGATIVE

**Lemma 32.** *Suppose Parameter Assumption 10 holds and suppose Induction Hypothesis 15 holds for all iteration $< t$ starting from $T_1$. Then,*

$$\forall i \in [k], \quad \forall r \in [m], \quad \forall l \in [2], \quad \langle w_{i,r}^{(t)}, v_{i,l} \rangle \geq -\tilde{O}(\sigma_0).$$

*Proof.* From Lemma D.27 in Allen-Zhu & Li (2023), we know $\langle w_{i,r}^{(t)}, v_{i,l} \rangle \geq -\tilde{O}(\sigma_0)$ for every iteration $t \leq T_1$. Now we consider any iteration $t > T_1$ so that $\langle w_{i,r}^{(t)}, v_{i,l} \rangle \leq -\tilde{\Omega}(\sigma_0)$. We start from this iteration to see how negative the next iterations can be. Without loss of generality, we consider the case when $\langle w_{i,r}^{(t')}, v_{i,l} \rangle \leq -\tilde{\Omega}(\sigma_0)$ holds for every $t' \geq t$. By Claim 19 and Claim 20,

$$\langle w_{i,r}^{(t+1)}, v_{i,l} \rangle \geq \langle w_{i,r}^{(t)}, v_{i,l} \rangle + \eta \mathbb{E}_{(X,y) \sim \tilde{\mathcal{Z}}_u} \Big[ \mathbb{I}_{y=i} (V_{i,r,l}(X) - \tilde{O}(\sigma_p P))(1 - \mathbf{logit}_i(F^{(t)}, X))$$
$$- \mathbb{I}_{y \neq i} \left( \mathcal{E}_1 + \mathcal{E}_3 + \mathbb{I}_{v_{i,l} \in \mathcal{P}(X)} V_{i,r,l}(X) \right) \mathbf{logit}_i(F^{(t)}, X) \Big]$$

Recall by Induction Hypothesis 15(a),

$$V_{i,r,l}(X) = \sum_{p \in P_{v_{i,l}}(X)} \overline{\mathrm{ReLU}}'(\langle w_{i,r}^{(t)}, x_p \rangle) z_p = \sum_{p \in P_{v_{i,l}}(X)} \overline{\mathrm{ReLU}}'(\langle w_{i,r}, v_{i,l} \rangle z_p \pm \tilde{o}(\sigma_0)) z_p.$$

Since we have assumed $\langle w_{i,r}^{(t)}, v_{i,l} \rangle \leq -\tilde{\Omega}(\sigma_0)$, so $V_{i,r,l}(X) = 0$, and we have

$$\langle w_{i,r}^{(t+1)}, v_{i,l} \rangle \geq \langle w_{i,r}^{(t)}, v_{i,l} \rangle - \eta \mathbb{E}_{(X,y) \sim \tilde{\mathcal{Z}}_u} \Big[ \mathbb{I}_{y=i} \tilde{O}(\sigma_p P)(1 - \mathbf{logit}_i(F^{(t)}, X)) \quad (17)$$
$$+ \mathbb{I}_{y \neq i} (\mathcal{E}_1 + \mathcal{E}_3) \mathbf{logit}_i(F^{(t)}, X) \Big].$$

We first consider every $t \leq T_1 + T_0$. Using Claim 23 we have $\mathbf{logit}_i(F^{(t)}, X) = O\left(\frac{1}{k}\right)$, which implies

$$\langle w_{i,r}^{(t)}, v_{i,l} \rangle \geq -\tilde{O}(\sigma_0) - O\left(\frac{\eta T_0}{k}\right)(\mathcal{E}_1 + \mathcal{E}_3) \geq -\tilde{O}(\sigma_0).$$

As for $t > T_1 + T_0$, combining with Claim 26 and the fact that $\mathbf{logit}_i(F^{(t)}, X) \leq 1 - \mathbf{logit}_y(F^{(t)}, X)$ for $i \neq y$, we have

$$\langle w_{i,r}^{(t)}, v_{i,l} \rangle \geq \langle w_{i,r}^{(T_0)}, v_{i,l} \rangle - \tilde{O}(k)(\mathcal{E}_1 + \mathcal{E}_3) \geq \langle w_{i,r}^{(T_0)}, v_{i,l} \rangle - \tilde{O}(\sigma_0) \geq -\tilde{O}(\sigma_0).$$

$\square$

### E.2.5   PROOF OF INDUCTION HYPOTHESIS 15

Now we are ready to prove our Induction Induction Hypothesis 15, the proof is similar to Theorem D.2 in Allen-Zhu & Li (2023).

**Lemma 33.** *Under Parameter Assumption 10, for any $m = \mathrm{polylog}(k)$ and sufficiently small $\eta \leq \frac{1}{\mathrm{poly}(k)}$, our Induction Hypothesis 15 holds for all iterations $t = T_1, T_1 + 1, \ldots, T_1 + T_2$.*

*Proof.* At iteration $t$, we first calculate

$$\forall p \in P_{v_{j,l}}(X): \quad \langle w_{i,r}^{(t)}, x_p \rangle = \langle w_{i,r}^{(t)}, v_{j,l} \rangle z_p + \sum_{v' \in \mathcal{V}} \alpha_{p,v'} \langle w_{i,r}^{(t)}, v' \rangle + \langle w_{i,r}^{(t)}, \xi_p \rangle, \quad (18)$$

$$\forall p \in [P] \setminus P(X): \quad \langle w_{i,r}^{(t)}, x_p \rangle = \sum_{v' \in \mathcal{V}} \alpha_{p,v'} \langle w_{i,r}^{(t)}, v' \rangle + \langle w_{i,r}^{(t)}, \xi_p \rangle. \quad (19)$$

By Allen-Zhu & Li (2023) we already know Induction Hypothesis 15 holds at iteration $t = T_1$. Suppose Induction Hypothesis 15 holds for all iterations $< t$ starting from $T_1$. We have established several lemmas:

$$\text{Lemma 29} \implies \forall i \in [k], \forall r \in [m], \forall l \in [2] : \langle w_{i,r}^{(t)}, v_{i,l} \rangle \leq \tilde{O}(1), \quad (20)$$

$$\text{Lemma 30} \implies \forall i \in [k], \forall r \in [m], \forall j \in [k] \setminus \{i\} : |\langle w_{i,r}^{(t)}, v_{j,l} \rangle| \leq \tilde{O}(\sigma_0), \quad (21)$$

$$\text{Lemma 32} \implies \forall i \in [k], \forall r \in [m], \forall l \in [2] : \langle w_{i,r}^{(t)}, v_{i,l} \rangle \geq -\tilde{O}(\sigma_0). \quad (22)$$

- To prove Induction Hypothesis 15(a), it suffices to plug Eq. (21), Eq. (22) into Eq. (18), use $\alpha_{p,v'} \in [0, \gamma]$, use $|\mathcal{V}| = 2k$, and use $|\langle w_{i,r}^{(t)}, \xi_p \rangle| \leq \tilde{o}(\sigma_0)$ from Lemma 31.

- To prove Induction Hypothesis 15(b), it suffices to plug Eq. (20), Eq. (21) into Eq. (18), use $\alpha_{p,v'} \in [0, \gamma]$, use $|\mathcal{V}| = 2k$, and use $|\langle w_{i,r}^{(t)}, \xi_p \rangle| \leq \tilde{O}(\sigma_0)$ from Lemma 31.

- To prove Induction Hypothesis 15(c), it suffices to plug Eq. (20), Eq. (21) into Eq. (19), use $\alpha_{p,v'} \in [0, \gamma]$, use $|\mathcal{V}| = 2k$, and use $|\langle w_{i,r}^{(t)}, \xi_p \rangle| \leq \tilde{O}(\sigma_0 \gamma k)$ from Lemma 31.

- To prove Induction Hypothesis 15(d), it suffices to note that Eq. (20) implies $\Phi_{i,l}^{(t)} \leq \tilde{O}(1)$, and note that Claim 24 implies $\Phi_{i,l}^{(t)} \geq \Omega(\Phi_{i,l}^{(0)}) \geq \tilde{\Omega}(\sigma_0)$.

- To prove Induction Hypothesis 15(e), it suffices to invoke Eq. (22).

$\square$

### E.3 PROOF OF THEOREM 13

Recall our training objective is

$$L^{(t)} = L_s^{(t)} + \lambda L_u^{(t)} = \mathbb{E}_{(X,y) \sim \mathcal{Z}_l}[-\log \mathbf{logit}_y(F^{(t)}, X)] + \mathbb{E}_{X \sim \tilde{\mathcal{Z}}_u}[-\log \mathbf{logit}_b(F^{(t)}, \mathcal{A}(X))].$$

From Allen-Zhu & Li (2023) we know $\mathbb{E}_{(X,y) \sim \mathcal{Z}_l}[-\log \mathbf{logit}_y(F^{(T_1)}, X)] \leq \frac{1}{\text{poly}(k)}$ holds at the end of learning Phase I, and according to Claim 27 we now this continues to hold true during learning Phase II. For $\mathbb{E}_{X \sim \tilde{\mathcal{Z}}_u}[-\log \mathbf{logit}_b(F^{(t)}, \mathcal{A}(X))]$, since we have for every data $(X, y) \sim \tilde{\mathcal{Z}}_u$ ($y = b$):

- if $\mathbf{logit}_y(F^{(t)}, \mathcal{A}(X)) \geq \frac{1}{2}$, then we know $-\log \mathbf{logit}_y(F^{(t)}, \mathcal{A}(X)) \leq O\left(1 - \mathbf{logit}_y(F^{(t)}, \mathcal{A}(X))\right)$;

- if $\mathbf{logit}_y(F^{(t)}, \mathcal{A}(X)) \leq \frac{1}{2}$, this cannot happen for too many tuples $(X, y, t)$ thanks to Claim 26, and when this happens we have a naive bound $-\log \mathbf{logit}_y(F^{(t)}, \mathcal{A}(X)) \in [0, \tilde{O}(1)]$ using Claim 23.

Therefore, by Claim 26, we know when $T_2 \geq \frac{\text{poly}(k)}{\eta}$,

$$\frac{1}{T_2} \sum_{t=T_1+T_0}^{T_1+T_2} \mathbb{E}_{(X,y) \sim \tilde{\mathcal{Z}}_u}\left[-\log \mathbf{logit}_y(F^{(t)}, X)\right] \leq \frac{1}{\text{poly}(k)}.$$

Moreover, since we are using full gradient descent and the objective function is $O(1)$-Lipschitz continuous, the objective value decreases monotonically. Specifically, this implies that

$$\mathbb{E}_{(X,y) \sim \tilde{\mathcal{Z}}_u}[1 - \mathbf{logit}_y(F^{(T)}, \mathcal{A}(X))] \leq \mathbb{E}_{(X,y) \sim \tilde{\mathcal{Z}}_u}[-\log \mathbf{logit}_y(F^{(T)}, \mathcal{A}(X))] \leq \frac{1}{\text{poly}(k)}$$

for the last iteration $T = T_1 + T_2$, which directly implies that the training accuracy is perfect.

As for the test accuracy, from Claim 28 and Claim D.16 in Allen-Zhu & Li (2023), we have for every $i, j \in [k]$,

$$\Phi_i^{(T)} - 0.4\Phi_j^{(T)} \geq \Omega(\log(k)), \quad \Phi_{i,1}^{(T)}, \Phi_{i,2}^{(T)}, \Phi_{j,1}^{(T)}, \Phi_{j,2}^{(T)} \geq \Omega(\log(k)).$$

This combined with the function approximation Claim 21 shows that with high probability $F_y^{(T)}(X) \geq \max_{j \neq y} F_j^{(T)}(X) + \Omega(\log k)$ for every $(X, y) \in \mathcal{D}_m, \mathcal{D}_s$, which implies that the test accuracy on both multi-view data and single-view data is perfect.

## F PROOF FOR FIXMATCH

In this section, we consider proving Theorem 11 on FixMatch. In this case, the formulation of strong augmentation $\mathcal{A}(\cdot)$ is defined in Eq. (10). For $(X, y) \in \tilde{Z}_{u,s}$ with $\hat{l}(X) = l^*$, the feature $v_{y,l^*}$ is removed with the probability $\pi_2$. When the patches of learned feature are removed, the left part is pure noise. In this way, same as Claim 23, for every $i \in [k]$, $\mathbf{logit}_i(F^{(t)}, \mathcal{A}(X)) = O(\frac{1}{k})$, and

$V_{i,r,l} = \mathbb{I}_{v_{i,l} \in \mathcal{V}(X)} \sum_{p \in \mathcal{P}_{v_{i,l}}(X)} \overline{\text{ReLU}}'(\langle w_{i,r}, x_p \rangle) z_p = o(\frac{1}{\text{polylog}(k)})$. When the patches of noises are removed, the left part is semantic patches of feature $v_{y,l^*}$. Since we have already captured this feature in learning Phase I, we have $\textbf{logit}_y(F^{(t)}, \mathcal{A}(X)) \geq 1 - \tilde{O}(\frac{1}{s^2})$. In both above cases, by Claim 19 and Claim 20, the training samples $(X, y) \in \tilde{\mathcal{Z}}_{u,s}$ contribute little to the weight update process.

For multi-view data $(X, y) \in \tilde{Z}_{u,m}$, when the patches of noises are removed with probability $1 - \pi_2$, since the learned feature $v_{y,l^*}$ of Phase I is still in the data, we have $\textbf{logit}_y(F^{(t)}, \mathcal{A}(X)) \geq 1 - \tilde{O}(\frac{1}{s^2})$. When the patches of unlearned feature $v_{y,3-l^*}$ are removed with probability $\pi_1 \pi_2$ or $(1 - \pi_1)\pi_2$ (depending on the value of $3 - l^*$), the learned feature of Phase I is also still in the data. Thus we also have $\textbf{logit}_y(F^{(t)}, \mathcal{A}(X)) \geq 1 - \tilde{O}(\frac{1}{s^2})$. In this way, the loss on all data points $(X, y) \in \tilde{\mathcal{Z}}_{u,m}$ that belongs to the above two cases keeps small ($\leq \frac{1}{\text{poly}(k)}$) and contributes negligible to the learning of unlearned features in Phase II. Finally, when the patches of learned feature $v_{y,l^*}$ are removed with probability $\pi_1 \pi_2$ or $(1-\pi_1)\pi_2$ (depending on the value of $l^*$), the remaining patches of feature $v_{y,3-l^*}$ are unlearned and the approximation of initial loss on this part of samples is the same as Claim 23. The loss on samples $(X, y) \sim \tilde{\mathcal{Z}}_{u,m}$ with learned features removed dominates the training objective in learning Phase II, and the rest proof schedule is the same as the proof of SA-FixMatch. However, now the size of data with learned features removed for class $i \in [k]$ is either $\pi_1 \pi_2 \cdot \tilde{N}_u^i$ or $(1-\pi_1)\pi_2 \cdot \tilde{N}_u^i$. Thus, similar to the proof of Theorem 13 and for the simplicity of notation, the requirement on the size of unlabeled data for FixMatch should be $\tilde{N}_u \geq \eta T_2 \cdot \text{poly}(k)/\min\{\pi_1\pi_2, (1-\pi_1)\pi_2\}$. Accordingly, we can derive that the relationship between the size of the unlabeled data in SA-FixMatch $N_c$ and FixMatch $N_u$ is given by $N_c = \max\{\pi_1\pi_2, (1-\pi_1)\pi_2\}N_u$.

## G    PROOF FOR FLEXMATCH, FREEMATCH, DASH, AND SOFTMATCH

Our analysis framework and theoretical results are also applicable to other FixMatch-like SSL, e.g., FlexMatch (Zhang et al., 2021a), FreeMatch (Wang et al., 2022b), Dash (Xu et al., 2021), and SoftMatch (Chen et al., 2023), since the main difference is the choice of confidence threshold $\mathcal{T}_t$ in unsupervised loss Eq. (5). Here we first introduce their choice of $\mathcal{T}_t$ and then explain how our theoretical results in Sec. 4 can be generalized to their case.

FlexMatch (Zhang et al., 2021a) designs an adaptive class-specific threshold $\mathcal{T}_t = \beta_t(b)\tau$ at iteration $t$, where $\beta_t(b) \in [0, 1]$ is the model's prediction confidence for class $b$ (L1 normalized). FreeMatch (Wang et al., 2022b) replaces $\tau$ in FlexMatch with an adaptive $\tau_t$, which is the average prediction confidence of the model on unlabeled data and increases as the training progresses. SoftMatch uses the average prediction confidence $\tau_t$ of the model as the threshold and sets the sample weight as 1.0 if $\textbf{logit}_b(F^{(t)}, \alpha(X_u)) \geq \tau_t$, otherwise a smaller constant according to a Gaussian function. Dash adopts the cross-entropy loss to design the indicator function $\mathbb{I}_{\{-\log \textbf{logit}_b(F^{(t)}, \alpha(X_u)) < \rho_t\}}$, where $\rho_t$ decreases as training processes. This is equivalent to a dynamically increasing threshold $\mathcal{T}_t$ in Eq. (5). Below we detail how our theoretical findings apply to each of these SSL algorithms.

FlexMatch (Zhang et al., 2021a) differentiates itself from FixMatch by modifying the constant threshold $\tau$ to include an adaptive class-specific threshold $\beta_t(b)$ for each class $b$. Under our multi-view data assumption as defined in Def. 7, the data distribution for each class is the same. Consequently, as suggested by Claim 24 and Claim D.10 in Allen-Zhu & Li (2023), all classes progress at a similar rate during training. This uniformity over all classes allows us to standardize $\beta_t(b) = 1, \forall b \in [k]$, thereby aligning the proof for FlexMatch with that of FixMatch.

For FreeMatch (Wang et al., 2022b) and SoftMatch (Chen et al., 2023), instead of applying a large constant threshold $\tau$ during the training process, they use an adaptive $\tau_t$ to involve more unlabeled data with correctly-predicted pseudo-label in the training of the network. Under our multi-view data assumption Def. 7, the majority of the data in training dataset is of multi-view (with probability $1 - \frac{1}{\text{poly}(k)}$), so we only consider the network's prediction confidence for multi-view data to determine $\tau_t$. We set the adaptive threshold $\tau_t$ as follows:

$$\tau_t = \begin{cases} \max_{X \in \mathcal{Z}_{u,m}}[\textbf{logit}_b(F^{(t)}, X)], & t = T_0, \\ \beta\tau_{t-1} + (1 - \beta)\max_{X \in \mathcal{Z}_{u,m}}[\textbf{logit}_b(F^{(t)}, X)], & t > T_0, \end{cases} \tag{23}$$

where $\beta$ is the momentum parameter, $b = \arg\max_i \mathbf{logit}_i(F^{(t)}, X)$, and $T_0 = \Theta(\frac{k}{\eta\sigma_0^{q-2}})$. Here we do not consider the unsupervised loss term Eq. (5) before $T_0$-th iteration in our analysis, since the model is bad at generating correct pseudo-label at the initial phase of training. We use $\max$ function here to ensure the high quality of pseudo-label for unlabeled data involved at each training step. After $T_0$-th iteration, according to Claim D.11 and Lemma D.22 in Allen-Zhu & Li (2023), the feature correlations increase to $\Lambda_i^{(t)} = \tilde{\Theta}(1)$ for $t \geq T_0$, while the off-diagonal correlations $\langle w_{i,r}, v_{j,l}\rangle$ $(i \neq j)$ keep small at the scale of $\tilde{O}(\sigma_0)$. Denote $\Phi^{(t)} = \max_{i\in[k],l\in[2]} \Phi_{i,l}^{(t)}$, recall from Claim D.9 in Allen-Zhu & Li (2023), for every $X \in \arg\max_{X\in\mathcal{Z}_{u,m}}[\mathbf{logit}_b(F^{(t)}, X)]$ with ground truth label $y$, we have $F_j^{(t)}(X) \leq 0.8001\Phi^{(t)}$ for $j \neq y$ and $F_y^{(t)}(X) \geq 0.9999\Phi^{(t)}$ with probability at least $1 - e^{-\Omega(\log^2 k)}$. Accordingly, we have $F_y^{(t)}(X) \geq \max_{j\neq y} F_j^{(t)}(X) + \tilde{\Theta}(1)$, which means that $F^{(t)}$ can correctly classify the unlabeled data with high probability (i.e., $b = y$). Therefore, when $t \geq T_0$, both the supervised loss Eq. (4) and unsupervised loss Eq. (5) take effect. Same as in Sec. 4, we use $\Phi_{i,l}^{(t)}$ here to monitor the feature learning process. For $(i, l) \in \mathcal{M}$, feature $v_{i,l}$ is partially learned during the first $T_0$ iterations in that $\Phi_{i,l}^{(T_0)} = \tilde{\Theta}(1) < \Omega(\log(k))$, while feature $v_{i,3-l}$ is missed in that $\Phi_{i,3-l}^{(T_0)} = \tilde{O}(\sigma_0) \ll \tilde{\Theta}(1)$. Start from $T_0$, feature $v_{i,l}$ is continued to be better learned with the help of supervised loss and unsupervised loss until $\Phi_{i,l}^{(t)} \geq \Omega(\log k)$, and feature $v_{i,3-l}$ start to be learned with the help of unsupervised loss. We can analyze this feature learning process using a similar approach as in Sec. E. The key intuition for the extension of the proof of FixMatch to FreeMatch and SoftMatch is that by setting an adaptive confidence threshold, the learning process of unlearned features begin at $T_0$ instead of $T_1 = \text{poly}(k)/\eta \gg T_0$ in FixMatch.

For Dash, it uses cross-entropy loss as the threshold indicator function rather than prediction confidence $\mathbb{I}_{\{-\log \mathbf{logit}_b(F^{(t)},\alpha(X_u))<\rho_t\}}$, where $\rho_t$ decreases as the training progresses. Since we have

$$-\log \mathbf{logit}_b(F^{(t)}, \alpha(X_u)) < \rho_t \iff \mathbf{logit}_b(F^{(t)}, \alpha(X_u)) > e^{-\rho_t},$$

we can set $\rho_t = -\log \tau_t$ and the rest of the analysis is the same as in SoftMatch and FreeMatch.

## H EFFECT OF STRONG AUGMENTATION ON SUPERVISED LEARNING

In this section, we show why using strong augmentation with probabilistic feature removal effect, such as CutOut, in supervised learning (SL) has minimal alternation to the feature learning process. In SL, strong augmentation $\mathcal{A}(\cdot)$ is utilized at the start of training, before any feature has been effectively learned, corresponding to Phase I of SSL. According to Assumption 9, $\mathcal{A}(\cdot)$ randomly removes its semantic patches and noisy patches with probabilities of $\pi_2$ and $1 - \pi_2$, respectively. Then for a single-view image, its only semantic feature is removed with probability $\pi_2$. For a multi-view image, one of the two features, $v_{i,1}$ or $v_{i,2}$ is removed with probabilities $\pi_1\pi_2$ and $(1 - \pi_1)\pi_2$, respectively. Thus, the size of single-view data in training dataset $\mathcal{Z}_l$ is increased, as $\mathcal{A}(\cdot)$ transfers $\pi_2 N_{l,m}$ multi-view samples to single-view.

However, $\pi_2 \in (0, 1)$ is small, since based on our data assumption in Def. 7, the number of patches associated with certain semantic feature is constant $C_p$ while the total number of patches is $P = k^2$. Therefore, when we do random masking in $\mathcal{A}(\cdot)$ (usually masks $1/4$ of all patches), we can approximate $\pi_2$ as $(C_p/P)^{C_p}$, which is $O(1/k^{C_p})$ based on our definition of $\mathcal{A}(\cdot)$ in Eq. (10).

Consequently, strong augmentation $\mathcal{A}(\cdot)$ only slightly increases the proportion of single-view data, and the majority of the training dataset remains multi-view, which dominates the supervised training loss Eq. (4) since no feature has been learned. The assumptions on the number of labeled single-view data $N_{l,s} \leq \tilde{o}(k/\rho)$ and $N_l \geq N_{l,s} \cdot \text{poly}(k)$ still hold after strong augmentation $\mathcal{A}(\cdot)$. Thus, according to Allen-Zhu & Li (2023) and Appendix B, the network learns one feature per class to correctly classify the majority multi-view data due to "view lottery winning", and memorizes the single-view data without learned feature during the training process of SL. We also validate the limited effect of CutOut on SL through experimental results in Appendix K.3.

## I    COMPARISON WITH PIONEERING WORK

While this work follows the data assumption and proof framework of Allen-Zhu & Li (2023), analysis of the feature learning process is significantly different. Firstly, this work focuses on SSL, where supervised loss on labeled data and unsupervised loss on unlabeled data result in rather different feature learning processes compared with supervised distillation loss on only labeled data in Allen-Zhu & Li (2023). Secondly, SSL uses the on-training model as an online teacher which varies along training iterations, while the SL setting in Allen-Zhu & Li (2023) uses a well-trained and fixed model as an offline teacher. Indeed, the online teacher in SSL setting is more challenging to analyze, as the evolution of its performance is harder to characterize, and has a rather different learning process.

## J    (SA-)FIXMATCH ALGORITHM

In this section, we present the detailed algorithm framework for FixMatch (Sohn et al., 2020) and SA-FixMatch. At iteration $t$, we first sample a batch of $B$ labeled data $\mathcal{X}^{(t)}$ from labeled dataset $\mathcal{Z}_l$, and a batch of $\mu B$ unlabeled data $\mathcal{U}^{(t)}$ from unlabeled dataset $\mathcal{Z}_u$. Then, according to Algorithm 1, we calculate the loss for current iteration, and use it for the update of the neural network model $F^{(t)}$. The only difference between FixMatch and SA-FixMatch is in line 6, where FixMatch adopts CutOut in its strong augmentation of unlabeled data $\mathcal{A}$, while SA-FixMatch adopts SA-CutOut.

---

**Algorithm 1** (SA-)FixMatch algorithm.

1: **Input:**  Labeled batch $\mathcal{X}^{(t)} = \{(X_i, y_i) : i \in (1, \dots, B)\}$, unlabeled batch $\mathcal{U}^{(t)} = \{U_i : i \in (1, \dots, \mu B)\}$, confidence threshold $\tau$, unlabeled data ratio $\mu$, unlabeled loss weight $\lambda$.
2: $L_s^{(t)} = \frac{1}{B} \sum_{i=1}^{B} \left( -\log \mathbf{logit}_{y_i}(F^{(t)}, \alpha(X_i)) \right)$ {*Cross-entropy loss for labeled data*}
3: **for** $i = 1$ **to** $\mu B$ **do**
4:     $v_i = \arg\max_j \{\mathbf{logit}_j(F^{(t)}, \alpha(U_i))\}$ {*Compute prediction after applying weak data augmentation of $U_i$*}
5: **end for**
6: $L_u^{(t)} = \frac{1}{\mu B} \sum_{i=1}^{\mu B} \left( -\mathbb{I}_{\{\mathbf{logit}_{v_i}(F^{(t)}, \alpha(U_i)) \geq \tau\}} \log \mathbf{logit}_{v_i}(F^{(t)}, \mathcal{A}(U_i)) \right)$ {*Cross-entropy loss with pseudo-label and confidence for unlabeled data*}
7: **return:** $L_s^{(t)} + \lambda L_u^{(t)}$

---

## K    EXPERIMENTAL DETAILS

### K.1    EFFECT OF STRONG AUGMENTATION

In this section, we conduct experiments to evaluate the impact of different strong augmentation operations employed in FixMatch (Sohn et al., 2020). We assess their effects by applying these strong augmentations to test images and observing the resulting changes in test accuracy. To ensure a fair comparison, we train neural networks using weakly-augmented labeled data, where the weak augmentation consists of a random horizontal flip and a slight extension of the image around its edges before cropping the main portion. Subsequently, we apply a single strong augmentation operation at a time to the test dataset and record the corresponding test accuracy on the pretrained model. The pool of strong augmentation operations from RandAugment (Cubuk et al., 2020) includes: Colorization, Equalize, Posterize, Solarize, Rotate, Sharpness, ShearX, ShearY, TranslateX, and TranslateY. The experimental results are summarized in Tables 5 and 6.

| Original | CutOut | ShearX | Solarize | TranslationX |
|----------|--------|--------|----------|--------------|
| 80.95 | 38.67 | 78.38 | 52.70 | 80.69 |
| Sharpness | Posterize | Equalize | Rotate | Color |
| 72.99 | 76.17 | 60.14 | 67.97 | 69.71 |

Table 5: Pretrained model test accuracies (%) with different strong augmentation operations for test images on CIFAR-100.

| Original | CutOut | ShearX | Solarize | TranslationX |
|----------|--------|--------|----------|--------------|
| 86.44 | 62.94 | 85.01 | 78.94 | 84.44 |
| Sharpness | Posterize | Equalize | Rotate | Color |
| 82.17 | 86.19 | 80.74 | 82.29 | 84.34 |

Table 6: Pretrained model test accuracies (%) with different strong augmentation operations for test images on STL-10.

From Tables 5 and 6, we observe that CutOut is the strong augmentation operation with the most significant impact on model performance. Additionally, transformations such as Solarize and Equalize from RandAugment also have a noticeable effect on model performance. To better understand the influence of these transformations on input images, we visualize the effects of CutOut, Solarize, and Equalize on CIFAR-100 images in Figure 3. From the first and second rows of Figure 3, we can see that both CutOut and Solarize have the potential to remove semantic features by masking parts of the images. From the third row of Figure 3, we observe that Equalize tends to remove color features of images while retaining shape features. In all cases, these effective strong augmentation operations have the potential to remove partial semantic features. Therefore, in Assumption 3 for strong augmentation $\mathcal{A}(\cdot)$, we focus on its probabilistic feature removal effect.

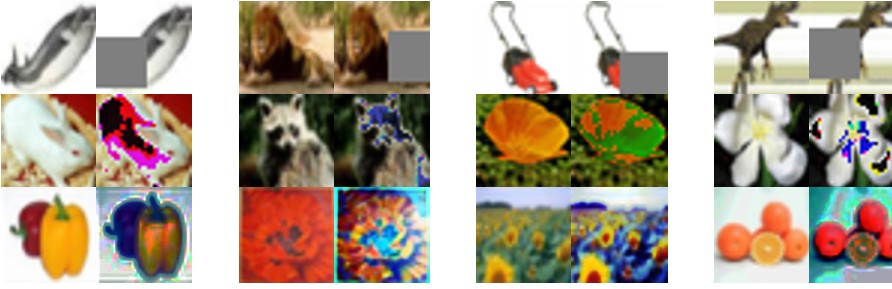

Figure 3: Visualization of the effects of CutOut (first row), Solarize (second row), and Equalize (third row) on CIFAR-100 images.

### K.2 Effect of Weak Augmentation

Since weak augmentation consists only of a random horizontal flip and a random crop with a small padding of 4 pixels, followed by cropping the padded image back to the original size, it minimally alters the semantic features of the image. This allows us to treat weak augmentation $\alpha(\cdot)$ as an identity mapping for our theoretical analysis in Sec. 4. In this section, we conduct experiments by training FixMatch without weak augmentation on CIFAR-100 with 10000 labeled samples and STL-10 with 1000 labeled samples, comparing the test performance to that of the original FixMatch. As shown in Table 7, weak augmentation does not significantly impact the model's performance.

| Dataset | STL-10 | CIFAR-100 |
|---------|--------|-----------|
| Weak Augmentation | 92.65 | 77.27 |
| No Weak Augmentation | 91.83 | 77.19 |

Table 7: Comparison of test accuracies (%) of FixMatch with and without weak augmentation.

### K.3 Comparison of CutOut in SL and FixMatch

Data augmentation operations like CutOut can help SL, but cannot improve as much as in deep SSL with limited labeled data. On STL-10 dataset with 40 labeled data, when we remove CutOut from SL, the test accuracy (%) does not drop a lot as shown in Table 8. In contrast, removing CutOut from the strong augmentation $\mathcal{A}(\cdot)$ in FixMatch's unsupervised loss $L_u^{(t)}$ leads to a severe performance degradation.

| Method | SL | FixMatch |
|--------|-------|----------|
| CutOut | 23.98 | 68.30 |
| No CutOut | 22.88 | 53.64 |

Table 8: Comparison of test accuracies (%) of SL and FixMatch with and without CutOut.

### K.4 DATASET STATISTICS

For each experiment in Sec. 5, following Sohn et al. (2020); Zhang et al. (2021a); Xu et al. (2021); Wang et al. (2022b); Chen et al. (2023), we randomly select image-label pairs from the entire training dataset according to labeled data amount, set images from the whole training dataset without labels as unlabeled dataset, and we use the standard test dataset. The table below details data statistics across different datasets.

| Dataset | Total Training Data | Total Labeled Data in Training Data | Test Data |
|---------|---------------------|-------------------------------------|-----------|
| STL-10 | 105000 | 5000 | 8000 |
| CIFAR-100 | 50000 | 50000 | 10000 |
| Imagewoof | 9025 | 9025 | 3929 |
| ImageNet | 1281167 | 1281167 | 50000 |

Table 9: Summary of Datasets.

### K.5 TRAINING SETTING AND HYPER-PARAMETERS

All experiments are conducted on four RTX 3090 GPUs (24GB memory). Due to resource limitations, we did not follow Sohn et al. (2020) by training for 1024 epochs with 1024 iterations per epoch ($2^{20}$ iterations in total). Instead, we run 150 epochs with 2048 iterations per epoch (307,200 iterations) for the standard setting in Sec. 5.1, and 50 epochs with 2048 iterations per epoch (102,400 iterations) for the same training dataset setting in Sec. 5.4. As shown in Sec. 5, our test accuracy results closely match those of Sohn et al. (2020). For FixMatch experiments, we base our implementation on Kim (2020), while all other experiments follow Wang et al. (2022a). Each SSL experiment takes 48 to 120 hours on a single RTX 3090 GPU, depending on the model and dataset.

For CIFAR-100, STL-10, Imagewoof, and ImageNet, their input image size are respectively $32 \times 32$, $96 \times 96$, $96 \times 96$, $224 \times 224$ and their mask size in CutOut are respectively $16 \times 16$, $48 \times 48$, $48 \times 48$, $112 \times 112$. For the application of SA-CutOut, according to our theoretical analysis in Sec. 4 and Appendix G, we only need it after partial feature already been learned to learn comprehensive features in the dataset. Therefore, in practice, we only apply SA-CutOut to deep SSL methods in the last 32 epochs of training, and the total running time of SA-FixMatch is roughly 1.15 times of FixMatch.

For hyper-parameters, we use the same setting following FixMatch (Sohn et al., 2020). Concretely, the optimizer for all experiments is standard stochastic gradient descent (SGD) with a momentum of 0.9 (Sutskever et al., 2013). For all datasets, we use an initial learning rate of 0.03 with a cosine learning rate decay schedule (Loshchilov & Hutter, 2016) as $\eta = \eta_0 \cos\left(\frac{7\pi k}{16K}\right)$, where $\eta_0$ is the initial learning rate, $k$ is the current training step and $K$ is the total training step that is set to 307200. We also perform an exponential moving average with the momentum of 0.999. The hyper-parameter settings are summarized in Table 10.

### K.6 SAMPLES IN CIFAR-100, STL-10, IMAGEWOOF, AND IMAGENET

As we can observe from Figure 4, for images in CIFAR-100 and STL-10, the semantic subject in the image occupies the majority of the image. On the other hand, for images in Imagewoof and ImageNet dataset, most semantic subject only occupies less than a quarter of the image.

| Dataset | CIFAR-100 | STL-10 | Imagewoof | ImageNet |
|---|---|---|---|---|
| Model | WRN-28-8 | WRN-37-2 | WRN-37-2 | ResNet-50 |
| Weight Decay | 1e-3 | 5e-4 | 5e-4 | 3e-4 |
| Batch Size | 64 | | | 128 |
| Unlabeled Data Raion $\mu$ | 7 | | | 1 |
| Threshold $\tau$ | 0.95 | | | 0.7 |
| Learning Rate $\eta$ | 0.03 | | | |
| SGD Momentum | 0.9 | | | |
| EMA Momentum | 0.999 | | | |
| Unsupervised Loss Weight $\lambda$ | 1 | | | |

Table 10: Complete hyper-parameter setting.

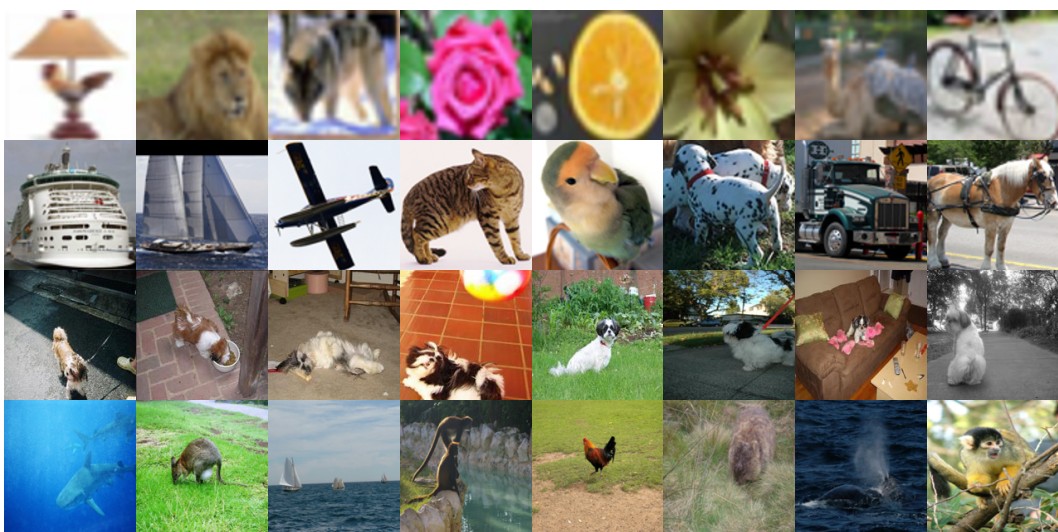

Figure 4: Samples from CIFAR-100, STL-10, Imagewoof, and ImageNet datasets. Samples in the first row are from CIFAR-100, samples in the second row are from STL-10, samples in the third row are from Imagewoof, and samples in the last row are from ImageNet.

## L  EXPLANATION OF MULTI-VIEW DATA ASSUMPTION

In this section, we make more detailed explanations of our multi-view data assumption Def. 1 by breaking it down and explain each part with specific examples from car images in ImageNet.

From Figure 1 we know that wheel and front light can be viewed as two discriminative semantic features of car, and each can be used independently for the learning model to make correct class prediction. In Figure 5, we give some examples of single-view data and multi-view data in car images that contains either only one of the two features or both. Then, we use them as examples to explain the multi-view data assumption Def. 1 in detail.

In Def. 1, the data distribution $\mathcal{D}$ is composed of samples from the multi-view data distribution $\mathcal{D}_m$ with probability $1 - \mu$, and from the single-view data distribution $\mathcal{D}_s$ with probability $\mu = \frac{1}{\text{poly}(k)}$. In the context of car images, this implies that the majority of images are multi-view, containing both wheel and front light features, while a small fraction of images are single-view, containing only one of these features.

Then, Def. 1 defines $(X, y) \sim \mathcal{D}$ by first randomly selecting a label $y \in [k]$ uniformly, and generate the data $X$ as follows:

**(a)** A set of noisy features $\mathcal{V}'$ is sampled uniformly at random from $\{v_{i,1}, v_{i,2}\}_{i \neq y}$, each with probability $s/k$. The complete feature set of $X$ is then defined as $\mathcal{V}(X) = \mathcal{V}' \cup \{v_{y,1}, v_{y,2}\}$, which includes both the noisy features $\mathcal{V}'$ and the main features $\{v_{y,1}, v_{y,2}\}$.

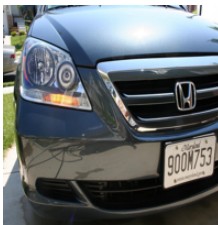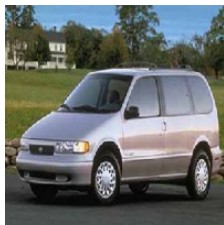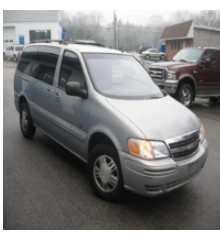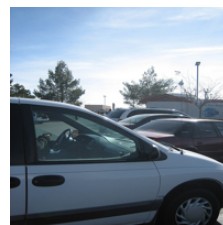

Figure 5: The first single-view image contains only the front light feature, while the middle two multi-view images contain both wheel and front light features, and the last single-view image contains only the wheel feature.

In the context of car images, (a) corresponds to the semantic features specific to cars (wheel and front light) being present in car images, along with noisy features from other classes, such as houses or trees in the background.

**(b)** For each $v \in \mathcal{V}(X)$, pick $C_p$ disjoint patches in $[P]$ and denote them as $\mathcal{P}_v(X)$. For a patch $p \in \mathcal{P}_v(X)$, we set $x_p = z_p v + \text{``noises''} \in \mathbb{R}^d$, where the coefficients $z_p \geq 0$ satisfy:
**(b1)** For "multi-view" data $(X, y) \in \mathcal{D}_m$, $\sum_{p \in \mathcal{P}_v(X)} z_p \in [1, O(1)]$ when $v \in \{v_{y,1}, v_{y,2}\}$ and $\sum_{p \in \mathcal{P}_v(X)} z_p \in [\Omega(1), 0.4]$ when $v \in \mathcal{V}(X) \setminus \{v_{y,1}, v_{y,2}\}$.
**(b2)** For "single-view" data $(X, y) \in \mathcal{D}_s$, pick a value $\hat{l} \in [2]$ randomly uniformly as the index of the main feature. Then $\sum_{p \in \mathcal{P}_v(X)} z_p \in [1, O(1)]$ when $v = v_{y,\hat{l}}$, $\sum_{p \in \mathcal{P}_v(X)} z_p \in [\rho, O(\rho)]$ $(\rho = k^{-0.01})$ when $v = v_{y,3-\hat{l}}$, and $\sum_{p \in \mathcal{P}_v(X)} z_p = \frac{1}{\text{polylog}(k)}$ when $v \in \mathcal{V}(X) \setminus \{v_{y,1}, v_{y,2}\}$.

In the context of car images, (b1) indicates that multi-view data features a significant presence of two prominent class-specific semantic features, while the proportion of noisy features in the image is relatively small. For example, in the middle two multi-view car images shown in Figure 5, the wheel and front light features are more prominent than the background elements, such as houses and trees. (b2) indicates that single-view data contains only one prominent class-specific semantic feature, with the other semantic feature and noisy features being minimal. As shown in the first and last single-view car images in Figure 5, these images prominently display either the wheel or the front light, while the other semantic feature and noisy background elements are scarcely present.

**(c)** For each purely noisy patch $p \in [P] \setminus \cup_{v \in \mathcal{V}} \mathcal{P}_v(X)$, we set $x_p = \text{``noises''}$.

In the context of car images, purely noisy patches correspond to regions such as road or sky patches that do not contain semantic information relevant to classification.

For a neural network capable of learning comprehensive semantic features after training–such as both the wheel feature and the front light feature for car images–can accurately predict both multi-view samples, such as the middle two images in Figure 5, and the single-view samples, such as the first and last images in Figure 5. However, if the network only learns partial semantic features, either the wheel or the front light feature, it will misclassify the single-view images that do not contain this feature, either the first or the last image in Figure 5 and result in inferior generalization performance.

