# OpenReview forum: "Towards Understanding Why FixMatch Generalizes Better Than Supervised Learning"
_ICLR.cc/2025/Conference — ICLR 2025 Oral_

### Official Review · Reviewer_6Nkk · 2024-10-31

**Soundness:** 3
**Presentation:** 3
**Contribution:** 4
**Rating:** 8
**Confidence:** 4

**Summary:**

This paper explores the theoretical aspects of why the SSL method FixMatch outperforms supervised learning method in generalization for deep neural networks (DNNs). Previous studies have shown that SSL methods like FixMatch achieve higher test accuracy, but the mechanisms behind this advantage are not obvious. The authors provide theoretical justification for the enhanced generalization of FixMatch for convolutional neural networks. Their analysis reveals that FixMatch captures all relevant discriminative features for each class, whereas SL approaches tend to capture only a random subset of features, an effect attributed to the lottery ticket hypothesis. This framework is shown to extend to other SSL methods similar to FixMatch, such as FlexMatch, FreeMatch, Dash, and SoftMatch. Based on these findings, the authors propose an enhanced version of FixMatch, called Semantic-Aware FixMatch (SA-FixMatch), which is validated experimentally, demonstrating improved generalization.

**Strengths:**

The theory presented is compelling. The authors provide a strong argument, without relying on overly strict assumptions, that training a realistic neural network (a 3-layer ConvNet) with FixMatch-type algorithms allows us to (1) fit the training data and (2) generalize well to unseen samples. This stands in contrast to supervised learning, where the model often fails to generalize well to certain types of samples within the distribution.

Additionally, the authors propose an improved variation of a FixMatch algorithm, demonstrating that their theory not only explains the success of this family of algorithms but also predicts new results.

**Weaknesses:**

The main weakness of this paper lies in its technical presentation.

While I appreciate that the theoretical framework developed here is complex, making it challenging to present in an accessible way, I believe certain aspects could have been simplified for clarity.

This could be achieved by following these guidelines:
1. Use standard notations. For instance, the authors use symbols like $Z_l$ to denote a labeled dataset, whereas $S$ is typically used for sample sets.
2. Avoid re-using variables. In lines 126-128, for example, the symbol $i$ is used for multiple purposes, such as indexing both patches and classes, which can be confusing.
3. Simplify complex definitions. Concepts like Definition 1 could be broken down and explained in more detail, with examples illustrating each component. Providing an example of a distribution that meets these conditions would clarify the distinction between single- and multi-view samples and help readers appreciate the significance of the conclusions in lines 284-287.

Minor comment:
In the theorems (e.g., Theorem 4), instead of writing "for any \((x,y) \sim D\) with probability ..., we have ...," I would suggest phrasing it as "with probability ... over the selection of \((x,y) \sim D\), we have ...". It is just more mathematically accurate and is consistent with the appendix.

**Questions:**

1. The theory relies on 3-layer ConvNets. However, the experiments obviously hold for a wider range of architectures. Is it possible to extend it to more sophisticated architectures. For example, ConvNets with residual connections, additional layers, ViTs? If so, would it change the results somehow? Can we derive conclusions that certain architectures generalize better with SL compared to other architectures? That could be really exciting!

2. Can you explain in theorem 4 why the margin scales as log(k) (where k is the number of classes). How come we get better classification margin for a more complex task with more classes?

3. In theorem 4 you use $T=poly(k)/\eta$ to represent the amount of iterations until convergence. What should I expect the degree of the polynomial and its leading coefficient to be? I want to have some concept of how many iterations we need.

---

> ### Author Response · Authors · 2024-11-21
>
> Thank you for the insightful and positive comments! Below, we provide a point-by-point response to your concerns.
>
> > The main weakness of this paper lies in its technical presentation.
>
> We greatly appreciate your suggestions for improving the technical presentation of our paper to enhance its clarity and accessibility. In response, we have addressed the re-use of variables and restated Theorem 4(b) in our revision, as per your recommendation. Note that Theorem 4(a) remains unchanged, as the probability there stems from network initialization rather than dataset sampling.
>
> To further clarify Definition 1, we have deconstructed it in Appendix L and provided detailed explanations with specific examples to better illustrate the data assumption. Additionally, we will adopt standard notations, such as using $\mathcal{S}$ instead of $\mathcal{Z}$ to represent the dataset, in the final revision. However, we have refrained from making this change in the current revision to avoid causing confusion for other reviewers.
>
> > The theory relies on 3-layer ConvNets. Is it possible to extend it to more sophisticated architectures? Would it change the results somehow? Can we prove certain architectures generalize better with SL compared to other architectures?
>
> While our experiments on SSL and SL are applicable to deeper and more sophisticated neural network architectures, our theoretical analysis focuses on a 3-layer convolutional neural network (CNN). This choice allows us to align with Allen-Zhu & Li (2023) and directly compare our (SA-)FixMatch results with their SL analysis. Extending the current theoretical framework to deeper and more sophisticated architectures poses significant challenges due to the highly non-convex nature of the loss surfaces. These complexities lead to local minima and saddle points, making the semantic feature learning process more difficult to analyze.
>
> Specifically, such an extension would require identifying a suitable indicator $\Phi_{i,l}^{(t)}$ to represent the learning status of each semantic feature $v_{i,l}$ and more fine-grained control of the noise in the images, which becomes increasingly necessary with more sophisticated networks. Nevertheless, we believe our two-phase feature learning analysis for FixMatch-like SSLs can be generalized to other network architectures with appropriate configurations. We leave this as an avenue for future work.
>
> Regarding the results, we hypothesize that the improved generalization performance and more comprehensive semantic feature learning exhibited by SSL compared to SL will persist across other neural network architectures under the multi-view data assumption. As for proving that certain architectures generalize better with SL compared to others, we agree this is a fascinating direction. However, achieving this within our current proof framework would require additional time and exploration.
>
> > Why does the margin scale as $\log(k)$ in Theorem 4 (where $k$ is the number of classes)? How do we get better classification margins for a more complex task with more classes?
>
> In Theorem 4, the margin scales as $\log(k)$ due to the following reasons:
>
> 1. According to Theorem 5, the feature learning indicator $\Phi_{i,l}^{(T)} \geq \Omega(\log k)$ after FixMatch's two-phase feature learning process. Further details can be found in Claim 28 in Appendix E.1.
>
> 2. From Claim 21 in Appendix D, we approximate the prediction function $F_i^{(t)}(X)$ as: $F_i^{(t)}(X) = \sum_{l \in [2]} \left( \Phi_{i,l}^{(t)} \times Z_{i,l}^{(t)}(X) \right) \pm O\left(\frac{1}{\text{polylog}(k)}\right),$
>    where $Z _{i,l}^{(t)}(X) = \mathbb{I} _{v _{i,l} \in \mathcal{V}(X)} (\sum _{p \in \mathcal{P} _{v _{i,l}}(X)} z _p )$ for $i \in [k]$ and $l \in [2]$.
>
> Based on the multi-view data assumption in Definition 1, for $(X, y) \in \mathcal{D}_m$, we have the following:
>
> - $\sum _{p \in \mathcal{P} _v(X)} z_p \in [1, O(1)]$ when $v \in \{v _{y, 1}, v _{y, 2}\}$.
> - $\sum _{p \in \mathcal{P} _v(X)} z_p \in [\Omega(1), 0.4]$ when $v \in \mathcal{V}(X) \setminus \{ v _{y, 1}, v _{y, 2} \} $.
>
> Thus, the margin satisfies: $F_y^{(T)}(X) \geq \max_{j \neq y} F_j^{(T)}(X) + \Omega(\log k).$
>
> To achieve better classification margins for more complex tasks with an increasing number of classes, we recommend employing more sophisticated data augmentation techniques, such as CutMix, or training the network for additional iterations. Applying more sophisticated data augmentations, such as CutMix, increases the classification difficulty for each training sample. This, in turn, contributes to a larger $\Phi_{i,l}^{(T)}$ at the end of training, as the network must learn more robust semantic features to achieve a low training loss. Additionally, training the network for more iterations further reduces the training loss. According to the proof in Claim 28, this results in a larger $\Phi_{i,l}^{(T)}$ at the conclusion of training.

---

> > ### Author Response · Authors · 2024-11-21
> >
> > > What should I expect the degree of the polynomial $T = \frac{\text{poly}(k)}{\eta}$ in Theorem 4 and its leading coefficient to be?
> >
> > The degree of the polynomial in $T = \frac{\text{poly}(k)}{\eta}$ is determined to be $k^5$, as established in Claim 26 and Claim 28. However, the exact value of the leading coefficient cannot be specified, as our data assumptions and analytical framework focus on the asymptotic scaling with respect to $k$, assuming a large number of classes, while disregarding constant factors. For example, for $(X, y) \in \mathcal{D} _m$, we assume that $\sum _{p \in \mathcal{P} _v(X)} z _p \in [1, O(1)]$ without imposing restrictions on the precise constant. This reflects the emphasis on understanding growth behavior rather than pinning down exact coefficients.

---

### Official Review · Reviewer_S4MR · 2024-11-03

**Soundness:** 3
**Presentation:** 4
**Contribution:** 3
**Rating:** 8
**Confidence:** 3

**Summary:**

- This paper provides a theoretical analysis, aimed at answering why FixMatch-like algorithms (for Semi-Supervised Learning, or, SSL) generalizes better than supervised learning.
- The analysis is focused on CNNs (unlike previous comparison works that provide analysis by using linear model assumptions)
- The paper proposes a improvement to FixMatch, called Semantic-Aware FixMatch (SA-FixMatch). The SA-FixMatch essentially masks out the semantically relevant parts of a high-confidence image sample (the region that is identified by GradCAM) in a CutOut-like fashion.

**Strengths:**

- The presentation of this work is impressive. The paper is not only easy to read, but the authors do a good job of highlighting their contributions and how it differs from previous works. The writing is clear and concise, and the figures and tables (although there are not that many) are not needlessly overcomplicated.
- The proposed SA-FixMatch seems like a intuitive improvement to FixMatch, and does show to improve on the performance of FixMatch.
- The theoretical justification in Section 4 seem to be sound.

**Weaknesses:**

- My main concern of this paper is the overall motivation. My main question for the authors is: Why do we need to have a good theoretical understanding of why FixMatch generalizes better than Supervised Learning? The following is my thought process: Let's say we have a dataset that is fully labeled. In this case, we would obviously use supervised learning (since we have all labels) to train the model. But now, let's consider the case where only 10% of the data is labeled. Obviously, given that SSL can leverage 90% of the dataset while SL can only leverage 10% (9x the size), we would apply SSL to train the model. We already know that leveraging more data will lead to better performance - so then what is the point of trying to theoretically understand why FixMatch generalizes better then SL, given that SL in this case is using a subset of the data that FixMatch is using? The worst case for SSL is that it performs equally as SL. As shown in the paper, FixMatch learns more semantic features, but that seems a bit obvious, since FixMatch is able to utilize the unlabeled samples, while SL receives no training from these unlabeled samples. Perhaps a fairer (and more interesting) setting would be to compare SSL vs Supervised learning, given the same number of total training samples (where the 'unlabeled' samples of the SSL dataset is labeled for SL). I hope I am not coming across as too offensive with this comment, but I am just trying to understand the significance of such analysis. I hope the authors can convince me otherwise.

- The implications of the analysis is somewhat underwhelming.
  - The proposed SA-Cutout does not feel like a novel contribution, given that there are previous works that use guided data augmentation for other tasks (e.g., "Crafting Better Contrastive Views for Siamese Representation Learning" in CVPR 2022). Also, there are some gradient-based masking techniques, such as "Adversarial Dropout for Supervised and Semi-supervised Learning" in AAAI 2018 that have very similar motivations as SA-Cutout, and the resulting solution is quite similar as well (masking out highly semantic regions).
  - Are there any other takeaways from this analysis? For example, could this type of analysis be extended to a broader scope?

---

**Post Rebuttal**

My concerns have been addressed.

**Questions:**

Questions were asked in the section above.

---

> ### Author Response · Authors · 2024-11-19
>
> > Why need a good theoretical understanding of why FixMatch generalizes better than SL?
>
> (1) We appreciate your thoughtful feedback and understand your concerns about the motivation behind the theoretical investigation of SSL in this work, as well as in numerous prior studies (e.g., Rigollet, 2007; Guo et al., 2020, and see more in submission). We would like to address this with three key points:
>
> a)	The additional data in SSL are unlabelled and are fundamentally different from labeled data in SL: Unlabelled data lack explicit supervision, and thus do not guarantee improved performance. Indeed, unlabeled data can sometimes degrade performance, particularly when their pseudo-labels are highly inaccurate. This distinction makes it critical to understand why and how SSL algorithms like FixMatch achieve superior generalization than SL. This question is far from trivial, and has been the subject of extensive prior research (see them in submission). These observations underscore the importance of developing a robust theoretical foundation to explain SSL's strengths, limitations, and the conditions under which it excels.
>
> b)	Unlabeled data are abundant and easily accessible (e.g., from the web), whereas labeled data require costly and time-consuming manual annotation, often in limited supply. This is particularly relevant in domains like medical imaging, where acquiring labeled data can be prohibitively expensive. SSL's ability to leverage unlabeled data makes it a practical and scalable solution for such settings. Understanding the theoretical principles behind SSL's effectiveness can help unlock its full potential, ensuring that it reliably outperforms SL in resource-constrained scenarios.
>
> c)	A solid theoretical understanding of why SSL generalizes better than SL is not just academic. Indeed, it directly informs the design of better SSL algorithms. For instance, in our analysis of FixMatch's feature learning process (Section 4.2), we discover the critical role of strong augmentation $\mathcal{A}(\cdot)$. This augmentation removes learned semantic features in Phase I from input images, forcing the model to learn new, unobserved semantic features in Phase II, thereby enhancing generalization. Inspired by this insight, we developed SA-FixMatch with SA-CutOut, which deterministically removes learned semantic features during Phase II. This strategy improves the efficiency of unlabeled data usage, enabling the model to learn a more comprehensive set of semantic features and achieve better generalization performance.
>
> Moreover, our theoretical analysis also provides a novel perspective on semantic feature learning that can guide future SSL research. By extending our two-phase feature learning framework (Section 4) to other neural network architectures, researchers can develop more sophisticated SSL algorithms tailored to these architectures. Such advancements can further facilitate the learning of comprehensive semantic features with limited labeled data, ultimately leading to even better generalization performance across diverse domains.
>
> > compare SSL vs SL with same number of total training samples
>
> (2) With the same labeled training dataset $\mathcal{D}$, SSL still outperforms SL both theoretically and empirically. In this setting, SL uses $\mathcal{D}$ for supervised training, while SSL uses $\mathcal{D}$ as its labeled dataset and simultaneously treats the label-ignored $\mathcal{D}$ as its unlabeled dataset.
>
> Theoretically, our analysis for SA-FixMatch can be extended to this scenario where SL and SSL share the same data. This is because SA-FixMatch assumes that the strong augmentation $\mathcal{A}_{SA}(\cdot)$ deterministically removes semantic features learned during Phase I from the unlabeled images (see Appendix E). As a result, even with the same labeled and unlabeled dataset, SA-FixMatch can still exploit the two-phase feature learning process to learn a more comprehensive set of semantic features compared to SL, ultimately achieving better generalization performance. This conclusion is rigorously supported by our proof in Appendix E.
>
> To validate this theory empirically, we conducted experiments comparing SA-FixMatch with SL under controlled settings. Following the experimental protocols of our manuscript and FixMatch, we trained WRN-28-8 on CIFAR-100 with 10,000 labeled samples and WRN-37-2 on STL-10 with 1,000 labeled samples. In both cases, SL and SA-FixMatch shared the same labeled dataset $\mathcal{D}$, with SA-FixMatch treating the label-ignored $\mathcal{D}$ as unlabeled dataset.
>
> The test accuracy (\%)  results are below, demonstrating that SA-FixMatch significantly outperforms SL even when both using the same training dataset. This not only highlights the superiority of SSL over SL but also further validates our theoretical insights. We have included the discussions and results in Appendix K.7 in our revision.
>
> ||CIFAR-100|STL-10|
> |------|------|------|
> |SL|63.48|67.29|
> |SA-FixMatch|68.30|79.74|

---

> > ### Author Response · Authors · 2024-11-19
> >
> > > there are previous works that use guided data augmentation for other tasks
> >
> > (3) While the overarching idea of SA-CutOut and the methods proposed in [1] and [2] is to leverage the model's learning status to enhance training, the specific methodologies are fundamentally different. SA-CutOut is designed to deterministically remove learned semantic feature patches from input images during Phase II. This is achieved by using Grad-CAM to localize the semantic regions in the images that have already been learned, ensuring that the model focuses on unlearned features for more comprehensive semantic representation.
> >
> > In contrast, [1] employs a Siamese network to identify semantic regions within an image, but its purpose is to generate contrastive views within these semantic regions rather than removing learned semantic features. The focus is on avoiding to generate false positive contrastive views rather than modifying the input to exclude previously learned features. Additionally, [2] takes a completely different approach by introducing adversarial dropout, which reconfigures the neural network architecture (specifically the dropout layers) to improve generalization performance. Unlike SA-CutOut, [2] does not operate on the input images but rather modifies the internal network structure to achieve its goals.
> >
> > [1] "Crafting Better Contrastive Views for Siamese Representation Learning" in CVPR 2022
> >
> > [2] "Adversarial Dropout for Supervised and Semi-supervised Learning" in AAAI 2018
> >
> >
> > > other takeaways from this analysis, could this type of analysis be extended to a broader scope?
> >
> > (4) Our two-phase feature learning analysis of FixMatch-like SSL methods can also be extended to explain the robustness of SSL to spurious correlations by adjusting our data assumption (Definition 1). In the original Definition 1, we assume that the scales of the two discriminative features, $v_{y,1}$ and $v_{y,2}$, for class $y$ are equivalent in the multi-view data (which constitutes the majority of the dataset), i.e., $\sum _{p \in \mathcal{P}_v(X)} z_p \in [1, O(1)]$ for $v \in \{v _ {y,1}, v _{y,2}\}$. This assumption ensures that both semantic features are equally challenging for the network to learn.
> >
> > To address the role of spurious correlations, we modify this assumption by introducing a spurious feature $v_{y,1'}$ whose scale is significantly greater than that of the true feature $v_{y,2}$. Under this setting, $v_{y,1'}$ turns out to be an easier feature for the model to learn. Model trained with SL is likely to rely solely on $v_{y,1'}$, overlooking the true feature $v_{y,2}$, thereby becoming biased toward the spurious correlation. In contrast, SSL's two-phase learning mechanism allows it to learn the true feature $v_{y,2}$ during the second phase, after the spurious feature $v_{y,1'}$ has already been leveraged. This highlights SSL's inherent robustness to spurious correlations compared to SL.
> >
> > Moreover, our proposed SA-CutOut method further enhances SSL's ability to address spurious correlations. By deterministically masking the spurious feature $v_{y,1'}$, SA-CutOut encourages the model to focus on learning $v_{y,2}$, the true feature, during the second learning phase. This approach not only mitigates the reliance on spurious correlations but also facilitates more comprehensive semantic feature learning, leading to improved generalization.
> >
> > This robustness is especially relevant in real-world applications. For instance, in medical imaging, spurious correlations can arise from biases such as scanner artifacts or irrelevant demographic features (e.g., age or gender) present in the data. An SSL framework equipped with SA-CutOut can prioritize learning the true pathological features (e.g., tumor shapes or densities), reducing the risk of biased predictions. Similarly, in autonomous driving, SSL can help the model learn critical features like road signs or pedestrian movements instead of spurious cues like weather conditions or shadows, ensuring safer and more reliable performance.
> >
> > While these insights are promising, providing a complete theoretical justification requires more time, and we leave this extension as future work. Nevertheless, the potential applications and implications of this approach underscore its practical importance.

---

> ### Comment · Reviewer_S4MR · 2024-11-23
> **Re: Official Comment by Authors**
>
> I'd like to thank authors for engaging in discussions.
>
> ---
> **On the motivation of the paper**
>
> I understand the importance of SSL, given that obtaining labeled samples can be difficult in certain domains (*e.g.,* Medical, as you mentioned).
>
> My concern was that when you frame it as an SSL vs SL analysis, it is not an apples to apples comparison, since SSL inherently gets access to more data; if the SSL algorithm is able to receive **any** amount of learning signal from this unlabeled data, it would outperform SL. This is not just true for FixMatch-based SSL algorithms, but other types as well.
>
> Based on the contents of the paper and your response, I would say it's more suitable to frame this type of analysis as a theoretical understanding of why consistency regularization can help generalization. Ultimately, the underlying paradigm that FixMatch (and perhaps other SSL algorithms) is using is a form of consistency regularization, where two differing inputs are made to produce consistent outputs. In fact, this type of theoretical understanding, I would say, is intriguing, and also has the added benefit of having a broader impact (could also be relevant to self-supervised learning). However, when you frame it more narrowly as just SL vs SSL (FixMatch), maybe not so much.
>
> With that said, I appreciate the comparison of SSL vs SL on the same number of training samples. I think this experimental setup is very appropriate and serves as a great demonstration of not just the proposed SA-FixMatch algorithm, but also the theoretical analysis presented in this paper. Do you also have some results using FixMatch, instead of SA-Fixmatch? Also, I'd love to see this experimental setup in the main paper instead of the Appendix, but ultimately it is your choice whether to make this change or not.
>
> ---
> **On the novelty of SA-Cutout**
>
> I don't agree that Adversarial Dropout [2] takes a "completely different approach". Dropout operates on the activations of a given layer. If you consider the NN as a recursive-like function, dropout is essentially masking the output of one layer, which is the input of the subsequent layer. If we generalize adversarial dropout to drop out the very first input, *i.e.,* the input image, we could draw parallels between SA-CutOut and Adversarial Dropout (as both methods use the gradient to select which "activations" to mask).
>
> In my original comment, I mentioned that SA-Cutout does not *feel* like a novel contribution. I still stand by this statement, particularly due to similarities with other gradient-based masking methods (Adv. dropout being one of them; there are a couple examples in the Continual Learning and Domain Generalization fields as well). However, I do acknowledge that this is more of a *gut feeling*, than a fact that I can ground with strong evidence.
>
> ---
> By the way, I am inclined to improve my evaluation, but I am still interested in what the authors think about my POV.
> Thanks,

---

> > ### Author Response · Authors · 2024-11-24
> >
> > Thank you for your thoughtful feedback. Below, we provide a point-by-point response to address your concerns.
> >
> > > If the SSL algorithm is able to receive any amount of learning signal from this unlabeled data, it would outperform SL.
> >
> > We agree that if an SSL algorithm can effectively leverage any useful learning signal from unlabeled data, it would outperform SL. However, to the best of our knowledge, no prior work has provided a theoretical proof that FixMatch-like SSLs, when applied to CNNs, generalize better than SL on classification tasks by utilizing unlabeled data.
> >
> > Our paper fills this gap by providing the first theoretical analysis that demonstrates how and why FixMatch-like SSL algorithms outperform SL on CNNs, leveraging the availability of unlabeled data. This contribution bridges the divide between theoretical understanding and practical outcomes in SSL research.
> >
> > Furthermore, as discussed earlier, our theoretical insights have facilitated the development of improved SSL algorithms, such as our proposed SA-FixMatch. They also offer a novel perspective on semantic feature learning, which can serve as a foundation for future SSL research and other related fields. Indeed, as you acknowledged, our theoretical framework highlights the critical role of consistency regularization, a widely used technique in many domains like self-supervised learning, and suggests its potential for broader applications and impact.
> >
> > > The underlying paradigm that FixMatch is using is a form of consistency regularization; it's more suitable to frame this type of analysis as a theoretical understanding of why consistency regularization can help generalization.
> >
> > Thank you for your thoughtful feedback. We agree that consistency regularization plays a crucial role in improving generalization in SSL. This is evident for two reasons: (1) as discussed in our manuscript, SSL incorporates two training losses—the vanilla supervised loss and consistency regularization; and (2) SSL demonstrates superior generalization performance compared to SL which relies solely on supervised loss.
> >
> > Furthermore, our theoretical analysis offers valuable insights into other settings that utilize consistency regularization. For example, our analytical framework could be extended to explain how self-supervised learning facilitates the acquisition of more comprehensive semantic features during pretraining. However, transitioning our analysis from the semi-supervised learning setting to self-supervised learning would require additional steps, particularly to account for the fine-tuning stage involved in self-supervised learning for downstream tasks. We recognize this as an intriguing direction and plan to explore it in future work.
> >
> > Considering these factors, we appreciate your suggestion to frame our theoretical analysis more broadly as a study of how consistency regularization enhances generalization. In our final revision, we will highlight the critical role of consistency regularization and discuss its potential implications for other settings, such as self-supervised learning, where consistency regularization is widely adopted. However, implementing these changes requires careful consideration and a holistic revision of the manuscript, which is currently constrained by the page limit. While we are unable to make these additions immediately, we will ensure that these points are addressed comprehensively in our final version.
> >
> > Once again, thank you for your constructive suggestions, which we believe will enhance the impact and clarity of our work.

---

> > > ### Author Response · Authors · 2024-11-24
> > >
> > > > I appreciate the comparison of SSL vs SL on the same number of training samples. Do you also have some results using FixMatch instead of SA-FixMatch?
> > >
> > > Per your suggestion, we have conducted additional experiments for FixMatch under the same number of training samples and summarize the results below, along with the results of SL and SA-FixMatch. These results show that under this setup, both FixMatch and SA-FixMatch significantly outperform SL, with SA-FixMatch achieving better generalization performance than FixMatch. This further validates our theoretical analysis and the effectiveness of the proposed SA-FixMatch algorithm.
> > >
> > > ||CIFAR-100|STL-10|
> > > |------|------|------|
> > > |SL|63.48|67.29|
> > > |FixMatch|67.94|79.15|
> > > |SA-FixMatch|68.30|79.74|
> > >
> > > We also agree that this experimental setup serves as a strong validation not only for SA-FixMatch but also for our theoretical framework. To strengthen the comprehensiveness of our evaluation, we plan to conduct additional experiments under this same-training-dataset setup, encompassing all cases reported in Table 1 and Table 2 of our manuscript. Moreover, we will refine our theoretical analysis and update related statements to formally incorporate this experimental setup for (SA-)FixMatch.
> > >
> > > Due to time constraints during the rebuttal phase, we were unable to complete these additional experiments and revisions. Therefore, we have first included the preliminary results in the Appendix, and then will provide a complete table of experimental results for the same-training-dataset setup in Section 5 of our final revision.
> > >
> > >
> > > > We could draw parallels between SA-CutOut and Adversarial Dropout (as both methods use the gradient to select which "activations" to mask).
> > >
> > > Thank you for your insightful observation. We agree that SA-CutOut and Adversarial Dropout share a high-level similarity, as both utilize gradient information to target well-learned parameters or features for masking and thus learn more comprehensive features. In our revision, we have included a discussion of Adversarial Dropout to highlight its relevance.
> > >
> > > However, the two methods differ significantly in their motivations and implementations. SA-CutOut, as detailed in Sec. 4.3, aims to enhance data efficiency by deterministically removing learned semantic regions from input images. This process encourages comprehensive learning of less-represented semantic features in Phase II. Conversely, Adversarial Dropout is rooted in adversarial training and self-ensembling. It uses the divergence between a randomly dropped network and an adversarially dropped network as a regularization strategy to improve training process for comprehensive feature learning.
> > >
> > > Methodologically, SA-CutOut employs Grad-CAM to localize well-learned semantic regions in the input image, which are then removed to facilitate targeted learning. In contrast, Adversarial Dropout leverages gradient information to approximate the optimal adversarial dropout configuration directly on the network’s activations.
> > >
> > > We hope this clarification addresses your concerns and highlights the distinctions between these methods. We are happy to engage in further discussions if needed.

---

> > > > ### Comment · Reviewer_S4MR · 2024-11-25
> > > >
> > > > Thank you for the ongoing discussion.
> > > >
> > > > ---
> > > >
> > > > The main issue I raised in my original review was: Why do we need a good theoretical understanding of FixMatch vs Supervised Learning, from the perspective that FixMatch inherently uses more data (albeit unlabeled) than Supervised Learning. I believe that the the authors have addressed this well in the rebuttal, and that we have been able to find a middle-ground that we can both agree on.
> > > >
> > > > To reiterate my perspective: I think that if we view this analysis from the perspective of why something like *consistency regularization* is effective for learning generalized features, this theoretical analysis provides good value. My concern with the original submission was that the authors seemed a bit fixated on SSL (specifically FixMatch) vs SL. To me, the scope felt a bit narrow, especially when you consider that SSL simply leverages more data than SL; and if the model can obtain *any positive* learning signal from additional unlabeled samples, we should expect SSL to outperform SL.
> > > >
> > > > The experiments on SL vs FixMatch vs SA-FixMatch using the same amount of data (+ unlabeled versions of the same data for SSL algorithms) adequately addressed this concern. As promised by the authors, I would love to see a similar experimental setup for the experiments in Table 1 and 2. Furthermore, I hope the authors will better highlight the implications of their work, not just in the domain of SSL, but in related domains as well.
> > > >
> > > > ---
> > > >
> > > > Regarding SA-Cutout, my comment was mostly based on gut feeling. I have seen a few papers that use Grad-CAM (or other forms of activation mapping) to mask out important features, which are then used in training. But again, I can't cite specific papers off the top of my head.
> > > >
> > > > With that said, I understand that the main point of the paper was to understand *why* FixMatch generalizes better than SL, not necessarily to propose a novel method. From this perspective, SA-Cutout serves its purpose as an example of how such understanding could aid in designing stronger algorithms. Thus, novelty in this case is not crucial, as long as it serves its value in the overall theme of the paper.
> > > >
> > > > ---
> > > >
> > > > Again, I'd like to thank the authors for engaging in discussions. I feel that I was harsh in my initial assessment, and the discussion phase has helped change my view. I have updated my score accordingly.

---

> > > > > ### Author Response · Authors · 2024-11-25
> > > > >
> > > > > Thank you for your thoughtful and constructive feedback, as well as for the opportunity to engage in meaningful discussions throughout the rebuttal process. We deeply appreciate the time and effort you have taken to review our work and share your valuable insights, which have greatly helped us refine and better position our paper!

---

### Official Review · Reviewer_V3gi · 2024-11-04

**Soundness:** 3
**Presentation:** 3
**Contribution:** 4
**Rating:** 8
**Confidence:** 3

**Summary:**

This paper proposes two contributions:
1. A theoretical analysis to explain why semi-supervised learning (SSL) techniques such as FixMatch generalize better than classical supervising learning (SL).
2. A new method FixMatch-SA (semantically aware) which builds on the analysis to further enhance FixMatch.
The improved performance of FixMatch serves to experimentally corroborate the theoretical analysis.

I understood the substantiating argument of the theoretical analysis as follows: the correct classification of sample is typically based on multiple features (at least 2). In SL, learning of all features is not necessary to minimize the loss. Meanwhile, in FixMatch, the strong augmentation drops some features and therefore requires the network to learn all the features to minimize the loss.

Disclaimer: the theoretical analysis felt above my skill, mathematically speaking. I tried to follow it to the best of my ability but there could be alternate conjectures which I am not aware of to explain the observed generalization gains.

**Strengths:**

1. The paper is well-written and gave me the impression that I was able to follow its goal.
2. The results from FixMatch-SA seem to confirm the pertinence of the analysis, and intuitively I found it made logical sense.
   - Some gains from CutOut-SA are truly impressive, including for recent FixMatch derivatives.
3. I particularly liked that the paper didn't limit itself to a theoretical analysis but also provided an experimental validation on common SSL benchmarks.
4. I find the FixMatch-SA method very elegant and effective and appears simple to implement which I consider a quality.

**Weaknesses:**

1. My own lack of knowledge on the theoretical side made it hard for me to estimate the originality of the approach. It's not per-se a weakness of the paper but rather a warning that I simply don't know.

Typos (obviously this didn't influence my rating, it's for authors to polish their manuscript)
- Line 87, wrong citation "FixMatch (Xie)" => "FixMatch (Sohn)"

**Questions:**

1. Do you feel there is more potential to be extracted from the CutOut-SA line of thinking? For example, could doing multiple cutouts on the image to enforce exactly one classifying feature being present in the strong augmentation be a future avenue of improvement? Or did you already try multiple variants of such schemes and found the one you eventually presented in the paper to be the best?

---

> ### Author Response · Authors · 2024-11-21
>
> > Do you feel there is more potential to be extracted from the CutOut-SA line of thinking? For example, could doing multiple cutouts on the image to enforce exactly one classifying feature being present in the strong augmentation be a future avenue of improvement?
>
> Thank you for your insightful comments. As described in Sec. 4.3, our current implementation of SA-CutOut leverages Grad-CAM to localize learned semantic regions and applies a mask centered on the region with the highest average attention score. To ensure a direct and fair comparison, we kept both the mask size and the number of masks identical to those in the original CutOut.
>
> You raise an excellent point about applying multiple square masks to more effectively eliminate learned semantic features. However, this approach carries an inherent risk of also removing unlearned semantic features. To explore your suggestion, we experimented with a "Double SA-CutOut" strategy. In this variant, after placing the initial mask centered on the region with the highest average attention score, we added a second mask centered on the point with the next-highest attention score outside the area covered by the first mask.
>
> Using the same experimental setup as detailed in our manuscript and FixMatch, we evaluated this approach by training WRN-28-8 on CIFAR-100 with 10,000 labeled samples and WRN-37-2 on STL-10 with 1,000 labeled samples. The test accuracy (%) results below show that "Double SA-FixMatch" can further improve SA-FixMatch’s performance, albeit with marginal gains in certain cases, such as STL-10.
>
> We will incorporate this multiple SA-CutOut approach into a full ablation study in our final revision to further analyze its effects.
>
> ||CIFAR-100|STL-10|
> |------|------|------|
> |FixMatch|77.27|93.88|
> |SA-FixMatch|77.40|94.13|
> |Double SA-FixMatch|77.64|94.16|
>
> Additionally, we see further potential in the SA-CutOut methodology. As analyzed in Sec. 4.2, removing learned semantic features through strong augmentations $\mathcal{A}(\cdot)$ enforces the learning of unlearned semantic features during Phase II of training. Furthermore, if we can not only remove the learned semantic features from input images but also increase the presence of unlearned features, the learning of comprehensive features in Phase II can be further facilitated, according to analysis in Appendix E.
>
> To leverage this insight, we propose an extension to SA-CutOut. Instead of replacing the learned semantic region with a gray patch (as in the current SA-CutOut), we suggest pasting a patch from another part of the same image that has low attention value, as detected by Grad-CAM, and is more likely to contain unlearned semantic features. This approach not only removes learned semantic features but also enriches the input image with unlearned features, making the learning of comprehensive features in Phase II more effective.
>
> While this idea appears promising, it requires additional implementation and experimental validation, which we aim to explore in future work.
>
>
> > Line 87, wrong citation "FixMatch (Xie)" => "FixMatch (Sohn)"
>
> Thank you for your kind reminder. We have corrected this in our revised version, and the change has been highlighted in blue for clarity.

---

> > ### Comment · Reviewer_V3gi · 2024-11-21
> >
> > Thanks for the detailed response, these look like very exciting prospects!

---

### Official Review · Reviewer_WDKZ · 2024-11-04

**Soundness:** 3
**Presentation:** 3
**Contribution:** 4
**Rating:** 8
**Confidence:** 4

**Summary:**

The paper studies the feature learning process of neural networks trained with the FixMatch method, which is a semi-supervised learning method, demonstrating its theoretical advantages on data distributions with a “multi-view” structure. The authors characterize the FixMatch learning process as a two-stage process: initially, the model learns like supervised learning and learns most of the features, followed by a second stage where it learns the missing features through unsupervised learning from augmented data. Based on these theoretical insights, the authors introduce a semantic-aware augmentation in FixMatch to enhance its performance.

**Strengths:**

1. This paper provides a new theoretical analysis of the FixMatch method, particularly on multi-view structured data distributions, demonstrating its effectiveness in learning features and its advantages over supervised learning. The characterization of FixMatch's two-stage learning process is insightful, offering a clearer understanding of how the model learns from both supervised and unsupervised data.

2. The authors propose a new semantic-aware augmentation technique that aligns with their theoretical findings, which improved the performance of FixMatch.

**Weaknesses:**

1. The assumptions regarding data augmentation appear artificial. The augmentation method knows which feature is in each patch and can distinguish between feature and noise patches. The augmentation randomly mask the noise patch and one of the feature, to enable the FixMatch to focus on the unlearned features. Even though such augmentation can be easily achieved in the theoretical setting, it is smarter than what is originally used in FixMatch.
2. The proposed SA-FixMatch, although is interesting and shares closer connection to the theory, introduces added complexity by using Grad-CAM for augmentation, which can slow down training.

**Questions:**

1. Why can’t the augmentation be agnostic about what the patch contains, what is the theoretical bottleneck here? What impact would a uniformly random mask have? Could there be a more realistic setting where distribution-agnostic data augmentation could still achieve similar results?
2. While the theory here follows very closely to that of AllenZhu and Li [2023], it seem to have missed some previous works exploring the effects of augmentation on feature learning process [1,2]. The authors can refer to the designs of augmentations and their corresponding analysis in these papers.

[1] Toward Understanding the Feature Learning Process of Self-supervised Contrastive Learning. Zixin Wen, Yuanzhi Li [ICML 2021]

[2] The Mechanism of Prediction Head in Non-contrastive Self-supervised Learning. Zixin Wen, Yuanzhi Li [NeurIPS 2022]

---

> ### Author Response · Authors · 2024-11-19
>
> Thank you for the insightful comments. Below we provide our point-by-point response and hope to address your concerns.
>
> > The assumptions regarding data augmentation appear artificial.
>
> For our assumption on strong augmentation $\mathcal{A}(\cdot)$ in Assumption 3, we focus on its probabilistic feature removal effect as explained in Sec. 3.2. According to multi-view data assumption in Def. 1, the inpute image is composed of feature patches of $v_{y,1}$ and $v_{y,2}$, and the noise patches. Therefore, for simplicity and clarity of analysis, we assume $\mathcal{A}(\cdot)$ to have probability $1-\pi_2$ to remove only the noise patches while retaining the feature patches of $v_{y,1}, v_{y,2}$, probability $\pi_1 \pi_2$ to remove only the patches of feature $v_{y,1}$ while retaining noise patches and patches of feature $v_{y,2}$, and probability $(1-\pi_1) \pi_2$ to remove only the patches of feature $v_{y,2}$. The reason for this assumption on strong augmentation $\mathcal{A}(\cdot)$ is that we want to represent the probability that $\mathcal{A}(\cdot)$ removes learned semantic feature of Phase I from the multi-view images, which is either $\pi_1 \pi_2$ or $(1-\pi_1) \pi_2$ according to Assumption 3. This portion of strongly-augmented unlabeled data containing only the unlearned feature and noise patches dominates the unsupervised loss, since the rest samples containing the learned feature are already correctly classified by the network after Phase I and contribute minimally to the training loss. Then, according to our analysis in Sec. 4.2, the network is able to learn the unlearned semantic features in Phase II and therefore achieve better generalization performance.
>
> > Why can’t the augmentation be agnostic about what the patch contains
>
> In practice, data-agnostic strong augmentation $\mathcal{A}$ may simultaneously remove portions of both the feature patches $v_{y,1}, v_{y,2}$ and noise patches. Nonetheless, our analysis remains valid under this setting. Under this setting, with probability $p = O(\frac{1}{k^{C_p}})$, strong augmentation $\mathcal{A}$ removes all $C_p$ feature patches of the learned feature from Phase I while preserving the $C_p$ feature patches of the unlearned feature. Assuming sufficient unlabeled data, where $|\mathcal{Z}_u| = \text{poly}(k) |\mathcal{Z}_l|$ and $p|\mathcal{Z}_u| > |\mathcal{Z}_l|$, the subset of strongly augmented unlabeled data that contains only the unlearned feature and noise patches becomes substantial in size and dominates the training loss since the rest samples containing the learned feature are already correctly classified by the network after Phase I and contribute minimally to the training loss. As outlined in Sec. 4.2, this enables the network to effectively learn comprehensive semantic features during Phase II of training.
>
>
> > The proposed SA-FixMatch can slow down training.
>
> According to our analysis in Section 4.2, the strong augmentation $\mathcal{A}$ only begins to take effect after the network has learned partial features during Phase I of the learning process. Based on this insight, we apply SA-CutOut exclusively during the final 32 epochs of training, as detailed in Appendix K.5. This selective application ensures computational efficiency while maintaining the intended benefits of SA-CutOut. As a result, the total runtime of SA-FixMatch is approximately 1.15 times that of FixMatch on all datasets, which we consider a reasonable trade-off given the significant improvement in generalization performance.
>
> > missed some previous works exploring the effects of augmentation on feature learning process
>
> Thank you for pointing out the missing references. While [1] analyzes the feature learning process of contrastive learning and emphasizes the critical role of data augmentation, our work focuses on semi-supervised learning (SSL) and its ability to learn comprehensive semantic features. In contrast, [1] targets contrastive learning, aiming to learn sparse features and avoid spurious dense features.
>
> Similarly, [2] investigates the significant role of the projection head in enabling non-contrastive self-supervised learning methods to learn comprehensive features. Their work also divides the learning process into multiple phases based on the nature of non-contrastive self-supervised learning. However, our study centers on FixMatch-like SSL methods, where the two-phase feature learning process arises from the combination of supervised and unsupervised loss objectives unique to SSL.
>
> We have incorporated these references into our revision and included a discussion of their relevance, with the updates highlighted in blue for clarity.
>
> [1] Toward Understanding the Feature Learning Process of Self-supervised Contrastive Learning. Zixin Wen, Yuanzhi Li [ICML 2021]
>
> [2] The Mechanism of Prediction Head in Non-contrastive Self-supervised Learning. Zixin Wen, Yuanzhi Li [NeurIPS 2022]

---

> > ### Comment · Reviewer_WDKZ · 2024-11-25
> >
> > Thanks for the response, particularly regarding the feature-agnostic data augmentations. I now believe the theoretical contribution of this work is solid and deserves a higher score.

---

> > > ### Author Response · Authors · 2024-11-25
> > >
> > > Thank you for your thoughtful feedback and for recognizing the strength of our theoretical contribution. We truly appreciate your support and are encouraged by your positive reassessment!

---

### Meta-Review · Area_Chair_mKk3 · 2024-12-17

**Metareview:**

This work theoretically justifies why FixMatch-like self-supervised learning methods outperform supervised learning (SL) in generalization for deep networks, showing that FixMatch learns all class features while SL captures only a subset. The authors introduce SA-FixMatch, an enhanced version of FixMatch, validated to improve generalization. Experimental results support the theoretical findings and the effectiveness of SA-FixMatch. The reviewers agree this is a positive contribution to the community and I agree that the paper presents new results that are interesting to the community.

**Additional Comments On Reviewer Discussion:**

The reviewers originally raised various questions, including the validity of the assumptions and how those can be extended beyond 3 layers and the authors have responded to the reviewers.

---

### Decision · Program_Chairs · 2025-01-22

Accept (Oral)